# InftyThink$^+$: Effective and Efficient Infinite-Horizon Reasoning via Reinforcement Learning

**Yuchen Yan**[1][2]  **Liang Jiang**[2]  **Jin Jiang**[3]  **Shuaicheng Li**[2]  **Zujie Wen**[2]
**Zhiqiang Zhang**[2]  **Jun Zhou**[2]  **Jian Shao**[1][*]  **Yueting Zhuang**[1]  **Yongliang Shen**[1][*]

## Abstract

Large reasoning models achieve strong performance by scaling inference-time chain-of-thought, but this paradigm suffers from quadratic cost, context length limits, and degraded reasoning due to lost-in-the-middle effects. Iterative reasoning mitigates these issues by periodically summarizing intermediate thoughts, yet existing methods rely on supervised learning or fixed heuristics and fail to optimize when to summarize, what to preserve, and how to resume reasoning. We propose InftyThink$^+$, an end-to-end reinforcement learning framework that optimizes the entire iterative reasoning trajectory, building on model-controlled iteration boundaries and explicit summarization. InftyThink$^+$ adopts a two-stage training scheme with supervised cold-start followed by trajectory-level reinforcement learning, enabling the model to learn strategic summarization and continuation decisions. Experiments on DeepSeek-R1-Distill-Qwen-1.5B show that InftyThink$^+$ improves accuracy by 21% on AIME24 and outperforms conventional long chain-of-thought reinforcement learning by a clear margin, while also generalizing better to out-of-distribution benchmarks. Moreover, InftyThink$^+$ significantly reduces inference latency and accelerates reinforcement learning training, demonstrating improved reasoning efficiency alongside stronger performance.

## 1. Introduction

Large reasoning models have demonstrated remarkable performance across a wide range of complex real-world tasks, including mathematical reasoning, logical reasoning, and code reasoning (Guo et al., 2025; OpenAI et al., 2025; Team et al., 2025c;a;b). These gains primarily stem from *inference-time scaling*: by producing exceptionally long chains-of-thought, models can perform problem decomposition, trajectory planning, multi-step reasoning, and self-reflection, thereby exhibiting advanced cognitive capabilities (Chen et al., 2025; OpenAI, 2024).

However, scaling reasoning length under the standard long-context paradigm encounters three fundamental barriers. First, the quadratic complexity of self-attention means that inference cost grows superlinearly with generation length, making very long reasoning traces prohibitively expensive (Vaswani et al., 2017; Liu et al., 2025c). Second, reasoning is hard-bounded by the model's maximum context window; when a problem demands a chain of thought exceeding this limit, generation terminates before reaching any conclusion, leaving the hardest problems unsolvable regardless of available compute (Kuratov et al., 2024). Third, as reasoning traces grow longer, models increasingly suffer from the "lost-in-the-middle" phenomenon, where critical early information becomes inaccessible, degrading reasoning quality even when context limits are not exceeded (Liu et al., 2024; Wang, 2025). We further discuss these phenomena in Appendix B.

These observations have motivated a growing body of work on *iterative reasoning*, where the generation process is periodically interrupted, the accumulated context is compressed or summarized, and reasoning continues with a refreshed, bounded context (Yan et al., 2025; Aghajohari et al., 2025). This paradigm promises to decouple reasoning depth from context length, enabling models to reason indefinitely while maintaining bounded computational cost per step. We conduct an efficiency analysis in Appendix D.

Yet existing iterative reasoning methods fail to address three fundamental questions: *when* to compress, *how* to compress, and *how to resume* after compression. Approaches based on token pruning or latent compression (Xia et al., 2025; Zhang et al., 2025) risk discarding information that later proves critical. The Markovian Thinker (Aghajohari et al., 2025) applies RL to fixed-size chunks, achieving linear compute

---

*Corresponding author. [1]Zhejiang University [2]Ant Group [3]Peking University. Correspondence to: Yuchen Yan <yanyuchen@zju.edu.cn>, Yongliang Shen <syl@zju.edu.cn>.

## Vanilla Reasoning Paradigm

## InftyThink Reasoning Paradigm

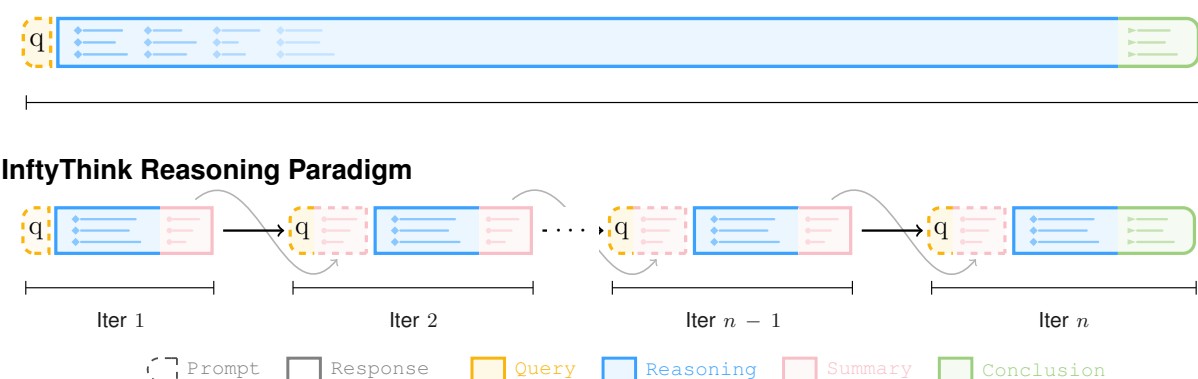

*Figure 1.* **InftyThink** reasoning paradigm VS. Vanilla reasoning paradigm. **Upper panel:** The vanilla reasoning paradigm generates a single, continuous long chain-of-thought in one pass. **Lower panel:** The InftyThink reasoning paradigm decomposes reasoning into multiple iterative rounds, where consecutive iterations are connected via self-generated global summaries.

scaling but imposing rigid boundaries that ignore the natural structure of reasoning. InftyThink (Yan et al., 2025) allows models to autonomously decide when to summarize, but relies exclusively on supervised fine-tuning: the model learns to *format* iterative reasoning by imitating training data. This analysis reveals a key insight: ***what makes iterative reasoning effective is not the format itself, but the ability to make optimal decisions at each iteration***. When to summarize, what to preserve, how to continue: these are sequential decisions with long-horizon consequences. A poor early summary can doom all subsequent reasoning; an unnecessary iteration wastes compute; a premature conclusion sacrifices accuracy. These tradeoffs demand trajectory-level optimization that supervised learning cannot provide.

We introduce **InftyThink$^+$**, an end-to-end reinforcement learning framework that directly optimizes the complete iterative reasoning trajectory. Building on InftyThink's paradigm of model-controlled iteration boundaries and explicit summarization (shown in Figure 1), our approach proceeds in two stages: a cold-start stage that uses supervised fine-tuning to establish the basic iterative reasoning format, followed by an RL stage that optimizes strategic decisions through trajectory-level learning. We carefully design the rollout strategy, reward formulation, and policy gradient estimation tailored to InftyThink's single-trajectory, multi-inference structure. This design separates format acquisition from strategy optimization, enabling the model to learn not only how to produce iterative reasoning, but also when to summarize, what to preserve, and how to effectively leverage self-generated summaries across iterations.

To demonstrate the effectiveness of InftyThink$^+$ in enhancing both reasoning performance and efficiency, we conduct extensive empirical experiments. We evaluate our method on DeepSeek-R1-Distill-Qwen-1.5B. On AIME24,

InftyThink$^+$ improves accuracy by $21\%$, and achieves an additional gain of $9\%$ compared to conventional long CoT–based reinforcement learning. On out-of-distribution GPQA_diamond benchmark, InftyThink$^+$ improves accuracy by $5\%$, and achieves an additional gain of $4\%$ than vanilla approach. In terms of inference efficiency, on AIME25, InftyThink$^+$ reduces reasoning latency by $32.8\%$ compared to the vanilla approach. This improvement consistently generalizes to larger-scale models, such as Qwen3-4B-Base, and extends to out-of-distribution tasks, including code and scientific reasoning. Moreover, InftyThink$^+$ yields a $18.2\%$ speedup in RL training, demonstrating a more efficient utilization of training resources. Our contributions can be summarized as follows[1]:

- We introduce reinforcement learning into the iterative reasoning paradigm, enabling end-to-end optimization of when to summarize, what to preserve, and how to continue across iterations.

- We develop InftyThink$^+$, comprising trajectory-level optimization with shared advantages, efficiency-aware reward shaping, and a cold-start training protocol.

- We demonstrate that InftyThink$^+$ consistently outperforms both SFT-based iterative reasoning and standard long-context RL, with analysis revealing adaptive iteration that emerge from trajectory-level optimization.

**Conflict of Interest.** The authors Yuchen Yan (as an intern), Liang Jiang, Shuaicheng Li, Zujie Wen, Zhiqiang Zhang and Jun Zhou are employed by Ant Group, which leads the development of Algorithm IcePop, which was used for stable RL training in this paper.

---

[1]Codes and models of InftyThink$^+$ are available at https://zju-real.github.io/InftyThink-Plus

## 2. Related Work

### 2.1. Reinforcement Learning for LLM Reasoning

Reinforcement learning (RL) has emerged as the dominant training paradigm for frontier reasoning models. By performing large-scale rollouts and assigning rewards to generated trajectories, RL guides models to converge toward more correct reasoning paths, thereby improving performance on reasoning tasks. Existing RL-based approaches for reasoning models can be broadly categorized into three lines of work. (1) **Data-centric methods**: these approaches focus on constructing more comprehensive and effective queries and verification schemes, providing RL with diverse, high-quality training samples (Albalak et al., 2025; He et al., 2025; Hu et al., 2025; Yu et al., 2025b). (2) **Reward-centric methods**: this line of work designs task-specific reward functions to optimize different objectives, such as reasoning accuracy, computational efficiency, or generation length (Dong et al., 2025; Shao et al., 2025; Wu et al., 2025b). (3) **Policy-gradient optimization methods**: these approaches develop practical RL algorithms to make optimization more stable and precise, reducing variance and improving convergence behavior (Guo et al., 2025; Yu et al., 2025b; Zheng et al., 2025b; Tang et al., 2025). Building upon existing RL datasets, InftyThink+ tailors both rollout and reward designs to the InftyThink (Yan et al., 2025) reasoning paradigm, and further proposes a gradient update scheme specifically adapted to the *single-trajectory, multi-generation* setting.

### 2.2. Context Management for Long-horizon Reasoning

Reasoning models exhibit a distinctive generation pattern in which they produce exceptionally long reasoning content. Through repeated decomposition, planning, inference, and reflection, these models achieve improved reasoning performance (Wang et al., 2025; Wu et al., 2024). However, a fundamental challenge faced by current reasoning models lies in their limited context window, which constrains their reasoning capability (Kuratov et al., 2024). This limitation becomes particularly severe in long-horizon agentic tasks, where the effective context budget is further reduced by extended interaction histories (Mei et al., 2025).

Existing efforts to mitigate this issue can be broadly categorized into two directions: *input-side context management* and *output-side context management*. On the input side, prior work focuses on compressing the available context by techniques such as generating summaries or discarding earlier reasoning (e.g., prior CoT tokens), thereby reserving more space for subsequent reasoning (Wu et al., 2025a; Xu et al., 2025; Yu et al., 2025a). In contrast, output-side context management requires online processing of generated reasoning tokens during inference. Representative approaches include removing low-information tokens or seg-

menting a long reasoning trajectory into multiple shorter reasoning segments, effectively expanding the usable context horizon (Aghajohari et al., 2025; Xia et al., 2025; Yan et al., 2025). InftyThink (Yan et al., 2025) belongs to the latter category, using explicit textual summaries to propagate information across iterations. While prior work on InftyThink relies on supervised learning with heuristic data construction, InftyThink+ introduces end-to-end RL optimization, enabling the model to learn effective summarization and continuation strategies through trajectory-level feedback.

## 3. Methods

In this section, we present the complete training recipe for **InftyThink+**. We first introduce the InftyThink reasoning paradigm that serves as the foundation of our approach (Section 3.1). We then describe the cold-start procedure, which enables the model to acquire the fundamental InftyThink reasoning format (Section 3.2). Finally, we detail the reinforcement learning strategy that optimizes the complete reasoning trajectory end-to-end (Section 3.3).

### 3.1. InftyThink Reasoning Paradigm

We first contrast the vanilla reasoning paradigm with the InftyThink reasoning paradigm to clarify the structural differences that motivate our approach.

**Vanilla Reasoning Paradigm.** Following the reasoning format popularized by DeepSeek-R1 (Guo et al., 2025), most existing reasoning models produce outputs consisting of two parts: a long reasoning trace enclosed by `<think>` and `</think>` tags, containing detailed step-by-step analysis; and a concise conclusion presenting the final answer. While effective, this paradigm couples reasoning depth directly to context length, inheriting all three barriers discussed in Section 1.

**InftyThink Reasoning Paradigm.** InftyThink decouples reasoning depth from context length by distributing the reasoning process across multiple iterations connected through explicit summaries. For a query $q$, at each reasoning round $i$, the model conditions on the summary $s_{i-1}$ from the previous iteration, generates reasoning $r_i$ for the current iteration, and produces an updated summary $s_i$. This process repeats iteratively until the model autonomously terminates by generating a conclusion $c$ instead of a summary. We provide a detailed description in Appendix C.

The key distinction from vanilla reasoning is that each iteration operates within a bounded context window: the model sees only the original query and the most recent summary, not the full reasoning history. This design achieves two goals simultaneously: it bounds the per-iteration computational cost regardless of total reasoning depth, and it forces

the model to distill essential information into summaries that must support all subsequent reasoning.

## 3.2. Cold Start

Before applying RL, we perform a cold-start stage that teaches the model the basic format of InftyThink-style reasoning. Specifically, we transform existing supervised data into InftyThink format and fine-tune the model to produce multi-iteration outputs with explicit summaries.

**Data Transformation.** We transform existing vanilla reasoning data into InftyThink format through a three-step process. Given a vanilla triple $(q, r, c)$ consisting of query, reasoning trace, and conclusion, we first partition $r$ into segments $\{r_1, \ldots, r_n\}$ using a hyperparameter $\eta$ that bounds segment length while preserving semantic coherence at sentence boundaries. We then employ a general-purpose language model to generate summaries $\{s_1, \ldots, s_{n-1}\}$, where each summary $s_i$ is conditioned on the previous summary $s_{i-1}$ and current reasoning $r_i$, matching the information flow at inference time. A hyperparameter $\gamma$ constrains summary length to ensure compression, ensuring that the number of tokens does not exceed $\gamma$, thereby enabling efficient summarization. The transformation yields training instances:

$$(q, r, c) \xrightarrow{\eta, \gamma} \begin{cases} (q, r_1, s_1) & \text{for } i = 1, \\ (q, s_{i-1}, r_i, s_i) & \text{for } 1 < i < n, \\ (q, s_{n-1}, r_n, c) & \text{for } i = n. \end{cases} \quad (1)$$

We provide a more detailed description of the data paradigm transformation pipeline in E.1.

**Supervised Initialization.** We augment the tokenizer with special tokens (`<summary>`, `</summary>`, `<history>`, `</history>`) and perform supervised fine-tuning on the transformed data. During training, loss is computed only over reasoning and summary tokens; query and history tokens are masked. After this stage, the model can produce syntactically valid InftyThink outputs, but it has learned only to imitate the format from training data. The model has not learned to determine the appropriate timing for summarization, identify which information is essential to preserve, or adapt the number of iterations to problem difficulty. These capabilities require trajectory-level optimization, which we address through reinforcement learning. We provide a more detailed description of SFT recipe in Appendix E.2.

## 3.3. Reinforcement Learning

The cold-start stage teaches format; reinforcement learning teaches strategy. We now describe how RL is adapted to the unique structure of InftyThink reasoning, where a single

problem induces a trajectory of multiple generations connected through summaries. The complete recipe for RL is provided in Appendix F.

**Trajectory-Level Rollout.** A key challenge in applying RL to InftyThink is that optimizing a single query requires rolling out the complete multi-iteration trajectory. We introduce a hyperparameter $\varphi$ that bounds the maximum number of iterations to ensure training efficiency. Given query $q$, we roll out the model iteratively: at each iteration $j$, we construct the prompt from $q$ and the previous summary $s_{j-1}$ (empty if $j = 1$), generate output $o_j$, and extract any summary for the next iteration. Rollout terminates when: (i) the model produces a conclusion instead of a summary, (ii) the model fails to produce valid InftyThink format, or (iii) the iteration count reaches $\varphi$. The $i$-th sampled trajectory for query $q$ is denoted as:

$$\mathcal{O}_i = \{o_i^1, o_i^2, \ldots, o_i^{n_i}\}, \quad n_i \leq \varphi, \quad (2)$$

where $o_i^j$ represents the output at the $j$-th iteration and $n_i$ is the total number of iterations in trajectory $i$.

**Reward Design.** In RL, a critical step is to assign rewards to the sequences being optimized, thereby evaluating model rollouts and guiding the policy to move toward or away from particular behaviors. In this work, we primarily introduce two types of rewards: a *task reward*, which assesses whether the model successfully solves the given problem, and an *efficiency reward*, which measures how efficiently the model arrives at a solution. Our reward assignment is performed at the trajectory level: for a trajectory $\mathcal{O}_i$, all round-wise outputs $o_i^j$ share the same scalar reward.

The **task reward** evaluates correctness by verifying the final output against the ground truth:

$$\mathcal{R}_{\text{task}}(\mathcal{O}_i) = \mathbb{I}\left[\text{Verify}(o_i^{n_i}, \text{gt}) = \text{Correct}\right], \quad (3)$$

where $o_i^{n_i}$ denotes the final output of trajectory $\mathcal{O}_i$, gt is the ground-truth answer, $\text{Verify}(\cdot, \cdot)$ is a problem-specific verification function, and $\mathbb{I}[\cdot]$ is the indicator function that returns 1 if the condition holds and 0 otherwise.

The **efficiency reward** encourages solving problems in fewer iterations when possible. We adopt a quadratic decay that penalizes additional iterations more heavily as the count grows:

$$\mathcal{R}_{\text{eff}}(\mathcal{O}_i) = 1 - \left(\frac{n_i - 1}{\varphi}\right)^2, \quad (4)$$

where $n_i$ is the number of iterations in trajectory $\mathcal{O}_i$ and $\varphi$ is the maximum allowed iterations. This reward takes values in $(0, 1]$, achieving its maximum of 1 when $n_i = 1$ and decreasing monotonically as $n_i$ increases. The quadratic form provides mild penalties for early iterations, allowing

exploration, while increasingly discouraging unnecessary iterations as the count approaches $\varphi$.

Following Shao et al. (2025), we combine the two rewards multiplicatively:

$$\mathcal{R}(\mathcal{O}_i) = \mathcal{R}_{\text{task}}(\mathcal{O}_i) \cdot \mathcal{R}_{\text{eff}}(\mathcal{O}_i), \tag{5}$$

This formulation ensures that efficiency rewards only affect correct trajectories: incorrect solutions receive zero reward regardless of iteration count, preventing the model from learning to terminate prematurely at the cost of accuracy.

**Policy Gradient.** We adopt Group Relative Policy Optimization (GRPO) (Shao et al., 2024) as our base RL algorithm. For a given query $q$, we sample $G$ reasoning trajectories, each consisting of multiple rounds of generation. All outputs across all iterations are optimized jointly using token-level loss averaging (Yu et al., 2025b):

$$\mathcal{J}(\theta) = \mathbb{E}_{\{\mathcal{O}_i\}_{i=1}^{G} \sim \pi_{\theta_{\text{old}}}(\cdot|q), \{o_i^j\}_{j=1}^{n_i} \sim \mathcal{O}_i}$$

$$\left[ \frac{1}{\sum_{i=1}^{G} \sum_{j=1}^{n_i} |o_i^j|} \sum_{i=1}^{G} \sum_{j=1}^{n_i} \overbrace{\mathcal{U}(o_i^j; \theta)}^{\text{round loss}} \right], \tag{6}$$

where $|o_i^j|$ denotes the number of tokens in output $o_i^j$, and $\mathcal{U}(o; \theta)$ is the clipped surrogate objective:

$$\mathcal{U}(o; \theta) = \sum_{t=1}^{|o|} \min\left( r_\theta(o_t)\hat{A}_t, \text{clip}(r_\theta(o_t), 1 - \epsilon^-, 1 + \epsilon^+)\hat{A}_t \right), \tag{7}$$

Here $r_\theta(o_t) = \pi_\theta(o_t)/\pi_{\theta_{\text{old}}}(o_t)$ is the importance sampling ratio for token $o_t$, $\epsilon^-$ and $\epsilon^+$ are clipping thresholds, and $\hat{A}_t$ is the advantage estimate for token $t$.

Critically, advantages are computed at the trajectory level and shared across all iterations within a trajectory. For any token $t$ in output $o_i^j$ belonging to trajectory $\mathcal{O}_i$, the advantage is:

$$\hat{A}_t = \frac{\mathcal{R}(\mathcal{O}_i) - \mu}{\sigma}, \tag{8}$$

where $\mu$ and $\sigma$ are the mean and standard deviation of rewards computed over all $G$ trajectories sampled for query $q$. This shared advantage design reflects the key insight that early iterations contribute to final success: a high-quality first summary that enables correct reasoning in later iterations receives positive gradient signal, even though the summary itself does not directly produce the answer.

**Training Stability.** In practice, the rollout phase and parameter update phase often use different computational backends for efficiency (Sheng et al., 2025). Following Team et al. (2025c); Liu et al. (2025a); Zheng et al. (2025a), we apply token-level gradient masking (IcePop) (Team et al., 2025c) to exclude tokens whose log probabilities differ substantially between the inference engine and the training engine, improving the robustness of RL training.

## 4. Experiments

### 4.1. Experimental Setup.

**Training.** We conduct experiments on two base models: DeepSeek-R1-Distill-Qwen-1.5B (Guo et al., 2025), which is distilled from DeepSeek-R1, and Qwen3-4B-Base (Yang et al., 2025), a pretrained model without post-training. All experiments based on DeepSeek-R1-Distill-Qwen-1.5B were conducted on 8 Nvidia GPUs, while all experiments using Qwen3-4B-Base were carried out on 32 GPUs.

For cold start, we adopt OpenThoughts-114K (Guha et al., 2025) as the training corpus. To convert vanilla reasoning trajectories into the InftyThink-style paradigm, we use Qwen3-4B-Instruct-2507 (Yang et al., 2025) to generate intermediate summaries, with the hyperparameters set to $\eta = 6k$ and $\gamma = 1k$. Model training is implemented with ms-swift (Zhao et al., 2025) and Megatron-Core (Shoeybi et al., 2020) as the backend. Detailed configurations for SFT are provided in the Appendix G.1.1.

For RL, we train on the DeepScaleR-Preview (Luo et al., 2025) dataset with a global batch size of 128 for 1,000 steps (500 steps for Qwen3-4B-Base). We employ the verl (Sheng et al., 2025) framework with AgentLoop to enable asynchronous inference, using SGLang (Zheng et al., 2024) as the inference backend and FSDP (Zhao et al., 2023) as the training backend. For the task reward, we adopt the verification scripts provided by PRIME-Math (Cui et al., 2025). For RL training under the InftyThink$^+$ paradigm, we set the maximum number of rollout iterations $\varphi$ to 5. In Appendix O, we present an ablation study of key hyperparameters. Additional implementation details for RL are deferred to the Appendix G.1.2. We also provide the detailed RL training dynamics in Appendix H, and a stability analysis for both training and evaluation in Appendix J.

**Evaluation.** We evaluate all models both before and after training on a comprehensive set of benchmarks, including in-distribution benchmarks MATH500 (Hendrycks et al., 2021; Lightman et al., 2023), AIME24, and AIME25, as well as the out-of-distribution benchmark GPQA_Diamond (Rein et al., 2024). All evaluations are conducted using SGLang for inference, with CompassVerifier-7B (Liu et al., 2025b) serving as the evaluator. To mitigate evaluation variance, all reported metrics are averaged over 32 generations, with the sampling temperature set to 0.7 and $\texttt{top\_p}$ set to 0.95. Detailed evaluation settings are provided in the Appendix G.2.

*Table 1.* Our main experimental results. The results are obtained by sampling the model 32 times with a temperature of 0.7. ACC stands for average accuracy (%), TOK stands for average number of generated tokens (K), and LAT stands for average inference time in seconds. ✗ denotes the setting with cold start only, without RL. ✓ T denotes the RL setting where only the task reward is used. ✓ T+E denotes the RL setting where both the task reward and the efficiency reward are used.

| RL | MATH500 ACC↑ | TOK | LAT↓ | AIME24 ACC↑ | TOK | LAT↓ | AIME25 ACC↑ | TOK | LAT↓ | GPQA_diamond ACC↑ | TOK | LAT↓ | Average ACC↑ | TOK | LAT↓ |
|---|---|---|---|---|---|---|---|---|---|---|---|---|---|---|---|
| | | | | | | | **Base Model: DeepSeek-R1-Distill-Qwen-1.5B** | | | | | | | | |
| *Vanilla* | | | | | | | | | | | | | | | |
| ✗ | 86.20 | 5.32 | 48.71 | 26.67 | 17.08 | 158.95 | 24.48 | 15.53 | 134.34 | 29.40 | 10.45 | 101.84 | 41.69 | 12.10 | 110.96 |
| ✓ T | 89.63 | 5.93 | 56.05 | 38.75 | 18.26 | 175.00 | 31.04 | 18.11 | 169.38 | 29.81 | 15.48 | 197.33 | 47.31 | 14.45 | 149.44 |
| Δ | +3.43 | +0.62 | +7.34 | +12.08 | +1.18 | +16.05 | +6.56 | +2.58 | +35.04 | +0.41 | +5.03 | +95.49 | +5.62 | +2.35 | +38.48 |
| *InftyThink+* | | | | | | | | | | | | | | | |
| ✗ | 86.54 | 5.77 | 34.82 | 29.48 | 20.23 | 103.04 | 27.92 | 19.18 | 98.10 | 32.31 | 11.77 | 74.31 | 44.06 | 14.24 | 77.57 |
| ✓ T | **91.56** | 6.10 | 34.26 | **50.94** | 23.36 | 102.85 | **35.83** | 26.34 | 113.78 | **37.50** | 24.27 | 149.93 | **53.96** | 20.02 | 100.21 |
| Δ | +5.02 | +0.33 | -0.56 | +21.46 | +3.13 | -0.19 | +7.91 | +7.16 | +15.68 | +5.19 | +12.50 | +75.62 | +9.89 | +5.78 | +22.64 |
| ✓ T+E | 89.96 | 3.36 | **17.71** | 43.96 | 13.13 | **57.50** | 32.92 | 7.45 | **68.39** | 35.46 | 8.69 | **49.87** | 50.58 | 10.66 | **48.37** |
| Δ | +3.42 | -2.41 | -17.11 | +14.48 | -7.10 | -45.54 | +5.00 | -1.73 | -29.71 | +3.15 | -3.08 | -24.44 | +6.51 | -3.58 | -29.20 |
| | | | | | | | **Base Model: Qwen3-4B-Base** | | | | | | | | |
| *Vanilla* | | | | | | | | | | | | | | | |
| ✗ | 91.97 | 4.52 | 139.6 | 44.06 | 15.02 | 439.62 | 33.65 | 14.93 | 448.98 | 45.65 | 8.10 | 254.73 | 53.83 | 10.64 | 320.73 |
| ✓ T | 92.89 | 6.32 | 254.09 | 50.31 | 17.16 | 571.18 | 38.31 | 18.70 | 733.78 | 47.02 | 12.32 | 579.22 | 57.13 | 13.63 | 534.57 |
| Δ | +0.92 | +1.80 | +114.49 | +6.25 | +2.13 | +131.56 | +4.66 | +3.78 | +284.80 | +1.37 | +4.22 | +324.49 | +3.30 | +2.98 | +213.84 |
| *InftyThink+* | | | | | | | | | | | | | | | |
| ✗ | 91.99 | 4.64 | 85.66 | 43.65 | 16.14 | 242.66 | 34.38 | 16.55 | 250.33 | 44.65 | 8.05 | 166.54 | 53.67 | 11.35 | 186.30 |
| ✓ T | **94.09** | 6.01 | 120.16 | **52.29** | 21.44 | 319.15 | **39.48** | 23.41 | 349.12 | **48.99** | 11.89 | 272.24 | **58.71** | 15.69 | 265.17 |
| Δ | +2.10 | +1.36 | +34.50 | +8.64 | +5.30 | +76.49 | +5.10 | +6.87 | +98.79 | +4.34 | +3.84 | +105.70 | +5.04 | +4.34 | +78.87 |
| ✓ T+E | 92.64 | 3.41 | **58.67** | 49.06 | 13.46 | 185.79 | 36.77 | 16.82 | **217.94** | 48.17 | 7.58 | 156.09 | 56.66 | 10.32 | **154.62** |
| Δ | +0.65 | -1.23 | -26.99 | +5.41 | -2.69 | -56.87 | +2.39 | +0.27 | -32.39 | +3.52 | -0.48 | -10.45 | +2.99 | -1.03 | -31.68 |

**Extended Experiments and Analyses** We extend experiments and analyses in the Appendix, covering:

- In Appendix I, we additionally report observations on a broader set of benchmarks (code reasoning and scientific reasoning), along with the model's performance throughout the RL training process.

- In Appendix M, we study how model performance evolves across reasoning iterations.

- In Appendix N, we characterize the sample-level inference latency distribution.

- In Appendix Q, we also provide a detailed comparison with Delethink (Aghajohari et al., 2025).

### 4.2. Main Results

**InftyThink+ Amplifies RL Benefits.** InftyThink+ consistently magnifies the effectiveness of reinforcement learning compared to the Vanilla setting. Under task-only RL (✓ T), InftyThink+ achieves substantially larger accuracy gains across all benchmarks, with the average ACC improvement reaching +9.89, compared to +5.62 for Vanilla. This gap is particularly striking on harder benchmarks such as AIME24, where InftyThink+ gains +21.46 points versus +12.08 for Vanilla, indicating that structured iterative summaries provide a more exploitable substrate for RL to improve correctness. These results suggest that RL does not merely encourage longer reasoning, but can more effectively optimize reasoning quality when intermediate summaries explicitly expose reusable high-level states.

**InftyThink+ Extends Reasoning Depth and Decreases Inference Latency.** Beyond accuracy, InftyThink+ fundamentally reshapes the trade-off between reasoning depth and inference cost. Even before RL, InftyThink+ already reduces latency compared to Vanilla (e.g., average LAT 77.57 vs. 110.96), despite using slightly more tokens, indicating more efficient downstream reasoning enabled by summaries. This efficiency gain stems from the bounded per-iteration context: instead of attending over an ever-growing sequence, each iteration operates within a fixed context window. After task-only RL, InftyThink+ allows the model to extend reasoning depth, reflected in increased TOK, while largely preserving latency on several benchmarks (e.g., near-zero LAT change on MATH500 and AIME24). This contrasts sharply with Vanilla, where deeper reasoning directly translates into severe latency inflation, showing that summarized

iterative reasoning decouples reasoning depth from wall-clock inference time.

**Efficiency Reward Enables a Better Trade-off.** When efficiency reward is further introduced (✓ **T+E** ), InftyThink$^+$ achieves a significantly better effectiveness–efficiency balance. Compared to the cold-start baseline, the T+E configuration improves average accuracy by +6.51 points while simultaneously reducing latency by 29.20 seconds (from 77.57s to 48.37s). Compared to task-only RL, it trades a modest accuracy decrease ($53.96 \rightarrow 50.58$ average) for substantial efficiency gains ($100.21s \rightarrow 48.37s$ latency, $20.02K \rightarrow 10.66K$ tokens). This demonstrates that efficiency-aware RL successfully guides the model to generate more compact summaries and terminate reasoning earlier without collapsing performance. Overall, these results confirm that combining InftyThink$^+$ with multi-objective RL enables controllable reasoning policies that are not only more accurate, but also substantially more efficient.

# 5. Analyses

Through end-to-end optimization, InftyThink$^+$ enables reasoning models to acquire *effective* and *efficient* iterative reasoning capabilities. In this section, we analyze the impact of InftyThink$^+$ from two complementary perspectives: effectiveness (Section 5.1) and efficiency (Section 5.2).

## 5.1. InftyThink$^+$ Enables More Effective Iterative Reasoning

Effective iterative reasoning is challenged by three key questions. *When to compress* determines the appropriate timing for abstraction, influencing the trade-off between reasoning depth and information loss. *How to compress* defines the mechanism by which essential reasoning states are distilled and propagated to subsequent iterations. *How to continue* specifies how the model conditions future reasoning steps on the compressed representations to ensure consistent and progressive inference. In the following, we analyze the effects of InftyThink$^+$ from each of these three perspectives.

### 5.1.1. LEARNING WHEN TO COMPRESS

To analyze the practical effect of InftyThink$^+$ on learning *when to compress*, we design an ablation study. Specifically, for models following the InftyThink reasoning paradigm with $\eta = 6k$, we introduce two alternative reasoning interruption strategies. The first is **Fixed**, where the model is forcibly interrupted after generating a fixed number of tokens and then required to produce a summary; in our experiments, this threshold is set to 5k tokens. The second is **Random**, where the model is interrupted after generating a random number of reasoning tokens before summarization, with the token budget sampled as

`random.randint(3000, 6000)`. We compare these two strategies against the adaptive interruption mechanism employed by InftyThink$^+$. The benchmark performance of all variants is reported in Table 2.

*Table 2.* Comparison of benchmark performance (%) across different summary timing strategies.

| Strategy | AIME24 | AIME25 | AMC23 |
|---|---|---|---|
| *w/o RL* | | | |
| InftyThink | 29.48 | 27.92 | 71.64 |
| Random | 28.54 -0.94 | 26.25 -1.67 | 72.58 +0.94 |
| Fixed | 28.44 -1.04 | 26.04 -1.88 | 72.03 +0.39 |
| *w RL* | | | |
| InftyThink+ | 50.94 | 35.83 | 85.86 |
| Random | 47.92 -3.02 | 33.83 -2.00 | 84.16 -1.70 |
| Fixed | 48.44 -2.50 | 33.00 -2.83 | 84.53 -1.33 |

**Adaptive timing is consistently superior.** In both w/o RL and w RL settings, InftyThink's adaptive timing outperforms Random and Fixed strategies. Without RL, non-adaptive timing causes clear drops on AIME24 (-0.94 to -1.04) and AIME25 (-1.67 to -1.88), with only marginal changes on AMC23 (+0.39 to +0.94), showing that static or random timing cannot reliably match the reasoning progress.

**RL strengthens timing selection.** With RL, overall accuracy increases, but the penalty for incorrect timing becomes larger. Under InftyThink$^+$, Random and Fixed timing lead to larger degradations on AIME24 (-2.50 to -3.02), AIME25 (-2.00 to -2.83), and consistent drops on AMC23 (-1.33 to -1.70). This indicates that *RL helps the model learn a more precise policy for when to summarize*, making adaptive timing increasingly critical.

### 5.1.2. LEARNING HOW TO COMPRESS

*How to compress* is crucial because it determines whether the summary can faithfully preserve the key intermediate conclusions and constraints needed for subsequent reasoning. To analyze the quality of the summaries generated by InftyThink$^+$ models, we design a controlled replacement experiment. Specifically, during inference, we replace the summaries autonomously produced by the model with summaries generated by an external LLM Qwen3-4B-Instruct-2507, following the same procedure used in cold-start data construction in Appendix E.1. We then evaluate the resulting performance changes on downstream benchmarks, with the results reported in Table 3.

Under the SFT-only setting (w/o RL), replacing the internally generated summaries with external summaries leads to consistent performance gains across all benchmarks: accuracy on AIME24 increases from 29.48% to 32.40%. These improvements suggest that SFT primarily teaches the model

*Table 3.* Comparison of benchmark performance (%) with different summarizers.

| Summarizer | AIME24 | AIME25 | AMC23 |
|---|---|---|---|
| *w/o RL* | | | |
| Internal | 29.48 | 27.92 | 71.64 |
| External | 32.40  +2.92 | 28.75  +0.83 | 73.75  +2.11 |
| *w RL* | | | |
| Internal | 50.94 | 35.83 | 85.86 |
| External | 48.42  -2.52 | 33.63  -2.20 | 84.62  -1.24 |

to adhere to the InftyThink procedural format, rather than instilling the ability to produce accurate and informative summaries. In contrast, under RL-trained settings (w/ RL), substituting internal summaries with external ones consistently degrades performance, with AIME24 dropping from 50.94% to 48.42%. This reversal suggests that RL enables the model to learn summary generation as an end-to-end policy component that is tightly coupled with downstream reasoning, leading to more effective summaries and, consequently, improved overall performance.

### 5.1.3. LEARNING HOW TO CONTINUE

*How to continue* determines whether the model can coherently leverage the compressed summary to resume reasoning without semantic drift or logical gaps, directly affecting response correctness. To verify that InftyThink$^+$ enables models with better continuation reasoning, we extract the summary from each reasoning iteration produced by an InftyThink$^+$ model and feed it into a vanilla-paradigm reasoning model, DeepSeek-R1-Distill-Qwen-1.5B, which is then tasked with continuing the reasoning process. We show this ablation result in Figure 2, the four blue bars (from left to right) correspond to continuation reasoning conditioned on the summaries from the 1st, 2nd, 3rd, and 4th iterations, respectively. The darker segments indicate the proportion of instances that InftyThink has already correctly solved, while the lighter segments represent additional correct cases obtained by vanilla continuation reasoning.

First, as shown in Figure 2(b), even when conditioned on InftyThink$^+$ summaries, vanilla continuation suffers from noticeable performance degradation, indicating that InftyThink$^+$ models are better at leveraging summaries to resume reasoning. Second, the additional gains achieved by vanilla continuation diminish as summaries are taken from later iterations; in Figure 2(b), performance gain nearly saturates after the 2nd iteration, suggesting that continuation from late-stage summaries is intrinsically more challenging. InftyThink$^+$ consistently translates summaries from later iterations into monotonic performance improvements, underscoring that *how to continue*, the strategy for resuming reasoning from compressed context, must be learned end-to-end for effective iterative reasoning.

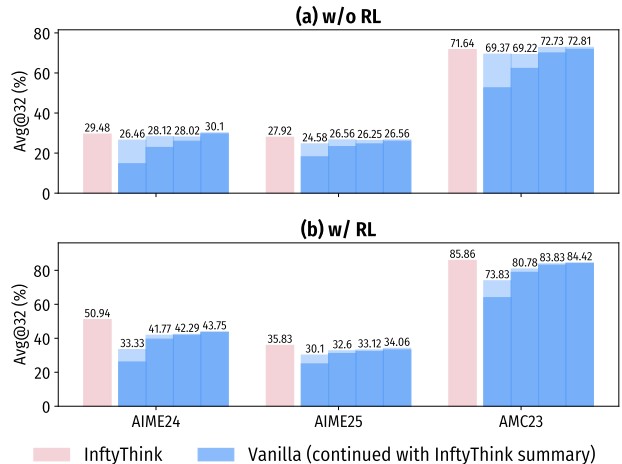

*Figure 2.* Performance of vanilla reasoning when using InftyThink summaries as context. The four blue bars (from left to right) correspond to continuation reasoning conditioned on the summaries from the 1st, 2nd, 3rd, and 4th iterations, respectively. The darker segments indicate the proportion of instances that InftyThink has already correctly solved, while the lighter segments represent additional correct cases obtained by vanilla continuation reasoning.

### 5.2. InftyThink+ Enables More Efficient Iterative Reasoning

As shown in Table 1, InftyThink$^+$ substantially reduces inference latency, achieving an average reduction of 30%–40%. Moreover, the introduction of an efficiency reward further amplifies this effect, leading to a latency reduction of 60%–70%. These gains stem from the $O(n \cdot \ell^2)$ complexity of iterative reasoning versus $O(L^2)$ for vanilla, as analyzed in Appendix K.2. The efficiency gains brought by InftyThink$^+$ are not limited to test-time inference; they also manifest during training. Specifically, RL under the InftyThink$^+$ paradigm enables faster rollouts and more efficient model updates. We present a comparison of RL training time in Figure 3, which clearly illustrates this advantage.

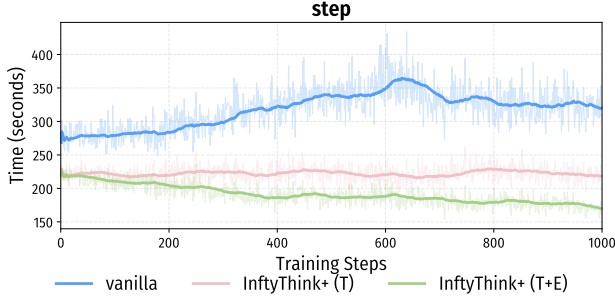

*Figure 3.* Per-step training time over the course of RL training.

Due to the efficient reasoning property of InftyThink, RL training under InftyThink$^+$ is substantially faster than

vanilla long-context RL. Specifically, vanilla long-context RL incurs an average cost of 300 seconds per step, whereas InftyThink$^+$ RL reduces this to 225 seconds per step, yielding an approximately 25% speedup. Moreover, introducing an efficiency reward further improves training efficiency, with the per-step time gradually decreasing over the course of training to an average of 175 seconds, corresponding to an approximately 40% speedup. In the current landscape where RL has become the dominant training paradigm for reasoning models, InftyThink$^+$ provides a more efficient training framework, enabling researchers to train on more data and perform more extensive optimization under the same computational budget. Further analysis is provided in Appendix H.1.

## 6. Conclusion

We propose InftyThink$^+$, an end-to-end RL framework that optimizes iterative reasoning at the trajectory level. By separating format learning from strategy optimization, InftyThink$^+$ enables models to learn when to compress, how to compress, and how to continue effectively. Experiments show consistent accuracy gains over SFT-based iterative reasoning and standard long-context RL, while significantly reducing inference latency. These improvements arise from learned adaptive behaviors rather than heuristics, demonstrating the importance of trajectory-level optimization. We further discuss the limitations of InftyThink$^+$ and its future directions in Appendix A.

## Acknowledgements

This work was supported by National Key Research and Development Project (No. 2024YFB3312900), National Natural Science Foundation of China (No. 62436007), National Natural Science Foundation of China (No. 62506332) and Ant Group Research Intern Program.

## Impact Statement

This paper presents InftyThink+, a framework for improving the effectiveness and efficiency of iterative reasoning in large language models. The primary societal benefits include reduced computational costs and lower energy consumption for reasoning tasks.

We acknowledge several considerations regarding broader impact. First, improved reasoning capabilities could be misused to generate more convincing misinformation or to solve problems that enable harmful applications. However, these risks are not unique to our method and apply broadly to advances in language model reasoning. Second, while InftyThink+ reduces per-query computational costs, improved capabilities may increase overall usage, partially offsetting

environmental benefits. Third, our evaluation focuses on mathematical and scientific reasoning benchmarks; the generalization of our findings to other domains requires further investigation.

We believe the benefits of more efficient and capable reasoning systems outweigh these risks, particularly as the research community develops better safeguards and alignment techniques. We will release our code and models to facilitate reproducibility and encourage further research on safe and efficient reasoning systems.

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

# A. General Discussions

In this section, we provide a deeper discussion of InftyThink+. Specifically, we first examine the philosophy underlying the proposed method, highlighting its conceptual connections to human reasoning and learning processes (Appendix A.1). We then analyze the limitations of InftyThink+, discussing the scenarios in which the method may be less effective as well as its inherent constraints (Appendix A.2). Finally, we outline several promising future directions, including potential applications of InftyThink+ to broader reasoning tasks and possible extensions to further improve its effectiveness and efficiency (Appendix A.3).

## A.1. Philosophy Behind InftyThink+

**Alignment with Human Reasoning.**    A core motivation behind InftyThink+ is its strong alignment with how humans perform complex reasoning. Human problem solving rarely unfolds as a single, uninterrupted chain of thought; instead, it naturally alternates between extended reasoning, abstraction, and reflection. At critical moments, humans pause to summarize intermediate conclusions, discard redundant details, and retain only the most salient constraints before continuing. InftyThink+ mirrors this process by explicitly structuring reasoning into iterative phases of generation, compression, and continuation. By allowing the model to decide *when* to compress, *how* to summarize, and *how* to continue reasoning from compressed context, the method encourages a form of abstraction-aware reasoning that more closely resembles human cognitive strategies. This perspective suggests that improved reasoning performance does not solely arise from longer CoT, but from learning to strategically manage and transform intermediate representations during the reasoning process.

**Reinforcement Learning and Human Learning.**    From a broader cognitive perspective, the role of RL in InftyThink+ closely parallels how humans acquire complex problem-solving skills. Humans do not learn by imitating a fixed, canonical reasoning format; instead, we learn through iterative trial and error, gradually internalizing more effective thinking strategies under outcome-driven feedback. In contrast, SFT primarily encourages models to replicate surface-level output patterns or reasoning formats, which is often insufficient for shaping deep, strategic behaviors. By optimizing interruption timing, summary generation and continuation strategies in an end-to-end RL framework, InftyThink+ enables the model to autonomously learn when to abstract and when to expand reasoning, guided jointly by task and efficiency rewards. This process closely resembles the development of human metacognitive abilities, specifically, knowing when to pause and consolidate intermediate conclusions versus when to continue deeper exploration, providing an intuitive explanation for why RL yields systematic improvements beyond pure SFT within the InftyThink+ paradigm.

## A.2. Limitations

**Task-structure assumptions.**    InftyThink+ implicitly assumes that the reasoning process can be decomposed into relatively independent stages, and that the essential information of each stage can be abstracted into a summary serving as an effective intermediate state for subsequent reasoning. While this assumption is well aligned with tasks such as mathematical reasoning and multi-constraint planning, it does not universally hold. For tasks with highly entangled reasoning processes, unclear stage boundaries, or strong reliance on continuous semantic flow, the paradigm of segmented compression and continuation may yield limited benefits.

**Limitations of natural language summaries.**    In the current framework, summaries are represented as unstructured natural language tokens. Although this representation offers high expressive flexibility, it lacks explicit mechanisms to control information organization and constraint strength. As a result, the importance, logical status, and relative priority of information are encoded implicitly in text, requiring the model to reinterpret and rebalance these factors during continuation. Such high-capacity but weakly constrained intermediate representations limit fine-grained control over compression granularity and information fidelity.

**Dependence on cold-start training.**    The InftyThink+ training pipeline relies on a cold-start stage to shift the model into the InftyThink reasoning paradigm. This stage primarily provides structural scaffolding, such as iteration boundaries, summary actions, and continuation formats, rather than directly optimizing reasoning strategies. However, this reliance implies that the framework depends on task-specific cold-start data design, which introduces additional engineering complexity when adapting the method to new domains or task distributions.

### A.3. Future Directions

**Long-Horizon Agentic Reasoning.**   A promising direction is to extend InftyThink+ to more long-horizon agentic tasks, where reasoning unfolds over substantially longer time scales and interaction loops. Many emerging agentic settings, such as deep research, autonomous debugging, or multi-step decision-making, require models to repeatedly invoke tools, retrieve external information, and incorporate intermediate results, leading to extremely long and evolving contexts (Hu et al., 2026; Xu et al., 2025; Yu et al., 2025a). In such scenarios, effective iterative compression and continuation are not merely efficiency optimizations but fundamental enablers for sustained reasoning. InftyThink+ provides a natural foundation for these tasks by explicitly structuring reasoning into multiple compressed iterations, allowing agents to maintain coherence and scalability over prolonged trajectories.

**Fine-grained Summary Representations.**   Another important direction is to explore more fine-grained and expressive summary modeling mechanisms. Beyond textual summaries, future work may investigate latent representations, such as latent tokens, learned memory slots, or hybrid symbolic–continuous summaries, that can capture abstract constraints, intermediate conclusions, or reusable reasoning states more compactly and faithfully. Such representations could be jointly optimized with downstream continuation policies, further strengthening the coupling between how to compress and how to continue. We believe that advancing summary representations will be crucial for pushing iterative reasoning systems toward greater abstraction, robustness, and long-horizon generalization.

## B. Context Hit Analysis

To analyze the practical context-window requirements of reasoning models during inference, we evaluate *DeepSeek-R1-Distill-Qwen-1.5B* under different `max_new_tokens` settings (8k, 16k, 32k, 48k, and 64k) on multiple benchmarks (MATH500, AIME24, AIME25 and AMC23). We report both the *completion rate* and the *accuracy*. The completion rate is defined as the fraction of instances for which the model successfully generates an `eos` token and terminates reasoning within the given token budget.

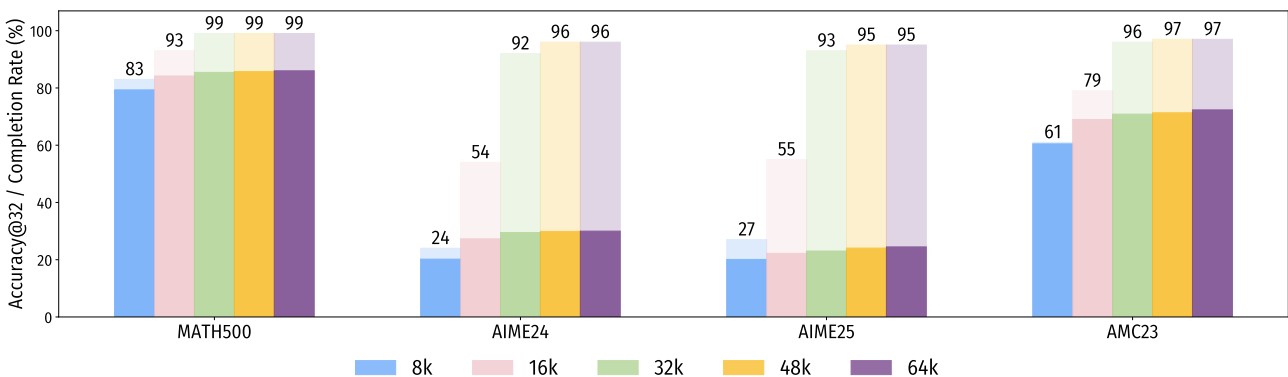

*Figure 4.* Completion rate and accuracy (%) of vanilla long-context reasoning under different `max_new_tokens` settings on benchmarks. Dark bars indicate accuracy, while light bars represent the completion rate.

From Figure 4, we observe that even when the maximum generation length is extended to 32k–64k tokens, the model still fails to complete a subset of highly challenging tasks, such as AIME24 and AIME25. Moreover, a noteworthy phenomenon emerges: under the 48k and 64k settings, the completion rate remains nearly unchanged. This suggests that as the available context length increases, reasoning models begin to suffer from the *lost-in-the-middle* effect, where the model is unable to effectively advance the reasoning process and instead engages in repetitive or unproductive deliberation.

In addition, we emphasize that increasing the generation length leads to a significant degradation in inference efficiency, as reflected by a substantial decrease in tokens generated per second. Taken together, these findings motivate the design of InftyThink+: enabling extended reasoning depth while preserving high inference efficiency.

## C. Detailed Introduction of InftyThink Paradigm

InftyThink is an iterative reasoning paradigm that enhances a model's reasoning depth by decomposing a single long chain-of-thought (CoT) into multiple shorter reasoning segments, while simultaneously reducing computational and memory overhead during inference. To more clearly elucidate the underlying mechanism of InftyThink, this section provides a detailed introduction. Specifically, we describe the model's inputs and outputs at each reasoning iteration and highlight the key differences between the InftyThink paradigm (described in Appendix C.2) and the conventional vanilla reasoning paradigm (described in Appendix C.1).

### C.1. Vanilla Paradigm

Contemporary reasoning models, exemplified by DeepSeek-R1 (Guo et al., 2025) and related models, predominantly adopt a single-round, long-form generation paradigm to solve complex reasoning tasks. Under this paradigm, the model produces an output consisting of two main components: (i) an explicit thinking phase that records the intermediate reasoning trajectory, and (ii) a final conclusion phase that summarizes and presents the solution in a structured form. This conventional reasoning process can be formalized as:

$$<|user|>q<|assistant|>\underline{\texttt{<think>}r\texttt{</think>}c}$$

where `<|user|>` and `<|assistant|>` denote special tokens defined by the chat template to delineate dialogue roles, $q$ represents the user query, and the tokens `<think>` and `</think>` explicitly enclose the model's internal reasoning process $r$. The final conclusion $c$ distills the reasoning into a concise and coherent response. The underlined segment corresponds to the model's generated output, while all preceding tokens constitute the prompt input.

Despite its effectiveness across a wide range of reasoning tasks, this paradigm exhibits a fundamental limitation: as task difficulty increases, the length of the reasoning trace $r$ grows substantially. This not only risks exceeding the model's context window but also incurs prohibitive computational and memory costs due to the quadratic complexity of self-attention with respect to sequence length. To overcome these limitations, InftyThink reformulates monolithic long-chain reasoning into an iterative reasoning process, interleaving generation with intermediate summarization to enable scalable and efficient deep reasoning.

### C.2. InftyThink Paradigm

In the InftyThink paradigm, the reasoning process is decomposed into a sequence of interconnected reasoning segments. Each segment operates under a bounded token budget to ensure computational efficiency, while a summary-based mechanism preserves the global coherence of the reasoning trajectory across iterations.

**The first reasoning iteration** ($i = 1$) is formalized as:

$$<|user|>q<|assistant|>\underline{\texttt{<think>}r_1\texttt{</think><summary>}s_1\texttt{</summary>}}$$

where $r_1$ denotes the initial reasoning segment with a constrained length, and $s_1$ is a compact summary distilled from $r_1$. Encapsulated by the special tokens `<summary>` and `</summary>`, this summary serves as a compressed representation of the current reasoning state, retaining essential information while discarding redundant or low-utility details.

**For subsequent iterations** ($i > 1$), the model conditions its reasoning on the summary generated in the previous iteration:

$$<|user|>q<|assistant|>\texttt{<history>}s_{i-1}\texttt{</history>}\ \underline{\texttt{<think>}r_i\texttt{</think><summary>}s_i\texttt{</summary>}}$$

where the `<history>` and `</history>` tokens delimit the previous summary $s_{i-1}$, which provides critical contextual information for generating the current reasoning segment $r\_i$. This iterative process enables the model to progressively extend its reasoning while maintaining a bounded per-iteration token length, with global information propagated through the summary channel.

**In the final iteration** ($i = n$), the model produces a conclusion instead of generating another summary:

$$<|user|>q<|assistant|>\texttt{<history>}s_{n-1}\texttt{</history>}\underline{\texttt{<think>}r_n\texttt{</think>}c}$$

Here, blue indicates reasoning segments, pink denotes intermediate summaries, and green represents the final conclusion. This formulation naturally accommodates edge cases: for problems that can be solved within a single reasoning step, the model omits summary generation and reduces to the standard vanilla reasoning paradigm.

During inference, the model repeatedly generates reasoning segments and their corresponding summaries, using each summary as the contextual input for the next iteration. The process terminates when the model outputs a conclusion rather than a summary, indicating that the reasoning task has been completed. To prevent unbounded iteration, we introduce a hyperparameter $\varphi$ that specifies the maximum number of allowed reasoning iterations; the process is forcibly terminated once this limit is reached.

## D. Reasoning Efficiency Analysis of InftyThink Paradigm

The motivation behind InftyThink arises from the fact that modern reasoning models often generate extremely long chains-of-thoughts, typically exceeding 10K tokens or more. However, current decoder-based LLMs rely on self-attention (Vaswani et al., 2017), whose computational and memory complexity grows quadratically ($O(n^2)$) with the sequence length. As a result, generating each additional token during late-stage reasoning incurs a rapidly increasing computational cost.

To mitigate this $O(n^2)$ complexity, InftyThink decomposes a long reasoning chain into multiple inference rounds, connected via summaries. This design reduces the computational burden during inference. The relationship can be expressed as:

$$R = R_1 + R_2 + \ldots + R_n;$$
$$R^2 \geq R_1^2 + R_2^2 + \ldots + R_n^2. \tag{9}$$

The core mechanism of InftyThink is an iterative reasoning process in which the model alternates between generating a partial reasoning segment, compressing its current reasoning state into a concise summary, and leveraging this summary to guide subsequent iterations. As illustrated in Figure 5, conventional reasoning paradigms (left, blue) inevitably terminate once the accumulated context reaches the model's maximum length, often before the reasoning process is complete. In contrast, InftyThink (right, pink) introduces periodic summarization that induces a characteristic sawtooth pattern in context usage, effectively bounding the memory footprint while allowing the reasoning process to continue indefinitely.

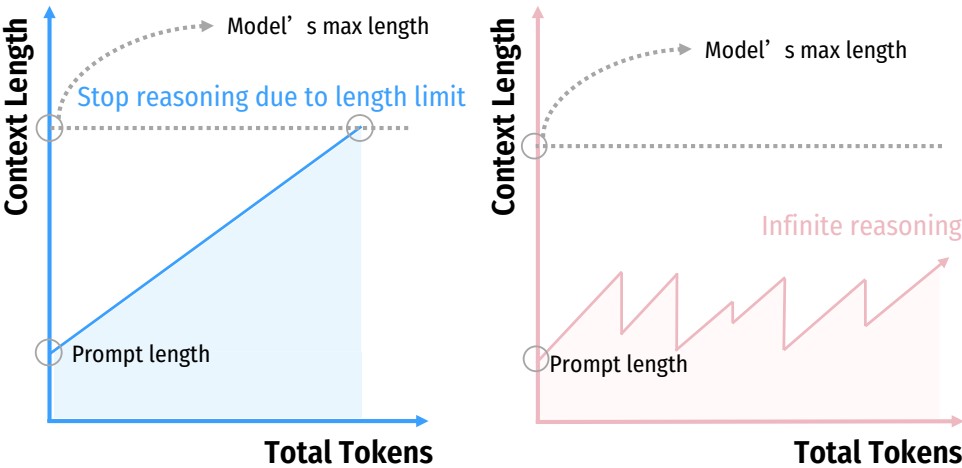

*Figure 5.* Computational complexity comparison between vanilla long-context reasoning (blue, left) and InftyThink (pink, right). The sawtooth pattern of InftyThink demonstrates how periodic summarization creates a bounded memory footprint, substantially reducing computational costs (smaller area under curve) while enabling deeper reasoning. We adopt the figure design style from Yan et al. (2025).

This design substantially reduces computational overhead, as reflected by the smaller area under the curve, and fundamentally removes the hard ceiling on reasoning depth imposed by fixed context-length constraints. Beyond efficiency gains, InftyThink offers a critical conceptual advantage: it enables reasoning of arbitrary depth without requiring any architectural modifications to the underlying model. By continuously summarizing and reusing intermediate reasoning in compact, structured segments, the model can systematically explore complex problem spaces that would otherwise exceed its context capacity. InftyThink converts a single long generation into multiple short generations, greatly reducing the computational overhead induced by the decoder's $O(n^2)$ complexity. Consequently, the model maintains lower latency even when generating more total tokens. (See Figure 5, the area under the curve.)

# E. Full Recipe of Cold-start Stage

In this paper, we introduce a critical *cold-start* stage in InftyThink+ (Section 3.2), whose goal is to effectively migrate the model's reasoning behavior to the InftyThink reasoning paradigm. To achieve this paradigm shift, we first convert supervised fine-tuning (SFT) data originally constructed under the vanilla reasoning paradigm into the InftyThink-style format (described in Appendix E.1). We then perform SFT on the transformed data, enabling the model to acquire and internalize InftyThink-style reasoning behaviors (described in Appendix E.2).

## E.1. Paradigm Transformation

In this paper, we follow the approach of Yan et al. (2025) and decompose the transformation of vanilla data into the InftyThink-style reasoning paradigm into three stages. First, we perform reasoning partition, where a long chain-of-thought (CoT) is segmented into multiple shorter reasoning chains according to a set of predefined rules. Second, we generate summaries by leveraging a general-purpose LLM to summarize the key reasoning steps. Third, we reconstruct the training data by integrating the generated summaries with the partitioned reasoning segments, thereby forming a new collection of InftyThink-style training samples. The overall workflow is illustrated in Figure 2. In the following, we describe the detailed methodology of each stage in turn.

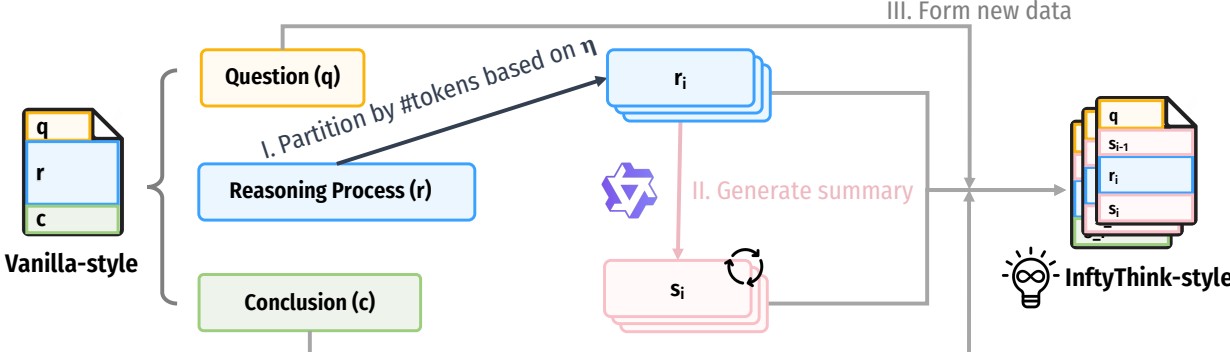

*Figure 6.* Systematic pipeline for reconstructing vanilla-style long-context reasoning data into the InftyThink-style format. **I.** Original reasoning processes are partitioned into optimally sized fragments based on parameter ($\eta$), preserving semantic coherence. **II.** Qwen3-4B-Instruct-2507 generates concise yet comprehensive summaries for each reasoning fragment. **III.** The original fragments and their generated summaries are systematically recombined to create InftyThink-style training instances that teach the model to reason iteratively. We adopt the figure design style from Yan et al. (2025).

**Step I: Reasoning Process Partition**  For each data instance, we partition the original reasoning process ($r$) into a sequence of shorter segments, guided by a hyperparameter $\eta$ that specifies the maximum token length allowed per segment. Instead of performing naive or arbitrary truncation, we adopt a semantically-aware segmentation strategy. Specifically, we first decompose the reasoning process into fine-grained semantic units by detecting natural boundaries such as sentence or paragraph breaks. These semantic units are then tokenized (in this paper, we used DeepSeek-R1-Distill-Qwen-1.5B's tokenizer) and incrementally merged into contiguous segments, prioritizing semantic coherence while ensuring that the token length of each segment does not exceed the threshold $\eta$. As a result, the original reasoning process is transformed into an ordered sequence of reasoning segments $\{r_1, r_2, \ldots, r_n\}$, which can be formally expressed as:

$$\text{Partition}(r, \eta) \rightarrow \{r_1, r_2, \ldots, r_n\}. \tag{10}$$

We implement the reasoning process partition as follows. First, we extract the complete reasoning content using the regular expression `^<think>\n(.+)\n</think>(.+)$`. Samples that cannot be matched by this pattern are discarded, as they do not conform to the standard format. Next, we segment the extracted reasoning content using the widely adopted delimiter `\n\n` in the CoT outputs of DeepSeek-R1, which preserves semantic completeness at the paragraph level. Each resulting segment is then tokenized using the tokenizer of DeepSeek-R1-distill-Qwen-1.5B, and its token length is recorded. We subsequently apply a greedy aggregation strategy: segments are concatenated in order as long as the total length does not exceed the predefined hyperparameter $\eta$. Any aggregated segment whose length exceeds $\eta$ is filtered out. Through this procedure, we obtain a set of partitioned reasoning processes with bounded length. Empirically, we observe that all filtering steps together remove fewer than 1‰ of the original samples.

**Step II. Summary Generation**    For each reasoning segment, we construct a concise summary that distills its key insights and reflects the incremental progress toward the final solution. We adopt a high-capacity foundation model $M$ for summary generation, specifically, Qwen3-4B-Instruct-2507 (Yang et al., 2025). Following Yan et al. (2025), it has been shown that the choice of the model used for summary generation has a negligible impact on the overall performance of InftyThink. Therefore, to enable fast yet accurate summary generation, we employ a relatively small but high-capacity LLM to produce the summaries. All summaries are generated using carefully designed prompts.

Formally, **the summary at iteration** 1 is defined as:

$$S_1 = \text{summarize}(M, r_1), \tag{11}$$

with the generation prompt as following.

---

**Summary Generation Prompt for Iteration #1 (PROMPT_1)**

Please summarize the reasoning and conclusions you reached in your previous truncated response. Here are the specific requirements:

1.  You need to summarize the key steps and corresponding important conclusions you took in all the reasoning processes in chronological order;
2. You need to summarize the steps and conclusions that helped to ultimately solve the problem;
3. You do not need to provide the final answer or any additional notes;
4. Please summarize as concisely as possible, but do not omit any important steps or conclusions;
5. Please note that your reasoning may not be complete;
6. Please do not provide any reasoning or conclusions that were not presented.
7. Please use '*' to list all summaries.

---

And **the summary at iteration** $i(1 < i < n)$ is defined as:

$$S_i = \text{summarize}(M, r_i, s_{i-1}), \tag{12}$$

with the generation prompt as following.

---

**Summary Generation Prompt for Iteration #i (PROMPT_2)**

Please update your reasoning history based on the reasoning and conclusions reached in the previous truncated response. The specific requirements are as follows:

1.  You need to summarize the key steps and corresponding important conclusions you took in all reasoning processes (including your entire reasoning history) in chronological order;
2. You need to summarize the steps and conclusions that helped to ultimately solve the problem;
3. You do not need to provide the final answer or any additional notes;
4. Please summarize as concisely as possible, but do not omit any important steps or conclusions;
5. Please note that your reasoning may not be complete;
6. Please do not provide any reasoning or conclusions that were not presented.
7. Please use '*' to list all summaries.

---

For intermediate reasoning segments $(1 < i < n)$, our approach introduces a subtle but important deviation from the original formulation in Yan et al. (2025). Specifically, Yan et al. (2025) generates the summary $s_i$ using the entire set of reasoning segments $\{r_1, \ldots, r_i\}$, thereby producing a *global* summary at each iteration. We argue that this design may lead to a potential misalignment with the model's actual inference-time behavior. In practice, when generating $s_i$ during inference, the model does not have access to the full preceding reasoning trajectory $\{r_1, \ldots, r_{i-1}\}$; instead, it only observes the current reasoning segment $r_i$ together with the previous summary $s_{i-1}$. Since $s_{i-1}$ is a compressed representation, it may omit information that would otherwise be necessary to faithfully reconstruct $s_i$. Training on data constructed with richer context than is available at inference time can therefore induce hallucination, where the model learns to introduce intermediate details that are not grounded in the provided summary.

To better align training with the model's inference-time reasoning pattern, we slightly modify the context used for summary generation. Concretely, we generate $s_i$ conditioned only on $r_i$ and $s_{i-1}$. This design ensures that the summarization process

operates under the same informational constraints as those encountered during actual reasoning, thereby reducing the risk of hallucination and enabling the model to produce more accurate and faithful summaries.

For **the final reasoning iteration** $n$, we do not generate a summary, as the model is expected to produce the final conclusion in this round rather than an intermediate summary.

We adopt a multi-turn conversational protocol for summary generation, rather than a single-pass generation. This design choice is motivated by the desire to more effectively leverage the model's post-alignment capabilities, thereby producing higher-quality summaries. Specifically, the multi-turn interaction allows the model to better contextualize the reasoning content, follow structured instructions, and refine its abstraction behavior in a manner consistent with its alignment training.

Concretely, for the first iteration ($i = 1$), the messages provided to the summarization model are defined as follows:

```
messages = [
    {"role": "user", "content": question},
    {"role": "assistant", "content": reasoning_process_1},
    {"role": "user", "content": PROMPT_1}
]
```

For an intermediate iteration $i$ ($1 < i < n$), the messages fed into the model are defined as:

```
messages = [
    {"role": "user", "content": question},
    {"role": "assistant", "content": last_summary},
    {"role": "user", "content": "Please continue your reasoning based on your past
        reasoning history."},
    {"role": "assistant", "content": reasoning_process_i},
    {"role": "user", "content": PROMPT_2}
]
```

Building upon the approach of Yan et al. (2025), we introduce a hyperparameter $\gamma$ to explicitly control the compression ratio of summaries. Specifically, during summary generation, we enforce a length constraint by verifying whether the number of tokens in the generated summary is below the predefined threshold $\gamma$. If the constraint is violated, we resample the summary, with up to 10 retry attempts. If the generated summary still exceeds the threshold after all retries, the corresponding sample is discarded. Empirically, we observe that the discard rate induced by this constraint is below 1‰, indicating that the proposed length control has a negligible impact on data efficiency.

During summary generation, we adopt SGLang (Zheng et al., 2024) as the inference engine (version 0.5.6), running on NVIDIA GPUs. All engine configurations are kept at their default settings. We leverage the asynchronous inference interface provided by SGLang. For sampling, the temperature is set to 0.5 and the top_p is set to 0.95, while all other sampling parameters remain at their default values.

**Step III. Training Instance Construction**  Based on the segmented reasoning traces and their corresponding summaries, we construct a set of training instances that explicitly supervise the model to perform iterative reasoning with intermediate summarization. Each instance is organized to align with the InftyThink reasoning paradigm and is defined as follows:

$$(q, r, c) \xrightarrow{\eta, \gamma} \begin{cases} (q, r_1, s_1) & \text{for } i = 1, \\ (q, s_{i-1}, r_i, s_i) & \text{for } 1 < i < n, \\ (q, s_{n-1}, r_n, c) & \text{for } i = n. \end{cases} \tag{13}$$

At the initial iteration ($i = 1$), the model is trained to generate the first reasoning segment along with its corresponding summary. For intermediate iterations ($1 < i < n$), the model learns to condition on the previously generated summary to extend the reasoning process and produce an updated summary. In the final iteration ($i = n$), the model is guided to leverage the last summary to complete the reasoning and output the final conclusion.

### E.2. Supervised Fine-tuning

**Cold Start via Supervised Fine-Tuning.**  The cold-start stage in this work is implemented via supervised fine-tuning (SFT), where the model is trained by directly supervising its output token probabilities. Specifically, we adopt the standard cross-entropy loss to supervise the likelihood of each token in the model-generated response.

**Vanilla Paradigm.** Under the vanilla paradigm, we follow the standard instruction fine-tuning procedure. The query and response are concatenated according to the tokenizer-specific `chat_template`, with special tokens inserted to indicate conversational roles and boundaries. During training, the loss is computed exclusively over the response tokens, while the query tokens and all special tokens introduced by the chat template are masked out from the loss computation. This training process can be formalized as:

```
input_ids = tokenizer.apply_chat_template(
    [{"role": "user", "content": question}],
    add_generation_prompt=True
)

response_txt = f"<think>\n{reasoning_process}\n</think>{conclusion}"
response_ids = tokenizer(response_txt).input_ids
```

**InftyThink Paradigm.** Under the InftyThink paradigm, we introduce a slight but crucial modification to the above supervision strategy. For the first reasoning iteration ($i = 1$), no history is involved in the input context. As a result, the input structure is identical to that of the vanilla paradigm, and we apply the same supervision and loss masking strategy. Formally, this process can be expressed as:

```
input_ids = tokenizer.apply_chat_template(
    [{"role": "user", "content": question}],
    add_generation_prompt=True
)

response_txt = f"<think>\n{reasoning_process_1}\n</think><summary>{summary_1}</summary>"
response_ids = tokenizer(response_txt).input_ids
```

For subsequent reasoning iterations ($i > 1$), the input context additionally includes a history segment summarizing previous reasoning steps. Since this history is provided as contextual information rather than an output to be generated, we explicitly prevent the model from learning to reproduce it. Concretely, after applying the chat template to the query, we append the history tokens to the resulting input sequence, forming the complete model input. The response remains unchanged. During training, we compute the loss only over the response tokens, while masking out both the query and the history tokens from loss computation. This procedure can be expressed as:

```
input_ids = tokenizer.apply_chat_template(
    [{"role": "user", "content": question}],
    add_generation_prompt=True
)
history_txt = f"<history>\n{summary_i-1}\n</history>"
history_ids = tokenizer(history_txt).input_ids
input_ids = input_ids + history_ids

response_txt = f"<think>\n{reasoning_process_i}\n</think><summary>{summary_i}</summary>"
response_ids = tokenizer(response_txt).input_ids
```

For the final iteration ($i = n$), the model no longer performs summarization. Instead, it directly generates the final conclusion based on the accumulated reasoning context. Accordingly, the supervision strategy remains consistent with previous iterations: the model conditions on the query and the history, while the loss is computed solely over the conclusion tokens. Formally, this process can be expressed as:

```
input_ids = tokenizer.apply_chat_template(
    [{"role": "user", "content": question}],
    add_generation_prompt=True
)
history_txt = f"<history>\n{summary_n_1}\n</history>"
history_ids = tokenizer(history_txt).input_ids
input_ids = input_ids + history_ids

response_txt = f"<think>\n{reasoning_process_n}\n</think>{conclusion}"
response_ids = tokenizer(response_txt).input_ids
```

## F. Full Recipe of Reinforcement Learning Stage

The core idea of InftyThink+ is to incorporate reinforcement learning (RL) into the optimization of the InftyThink reasoning paradigm. In this section, we provide a detailed description of the RL implementation details. Algorithm 1 illustrates the RL workflow for a single query $q$.

---

**Algorithm 1** InftyThink+ Reinforcement Learning Step

---

1: **Inputs:** query $q$; LLM policy $\pi_\theta$; LLM tokenizer $t$; Max InftyThink iteration rounds $\varphi$; InftyThink format extractor $F$; group size $G$; task reward function $\mathcal{R}_{\text{task}}$; efficiency reward function $\mathcal{R}_{\text{eff}}$; learning rate $\eta_{lr}$.
2: $\mathcal{O}, R \leftarrow \{\}, \{\}$                       ▷ InftyThink rollout trajectories and rewards
3:
4: // Generate InftyThink rollout trajectories
5: **for** $i \leftarrow 1$ to $G$ **do**
6:    $p \leftarrow t.\text{apply\_chat\_template}(q)$              ▷ Initial query with chat template
7:    **for** $j \leftarrow 1$ to $\varphi$ **do**
8:       **if** $j = 1$ **then**
9:          $x \leftarrow p$                     ▷ Prompt without a summary
10:       **else**
11:          $x \leftarrow p \oplus s_{j-1}$                ▷ Prompt with a summary
12:       **end if**
13:       $o \leftarrow \pi_\theta(x)$                         ▷ Generate
14:       $s_j \leftarrow F(o)$             ▷ Extract summary from the generation
15:       $\mathcal{O}[(i,j)] \leftarrow o$
16:       **if** $s_i$ is [NONE] **then**
17:          break                  ▷ No summary found, break the loop
18:       **end if**
19:    **end for**
20: **end for**
21:
22: // Assign rewards
23: **for** $i \leftarrow 1$ to $G$ **do**
24:    $n = |\{\mathcal{O}[(i,*)]\}|$                  ▷ Iteration number of trajectory
25:    $r_{\text{task}} = \mathcal{R}_{\text{task}}(\mathcal{O}[(i,n)]), r_{\text{eff}} = \mathcal{R}_{\text{eff}}(\mathcal{O}[(i,n)])$       ▷ Reward calculation
26:    **if** use_efficiency_reward **then**
27:       $r \leftarrow r_{\text{task}} \cdot r_{\text{eff}}$
28:    **else**
29:       $r \leftarrow r_{\text{task}}$
30:    **end if**
31:    **for** $j \leftarrow 1$ to $n$ **do**
32:       $R[(i,j)] \leftarrow r$                    ▷ Reward broadcast
33:    **end for**
34: **end for**
35:
36: // Estimate advantages
37: $\{\hat{A}[(i,j)]\}_{i=1,j=1}^{G,n_i} \leftarrow \text{ComputeAdvantage}\left(\{R[(i,j)]\}_{i=1,j=1}^{G,n_i}, R\right)$   ▷ Compute the advantages in this group
38:
39: // Updating policy model
40: $J \leftarrow \frac{1}{\sum_{i=1}^{G}\sum_{j=1}^{n_i}|\mathcal{O}[(i,j)]|} \sum_{i=1}^{G}\sum_{j=1}^{n_i} \mathcal{U}(\mathcal{O}[(i,j)];\theta)$   ▷ Compute the policy gradient loss according to Equation 6
41: $\theta \leftarrow \theta + \eta_{lr} \nabla_\theta J$

---

We detail the algorithmic components of InftyThink+ RL from four aspects: rollout (Appendix F.1), reward assignment (Appendix F.2), policy gradient optimization (Appendix F.3), and training stability (Appendix F.4).

### F.1. Rollout

**Trajectory-Level Rollout.**   For RL training under the InftyThink+ framework, we adopt a *trajectory-level rollout* strategy. Specifically, for each query, a single rollout corresponds to one complete InftyThink-style reasoning trajectory, spanning all iterative reasoning rounds until termination. Unlike tree-based or branching rollouts commonly used in search-based RL methods, we restrict training to a single, linear rollout per query, which substantially simplifies both rollout generation and policy optimization.

Formally, for the $i$-th rollout associated with a query $q$, the resulting trajectory can be represented as

$$\text{Rollout}(q, i) \rightarrow \mathcal{O}_i = \{o_i^1, o_i^2, \ldots, o_i^{n_i}\}, \tag{14}$$

where $o_i^j$ denotes the model output at the $j$-th reasoning iteration, and $n_i$ is the total number of iterations in trajectory $\mathcal{O}_i$.

**InftyThink-Style Iterative Reasoning.**   Trajectory-level rollouts follow the *InftyThink-style* reasoning paradigm, in which multiple rounds of reasoning are connected via model-generated summaries that serve as compact intermediate state representations.

For the first iteration ($j = 1$), the model performs inference directly conditioned on the original query after applying the chat template, without any intermediate summaries:

$$p = \texttt{apply\_chat\_template}(q), \tag{15}$$

$$o^1 = \pi_\theta(p). \tag{16}$$

For subsequent iterations ($j > 1$), we apply an *InftyThink-style* structured extraction function $F$ to the output of the previous iteration. This function parses the model output and extracts a summary that abstracts the essential reasoning state required for continuation:

$$s^{j-1} = F(o^{j-1}). \tag{17}$$

If no valid summary can be extracted, or if the iteration index $j$ reaches a predefined maximum reasoning depth $\varphi$, the trajectory is terminated. Otherwise, the extracted summary is concatenated with the original prompt and used as the input context for the next iteration:

$$o^j = \pi_\theta(p \oplus s^{j-1}). \tag{18}$$

**RL-Compatible Context Handling.**   To ensure compatibility with existing RL training frameworks, we do not treat the generated summaries as part of the `input_ids`. Since summary lengths are inherently variable and not explicitly controllable, directly including them as inputs would cause prompt-length validation failures in standard RL implementations.

Instead, for iterations $j > 1$, we prepend the tokenized summary history to the corresponding model output and treat the concatenation as a single sequence:

$$\{o^j\}_{j>1} = \{s^{j-1} \oplus o^j\}_{j>1}. \tag{19}$$

This design allows the RL framework to operate on fixed input prompts while still preserving the full iterative reasoning context within the trajectory.

**Loss Masking for History Tokens.**   Crucially, although summary tokens are included in the sequence representation, we do not intend to optimize the policy with respect to these history tokens. To this end, we construct a loss mask for each iteration that blocks gradient propagation through the summary portion:

$$\mathcal{M}^j = \texttt{concat}([0] \times |s^{j-1}|, \ [1] \times |o^j|). \tag{20}$$

During policy optimization, this mask ensures that only newly generated tokens contribute to the loss, preventing unintended updates to the summarized history while maintaining end-to-end compatibility with standard policy gradient training.

## F.2. Reward Assignment

**Trajectory-Level Reward Modeling.** For reward modeling, we compute rewards at the *trajectory level* and broadcast the resulting scalar reward to all outputs along the trajectory. This design enables trajectory-wise credit assignment while avoiding the need for fine-grained, step-level reward annotation, which is often noisy and difficult to define for long-horizon reasoning. Formally, for the $i$-th trajectory, all outputs $o_i^j \in \mathcal{O}_i$ share the same reward:

$$r_i^j \equiv r_i. \tag{21}$$

**Reward Computation.** The reward $r_i$ is computed solely based on the final outcome of the trajectory, reflecting the overall quality of the completed reasoning process. Concretely, we separately compute a *task reward*, which measures solution correctness or task completion quality, and an optional *efficiency reward*, which encourages concise and efficient reasoning.

When the efficiency reward is enabled, the final reward is defined as the product of these two components:

$$r_i = \begin{cases} \mathcal{R}_{\text{task}}(o_i^{-1}) \cdot \mathcal{R}_{\text{eff}}(o_i^{-1}), & \text{if } \texttt{use\_efficiency\_reward}, \\ \mathcal{R}_{\text{task}}(o_i^{-1}), & \text{otherwise}, \end{cases} \tag{22}$$

where $o_i^{-1}$ denotes the penultimate output of trajectory $\mathcal{O}_i$, i.e., the final reasoning output before termination.

This multiplicative formulation ensures that efficiency is rewarded only when the model produces a correct or high-quality solution, thereby preventing degenerate behaviors where the model overly optimizes efficiency at the expense of task performance.

## F.3. Policy Gradient Optimization

**Policy Gradient Optimization with GRPO.** For policy optimization, we adopt *Group Relative Policy Optimization* (GRPO) (Shao et al., 2024). Given a query $q$, we sample $G$ trajectories, each consisting of multiple reasoning outputs. For each output $o_i^j$, we associate a scalar reward $r_i^j$, which is broadcast from the final outcome of its corresponding trajectory. GRPO performs group-wise normalization over these output-level rewards to construct relative advantages, enabling stable policy optimization without an explicit value function.

Formally, let $\{r_i^j\}_{i=1,j=1}^{G,n_i}$ denote the rewards of all outputs sampled for query $q$. We compute the within-group mean and standard deviation as

$$\begin{aligned} \mu &= \text{mean}(\{r_i^j\}), \\ \sigma &= \text{std}(\{r_i^j\}), \end{aligned} \tag{23}$$

and define the normalized advantage for each output as

$$\hat{A}_i^j = \frac{r_i^j - \mu}{\sigma}. \tag{24}$$

**Token-Level Loss Aggregation.** Following prior work on long-horizon RL for language models (Yu et al., 2025b), we employ a *token-level averaging* scheme to aggregate the policy gradient loss. Specifically, the overall objective is given by

$$\mathcal{J}(\theta) = \mathbb{E}_{\{\mathcal{O}_i\}_{i=1}^G \sim \pi_{\theta_{\text{old}}}(\cdot|q)} \left[ \underbrace{\frac{1}{\sum_{i=1}^G \sum_{j=1}^{n_i} |o_i^j|}}_{\text{token-level mean}} \sum_{i=1}^G \underbrace{\sum_{j=1}^{n_i} \overbrace{\mathcal{U}(o_i^j, \mathcal{M}_i^j; \theta)}^{\text{output loss}}}_{\text{trajectory aggregation}} \right], \tag{25}$$

where $\mathcal{U}(o, \mathcal{M}; \theta)$ denotes the loss contribution of a single output $o$ with its corresponding loss mask $\mathcal{M}$.

**Output-Level Objective.** The output-level objective adopts a clipped policy gradient form:

$$\mathcal{U}(o, \mathcal{M}; \theta) = \sum_{t=1}^{|o|} \min\left( r_\theta(o_t)\, \hat{A}_t,\ \text{clip}\big(r_\theta(o_t), 1 - \epsilon_{\text{low}}, 1 + \epsilon_{\text{high}}\big)\, \hat{A}_t \right) \cdot \mathcal{M}_t, \tag{26}$$

where $r_\theta(o_t)$ is the importance sampling ratio at token $o_t$:

$$r_\theta(o_t) = \frac{\pi_\theta(o_t \mid \text{ctx}_t)}{\pi_{\theta_{\text{old}}}(o_t \mid \text{ctx}_t)}, \tag{27}$$

with $\text{ctx}_t$ denoting the model context at generation step $t$. We use asymmetric clipping thresholds $\epsilon_{\text{low}}$ and $\epsilon_{\text{high}}$, following prior GRPO-based implementations (Shao et al., 2024; Yu et al., 2025b).

**Output-Level Advantage Broadcasting.** The token-level advantage $\hat{A}_t$ is inherited from the output-level normalized advantage:

$$\hat{A}_t = \hat{A}_i^j, \quad \forall t \in o_i^j. \tag{28}$$

Thus, all tokens belonging to the same output share the same advantage value. This design is consistent with our output-level reward assignment while enabling fine-grained token-level policy optimization.

**Loss Masking.** The loss mask $\mathcal{M}_t \in \{0, 1\}$ controls whether a token contributes to the policy gradient update. In particular, $\mathcal{M}_t = 0$ masks out non-optimizable tokens such as history tokens or externally provided context, ensuring that gradients are applied only to newly generated tokens at each reasoning round.

**Summary.** Overall, this objective combines GRPO-style group-relative normalization at the *output level* with token-level loss aggregation and masking, yielding a stable and efficient policy gradient formulation for long-context and iterative reasoning.

### F.4. Stable Training: IcePop

Although GRPO provides a stable policy optimization objective in principle, we observe that in practice, long-horizon RL training can still suffer from instability when rollout generation and gradient optimization are handled by separate execution engines. Such instability is particularly pronounced for long chain-of-thought (CoT) reasoning, where small probability discrepancies may accumulate across many iterations.

To mitigate this issue, we adopt IcePop (Team et al., 2025c) as a stability-enhanced variant of GRPO. Rather than modifying the overall training pipeline, IcePop introduces a lightweight calibration mechanism that suppresses noisy gradient updates caused by training–inference probability mismatch.

**IcePop-Calibrated GRPO Objective.** Following IcePop, we introduce a token-level masking function $\mathcal{M}(\cdot)$ based on the probability ratio between the training and inference policies. For a token $o_t$ generated under the inference policy, we define

$$k_t = \frac{\pi_{\text{train}}(o_t \mid \text{ctx}_t; \theta_{\text{old}})}{\pi_{\text{infer}}(o_t \mid \text{ctx}_t; \theta_{\text{old}})}, \tag{29}$$

and apply the masking function

$$\mathcal{M}_{\text{IcePop}}(k_t) = \begin{cases} k_t, & k_t \in [\alpha, \beta], \\ 0, & \text{otherwise,} \end{cases} \tag{30}$$

where $\alpha$ and $\beta$ denote the lower and upper bounds of the trusted calibration region.

With this mask, the IcePop-calibrated round-level objective becomes

$$\mathcal{U}_{\text{IcePop}}(o, \mathcal{M}; \theta) = \sum_{t=1}^{|o|} \mathcal{M}_{\text{IcePop}}(k_t) \cdot \min\left(r_\theta(o_t)\,\hat{A}_t, \ \text{clip}\big(r_\theta(o_t), 1 - \epsilon_{\text{low}}, 1 + \epsilon_{\text{high}}\big)\hat{A}_t\right) \cdot \mathcal{M}_t, \tag{31}$$

where $r_\theta(o_t)$, $\hat{A}_t$, and $\mathcal{M}_t$ follow the same definitions as stated above.

**Discussion.** Intuitively, IcePop restricts policy updates to a region where the training and inference policies remain well aligned, and discards gradient contributions from tokens with excessive probability deviation. In our setting, IcePop is applied as a drop-in replacement for GRPO, without altering the rollout strategy, advantage normalization, or token-level loss aggregation. This simple modification substantially improves training stability in long-context reasoning RL, while preserving the efficiency and simplicity of GRPO.

# G. Experimental Details

In this section, we provide a comprehensive description of the experimental settings used throughout the paper, including the configurations of all hyperparameters during both training (Appendix G.1) and evaluation (Appendix G.2), as well as the hardware environment and the versions of the major third-party libraries employed.

## G.1. Training Details

Our model training comprises two complementary paradigms: supervised fine-tuning (SFT) and reinforcement learning (RL). We present the experimental settings and implementation details for SFT (Appendix G.1.1) and RL (Appendix G.1.2) separately in the following sections.

### G.1.1. SFT EXPERIMENTAL DETAILS

Our SFT experiments are conducted using the ms-swift (Zhao et al., 2025) training framework, with Megatron-Core(Shoeybi et al., 2020) serving as the training backend to enable efficient model parallelism and long-sequence training. To accelerate training and reduce computational waste caused by padding, we apply sample packing to the SFT data. Within each packed sequence, we leverage FlashAttention (Dao, 2023) with variable-length attention (`var_len_attn`) to preserve the independence of attention computation across individual samples. All experiments are performed on 8 NVIDIA GPUs with CUDA 12.8. The hyperparameter configuration for the SFT experiments is summarized in Table 4.

*Table 4.* Key hyperparameters for supervised fine-tuning.

| Hyperparameter | DeepSeek-R1-Distill-Qwen-1.5B | Qwen3-4B-Base |
|---|---|---|
| *Data* | | |
| max_epochs | 3 | 3 |
| max_length | 32,768 | 32,768 |
| packing | true | true |
| *Batch Size* | | |
| micro_batch_size | 1 | 1 |
| global_batch_size | 8 | 8 |
| *Learning Rate* | | |
| lr | 2e-5 | 5e-5 |
| min_lr | 0 | 0 |
| lr_warmup_fraction | 0.03 | 0.03 |
| lr_decay_style | cosine | cosine |
| *Megatron-Core* | | |
| tensor_model_parallel_size | 1 | 2 |
| sequence_parallel | true | true |
| recompute_granularity | full | full |
| recompute_method | uniform | uniform |
| recompute_num_layers | 1 | 1 |
| *Others* | | |
| attention_backend | flash | flash |
| cross_entropy_loss_fusion | true | true |

### G.1.2. RL EXPERIMENTAL DETAILS

Our RL experiments are conducted using verl (Sheng et al., 2025) as the training framework, with SGLang (Zheng et al., 2024) serving as the inference engine and FSDP (Zhao et al., 2023) as the training backend. During rollout, we adopt an *asynchronous* strategy, where different rollout trajectories are processed independently and in parallel. This design is particularly beneficial for InftyThink-style multi-round reasoning, as it significantly improves inference throughput and overall efficiency. Concretely, we leverage the `AgentLoop` module provided in `verl` v0.7.0 to implement the complete InftyThink rollout procedure as well as the corresponding reward assignment pipeline.

For the task reward, we employ the verification function from PRIME Math, which performs exact rule-based validation using symbolic computation libraries such as `SymPy`. To avoid blocking the training process, we introduce a timeout

mechanism in the verification stage: if a model-generated answer cannot be verified as correct within a predefined time limit, it is assigned a reward of zero. The detailed RL hyperparameters are summarized in Table 5.

*Table 5.* Key hyperparameters for reinforcement learning.

| Hyperparameter | DeepSeek-R1-Distill-Qwen-1.5B | | Qwen3-4B-Base | |
|---|---|---|---|---|
| | *vanilla* | *InftyThink+* | *vanilla* | *InftyThink+* |
| *Trainer* | | | | |
| total_training_steps | 1,000 | 1,000 | 500 | 500 |
| *Algorithm* | | | | |
| adv_estimator | grpo | grpo | grpo | grpo |
| rollout_correction.rollout_rs | token | token | token | token |
| rollout_correction.rollout_rs_threshold | 5.0 | 5.0 | 5.0 | 5.0 |
| rollout_correction.rollout_rs_threshold_lower | 0.5 | 0.5 | 0.5 | 0.5 |
| *Data* | | | | |
| train_batch_size | 128 | 128 | 128 | 128 |
| max_prompt_length | 2048 | 2048 | 2048 | 2048 |
| max_response_length | 30720 | 10240 | 30720 | 10240 |
| *Model* | | | | |
| model.use_remove_padding | true | true | true | true |
| model.enable_gradient_checkpointing | true | true | true | true |
| actor.ppo_mini_batch_size | 64 | 64 | 64 | 64 |
| actor.use_dynamic_bsz | true | true | true | true |
| actor.ppo_max_token_len_per_gpu | 32768 | 36864 | 32768 | 73728 |
| actor.clip_ratio_low | 0.20 | 0.20 | 0.20 | 0.20 |
| actor.clip_ratio_high | 0.26 | 0.26 | 0.26 | 0.26 |
| actor.optim.lr | 1e-6 | 1e-6 | 1e-6 | 1e-6 |
| actor.weight_decay | 0.0 | 0.0 | 0.0 | 0.0 |
| rollout.name | sglang | sglang | sglang | sglang |
| rollout.mode | async | async | async | async |
| rollout.tensor_model_parallel_size | 1 | 1 | 1 | 1 |
| rollout.gpu_memory_utilization | 0.9 | 0.9 | 0.9 | 0.9 |
| rollout.n | 8 | 8 | 8 | 8 |
| rollout.temperature | 1.0 | 1.0 | 1.0 | 1.0 |
| rollout.top_p | 1.0 | 1.0 | 1.0 | 1.0 |
| rollout.top_k | -1 | -1 | -1 | -1 |
| ref.log_prob_use_dynamic_bsz | true | true | true | true |
| ref.log_prob_max_token_len_per_gpu | 32768 | 36864 | 32768 | 147456 |

## G.2. Evaluation Details

To ensure a fair and controlled comparison, all models and all trained checkpoints are evaluated under an identical hardware setup and software environment. Specifically, all experiments are conducted on a single machine equipped with 8 NVIDIA GPUs, using our in-house evaluation framework. The framework adopts SGLang (Zheng et al., 2024) as the inference engine, with `tensor_parallel_size` set to 1 and `data_parallel_size` set to 8. All inference is performed in an asynchronous manner, with the concurrency control parameter `Semaphore` set to 1024.

For *vanilla* inference, we set `max_new_tokens` to 32k. For *InftyThink+*, `max_new_tokens` is set to 8k, and the maximum number of reasoning iterations is capped at 10. If the model fails to complete the reasoning process and produce a final conclusion within 10 iterations, the generation is forcibly terminated. For both paradigms, we use a temperature of 0.7 and $top\_p = 0.95$, and sample 32 completions per query.

For metric computation, we employ CompassVerifier-7B (Liu et al., 2025b) to judge the correctness of each completion against the reference answer, and report accuracy averaged over the 32 sampled completions. We compute the token usage (TOK) and end-to-end latency (LAT) using the statistics provided by SGLang. For InftyThink+, token counts and latencies are aggregated across all reasoning iterations to ensure a fair comparison with the vanilla paradigm.

# H. Training Dynamics of RL Experiments

To provide a clearer understanding of the reinforcement learning (RL) training process of InftyThink+, we present the training dynamics observed in the main experiments (shown in Section 4). Specifically, we analyze how key metrics evolve over the course of RL training, including: (1) the trend of per-step training time (Appendix H.1); (2) the evolution of internal model metrics (Appendix H.2); and (3) the dynamics of InftyThink-specific indicators (Appendix H.3). All trend curves correspond to experiments conducted with DeepSeek-R1-Distill-Qwen-1.5B as the base model. The reported statistics are collected using the internal tracking utilities provided by verl.

## H.1. Training-time Metrics

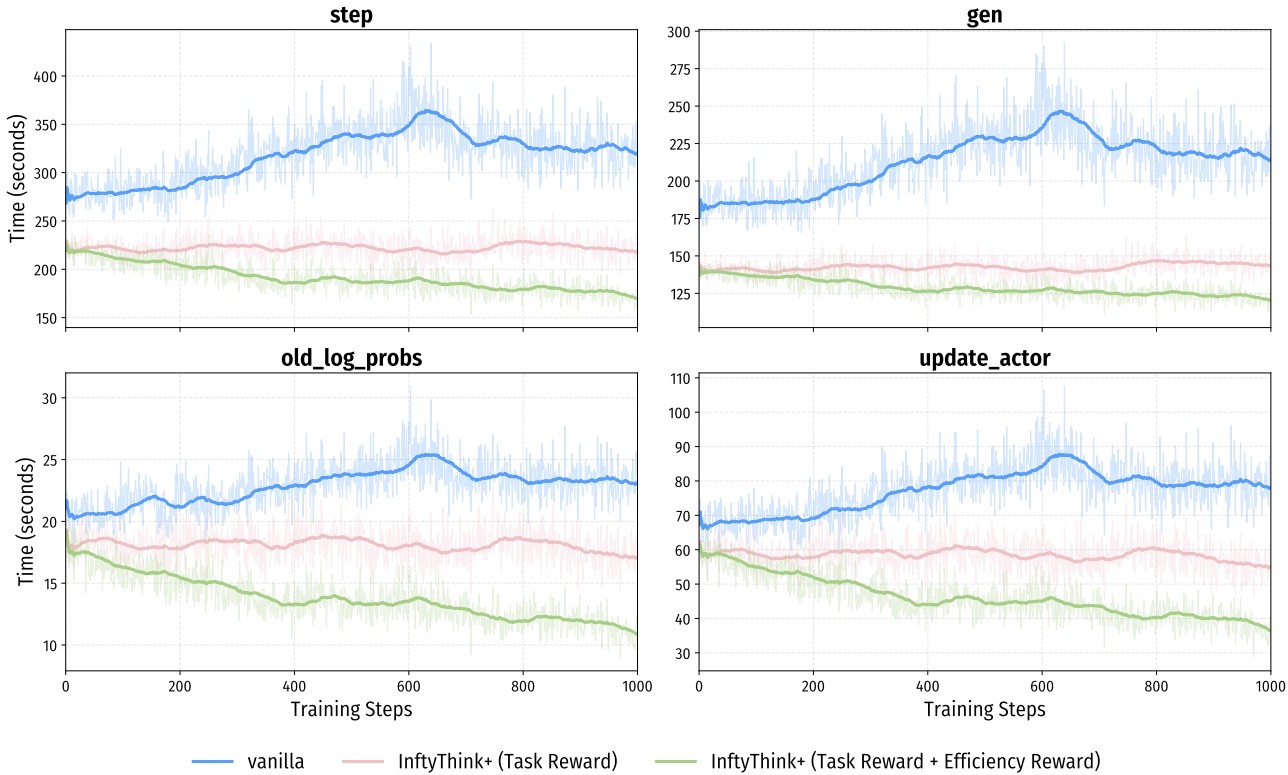

*Figure 7.* Evolution of per-step training time over the course of training.

By decomposing a long reasoning trace into multiple shorter reasoning segments, InftyThink+ mitigates the $O(n^2)$ computational and memory complexity induced by the self-attention mechanism in LLMs. This benefit manifests not only during inference but also translates into substantial acceleration of RL training. Figure 7 illustrates the evolution of per-step training time throughout the training process, from which we draw two key observations.

**InftyThink+ is an efficient RL training method.** First, InftyThink+ achieves markedly higher training efficiency than the vanilla approach. Across all major components, including rollout, log-probability computation, and actor updates, InftyThink+ consistently incurs shorter per-step latency. In aggregate, the average per-step training time of InftyThink+ is approximately 225 seconds, compared to around 325 seconds for the vanilla method, highlighting its more efficient utilization of training resources.

**Efficiency reward push this efficiency into next level.** Second, incorporating the efficiency reward further enhances training efficiency. As shown in the figure, when the efficiency reward is enabled, InftyThink+ exhibits a clear downward trend in training time over the course of learning, with the per-step training time decreasing from roughly 225 seconds to 175 seconds. This result empirically demonstrates the effectiveness of the efficiency reward in promoting faster and more resource-efficient RL training.

## H.2. Model-specific Metrics

In Figure 8, we present the evolution of several actor-related metrics during RL training, including the reward, advantage, policy gradient loss, and entropy, which collectively characterize the internal training dynamics of the model. As shown in the figure, InftyThink+ exhibits a substantially faster reward growth compared to the vanilla baseline. Specifically, over 1,000 training steps, the reward of the vanilla method increases from 0.55 to 0.62, whereas InftyThink+ improves from 0.37 to 0.53. This pronounced acceleration in reward improvement highlights the higher training efficiency of InftyThink+.

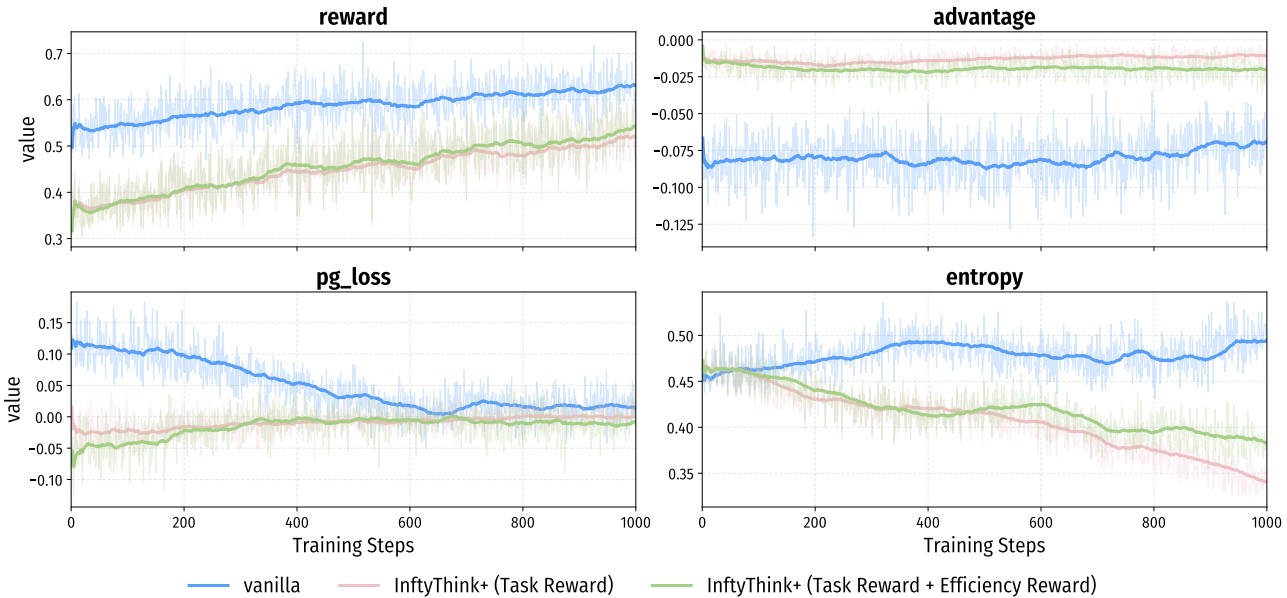

*Figure 8.* Actor-specific metrics over the course of training.

## H.3. InftyThink-specific Metrics

In Figure 9, we present the evolution of InftyThink-related metrics during RL training, including the task reward, efficiency reward, and the number of iteration rounds. Two key observations can be drawn from these results: (1) InftyThink+ consistently achieves higher task rewards compared to the vanilla RL baseline; and (2) incorporating the efficiency reward substantially reduces the number of iteration rounds required by InftyThink, indicating more efficient reasoning trajectories.

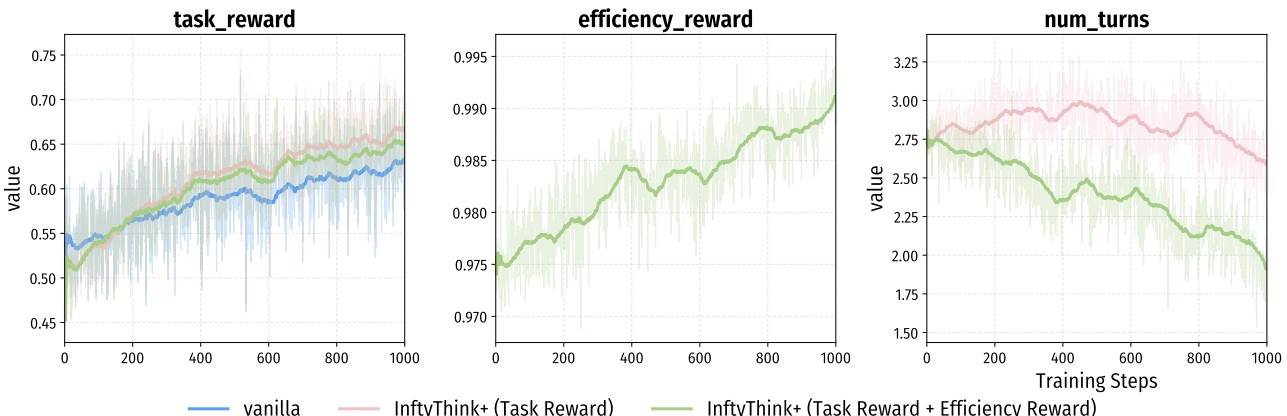

*Figure 9.* InftyThink-specific metrics over the course of training.

# I. Detailed Evaluation Results

Due to space limitations in the main text, we present additional evaluation results in this section. Specifically, this section consists of two parts. First, we report results on a broader range of domains to analyze the effectiveness of InftyThink+ under out-of-distribution (OOD) settings (Appendix I.1). Second, we examine how model performance evolves over the course of RL training, providing a more intuitive and fine-grained view of the model's capability progression (Appendix I.2).

## I.1. Evaluation Across More Domains

We further conduct a comprehensive evaluation of the 1.5B model used in the main experiments across three representative reasoning domains: mathematical reasoning, scientific reasoning, and code reasoning. For each evaluation benchmark, we report not only the final accuracy, but also the average number of generated tokens and the end-to-end inference latency, enabling a more fine-grained analysis of both effectiveness and efficiency. Among these domains, mathematical reasoning corresponds to the in-domain setting, as it aligns with the primary training distribution, whereas scientific reasoning and code reasoning are treated as out-of-domain (OOD) evaluations. This experimental design allows us to assess the generalization capability of the model beyond the training domain, and to examine whether the performance and efficiency gains observed in-domain can consistently transfer to more diverse and challenging reasoning scenarios.

*Table 6.* Evaluation results on in-distribution mathematical reasoning benchmarks. The results are obtained by sampling the model 32 times with a temperature of 0.7. ACC stands for average accuracy (%), TOK stands for average number of generated tokens (K), and LAT stands for average inference time in seconds. ✗ denotes the setting with cold start only, without RL. ✓ T denotes the RL setting where only the task reward is used. ✓ T+E denotes the RL setting where both the task reward and the efficiency reward are used.

| RL | MATH500 | | | AIME24 | | | AIME25 | | | AMC23 | | |
|---|---|---|---|---|---|---|---|---|---|---|---|---|
| | ACC↑ | TOK | LAT↓ | ACC↑ | TOK | LAT↓ | ACC↑ | TOK | LAT↓ | ACC↑ | TOK | LAT↓ |
| *vanilla* | | | | | | | | | | | | |
| ✗ | 86.20 | 5.32 | 48.71 | 26.67 | 17.08 | 158.95 | 24.48 | 48.71 | 134.34 | 70.31 | 48.71 | 63.68 |
| ✓ T | 89.63 | 5.93 | 56.05 | 38.75 | 18.26 | 175.00 | 31.04 | 56.05 | 169.38 | 81.25 | 56.05 | 64.91 |
| Δ | +3.43 | +0.62 | +7.34 | +12.08 | +1.18 | +16.05 | +6.56 | +7.34 | +35.04 | +10.94 | +7.34 | +1.23 |
| *InftyThink+* | | | | | | | | | | | | |
| ✗ | 86.54 | 5.77 | 34.82 | 29.48 | 20.23 | 103.04 | 27.92 | 48.71 | 98.1 | 71.64 | 48.71 | 50.85 |
| ✓ T | 91.56 | 6.10 | 34.26 | 50.94 | 23.36 | 102.85 | 35.83 | 56.05 | 113.78 | 85.86 | 56.05 | 44.57 |
| Δ | +5.02 | +0.33 | -0.56 | +21.46 | +3.13 | -0.19 | +7.91 | +7.34 | +15.68 | +14.22 | +7.34 | -6.28 |
| ✓ T+E | 89.96 | 3.36 | 17.71 | 43.96 | 13.13 | 57.5 | 32.92 | 56.05 | 68.39 | 82.97 | 56.05 | 25.14 |
| Δ | +3.42 | -2.41 | -17.11 | +14.48 | -7.10 | -45.54 | +5.00 | +7.34 | -29.71 | +11.33 | +7.34 | -25.71 |

| RL | MathOdyssey | | | HMMT Feb 25 | | | HMMT Nov 25 | | | Average | | |
|---|---|---|---|---|---|---|---|---|---|---|---|---|
| | ACC↑ | TOK | LAT↓ | ACC↑ | TOK | LAT↓ | ACC↑ | TOK | LAT↓ | ACC↑ | TOK | LAT↓ |
| *vanilla* | | | | | | | | | | | | |
| ✗ | 60.99 | 9.08 | 105.24 | 14.48 | 17.45 | 168.07 | 11.87 | 18.59 | 182.87 | 42.14 | 23.56 | 123.12 |
| ✓ T | 64.83 | 10.54 | 138.21 | 16.67 | 20.82 | 234.16 | 16.35 | 21.48 | 240.14 | 48.36 | 27.02 | 153.98 |
| Δ | +3.84 | +1.46 | +32.97 | +2.19 | +3.37 | +66.09 | +4.48 | +2.89 | +57.27 | +6.22 | +3.46 | +30.86 |
| *InftyThink+* | | | | | | | | | | | | |
| ✗ | 61.56 | 10.54 | 66.17 | 14.17 | 21.07 | 115.23 | 12.19 | 23.28 | 123.91 | 43.36 | 25.47 | 84.59 |
| ✓ T | 68.91 | 12.07 | 77.11 | 18.75 | 31.82 | 150.40 | 21.67 | 43.00 | 195.63 | 53.36 | 32.64 | 102.66 |
| Δ | +7.35 | +1.53 | +10.94 | +4.58 | +10.75 | +35.17 | +9.48 | +19.72 | +71.72 | +10.00 | +7.16 | +18.07 |
| ✓ T+E | 67.25 | 6.43 | 37.94 | 17.08 | 18.20 | 79.73 | 18.44 | 21.89 | 97.35 | 50.37 | 25.02 | 54.82 |
| Δ | +5.69 | -4.11 | -28.23 | +2.91 | -2.87 | -35.5 | +6.25 | -1.39 | -26.56 | +7.01 | -0.46 | -29.77 |

**Mathematical Reasoning.** For mathematical reasoning, we evaluate the model on a diverse set of widely adopted benchmarks, including MATH500, AIME24, AIME25, AMC23, MathOdyssey and HMMT. These benchmarks span a broad spectrum of difficulty levels, from competition-style problems to advanced olympiad-level mathematics, and require precise multi-step reasoning, symbolic manipulation, and rigorous logical deduction. As mathematical reasoning constitutes the primary training domain of our model, these benchmarks are treated as in-domain evaluations and serve as a reference point for assessing both reasoning accuracy and inference efficiency under the target distribution.

As shown in Table 6, on in-distribution mathematical reasoning benchmarks, InftyThink+ consistently amplifies the effectiveness of RL. For example, on AIME24, vanilla RL improves accuracy from 26.67% to 38.75% at the cost of higher latency, whereas InftyThink+ raises accuracy to 50.94% with almost no increase in inference time. This advantage persists in the average in-domain results, where InftyThink+ achieves larger accuracy gains than vanilla RL and, when combined with efficiency reward, even substantially reduces inference latency. These results indicate that InftyThink+ enables RL to translate more effectively into accurate and efficient in-distribution mathematical reasoning.

**Scientific Reasoning.** To assess out-of-domain generalization in scientific reasoning, we conduct evaluations on GPQA_diamond, MMLU_redux, and PHYBench. These benchmarks cover a wide range of scientific disciplines, including physics, chemistry, biology, and interdisciplinary scientific knowledge, and emphasize factual grounding, conceptual understanding, and multi-hop reasoning over technical content. Compared to mathematical benchmarks, scientific reasoning often involves more heterogeneous knowledge sources and less formalized solution structures, providing a complementary perspective on the model's robustness and its ability to transfer reasoning skills beyond the mathematical domain.

*Table 7.* Evaluation results on out-of-distribution scientific reasoning benchmarks. The results are obtained by sampling the model 32 times with a temperature of 0.7. ACC stands for average accuracy (%), TOK stands for average number of generated tokens (K), and LAT stands for average inference time in seconds. ✗ denotes the setting with cold start only, without RL. ✓ T denotes the RL setting where only the task reward is used. ✓ T+E denotes the RL setting where both the task reward and the efficiency reward are used.

| RL | GPQA_diamond | | | MMLU_redux | | | PHYBench | | | Average | | |
|---|---|---|---|---|---|---|---|---|---|---|---|---|
| | ACC↑ | TOK | LAT↓ | ACC↑ | TOK | LAT↓ | ACC↑ | TOK | LAT↓ | ACC↑ | TOK | LAT↓ |
| *vanilla* | | | | | | | | | | | | |
| ✗ | 29.40 | 10.45 | 101.84 | 50.37 | 3.45 | 28.54 | 16.25 | 13.44 | 149.38 | 32.01 | 9.12 | 93.25 |
| ✓ T | 29.81 | 15.48 | 197.33 | 51.10 | 5.85 | 69.27 | 20.11 | 20.46 | 320.06 | 33.67 | 13.93 | 195.55 |
| Δ | +0.41 | +5.03 | +95.49 | +0.73 | +2.4 | +40.73 | +3.86 | +7.02 | +170.68 | +1.67 | +4.82 | +102.3 |
| *InftyThink+* | | | | | | | | | | | | |
| ✗ | 32.31 | 11.77 | 74.31 | 51.34 | 3.12 | 17.61 | 19.14 | 14.60 | 94.55 | 34.26 | 9.83 | 62.16 |
| ✓ T | 37.50 | 24.27 | 149.93 | 54.64 | 4.81 | 30.42 | 30.34 | 30.34 | 219.01 | 40.83 | 19.81 | 133.12 |
| Δ | +5.19 | +12.5 | +75.62 | +3.3 | +1.69 | +12.81 | +11.2 | +15.74 | +124.46 | +6.56 | +9.98 | +70.96 |
| ✓ T+E | 35.46 | 8.69 | 49.87 | 53.94 | 2.29 | 12.55 | 29.62 | 14.63 | 97.57 | 39.67 | 8.54 | 53.33 |
| Δ | +3.15 | -3.08 | -24.44 | +2.6 | -0.83 | -5.06 | +10.48 | +0.02 | +3.02 | +5.41 | -1.29 | -8.83 |

As shown in Table 7, on out-of-distribution scientific reasoning benchmarks, InftyThink+ remains effective but exhibits a more nuanced behavior compared to in-domain mathematical tasks. As shown by the average results, vanilla RL yields only marginal accuracy improvements (+1.67) while substantially increasing inference cost. In contrast, InftyThink+ with task reward (✓ T) achieves a noticeably larger accuracy gain (+6.56 on average), indicating improved generalization of RL-trained reasoning strategies to unseen scientific domains, albeit with increased token usage and latency. When incorporating the efficiency reward (✓ T+E), InftyThink+ preserves most of the accuracy gains while significantly reducing inference cost, suggesting that efficiency-oriented RL is particularly beneficial for controlling overlong reasoning in OOD scientific tasks. Overall, these results highlight that while OOD scientific reasoning is inherently more challenging, InftyThink+ enables RL to generalize more effectively and maintain a better balance between accuracy and efficiency.

**Code Reasoning.** For code reasoning, we evaluate the model on HumanEval and MBPP (Austin et al., 2021; Chen et al., 2021; Liu et al., 2023), which collectively test program synthesis, algorithmic problem solving, and general-purpose coding ability across multiple programming tasks. These benchmarks require the model to generate executable code that satisfies functional correctness constraints, often under implicit edge cases and efficiency considerations. As code reasoning differs substantially from mathematical reasoning in both output structure and evaluation criteria, these benchmarks serve as challenging out-of-domain tests, enabling us to examine whether the reasoning strategies learned during training can effectively generalize to programming-centric scenarios.

Table 8 shows that *InftyThink+* achieves a substantially better accuracy–efficiency trade-off than *vanilla*. Without RL, InftyThink+ already reduces average latency from 64.17 to 26.93s while improving accuracy. With task-only RL (✓ T), vanilla obtains limited accuracy gains but incurs a large cost in TOK/LAT, whereas InftyThink+ yields much larger improvements (average ΔACC +7.61, ΔACC⁺ +6.79). Adding the efficiency reward (✓ T+E) preserves most of the accuracy gains (average ΔACC +6.48) while slightly *reducing* TOK and LAT relative to cold start, indicating that the

efficiency reward effectively regularizes generation length and improves the Pareto frontier.

*Table 8.* Evaluation results on out-of-distribution code reasoning benchmarks. The results are obtained by sampling the model 32 times with a temperature of 0.7. ACC stands for average accuracy (%) on base test case set, $ACC^+$ stands for average accuracy (%) on extended test case set, TOK stands for average number of generated tokens (K), and LAT stands for average inference time in seconds. ✗ denotes the setting with cold start only, without RL. ✓ T denotes the RL setting where only the task reward is used. ✓ T+E denotes the RL setting where both the task reward and the efficiency reward are used.

| RL | HumanEval | | | | MBPP | | | | Average | | | |
|---|---|---|---|---|---|---|---|---|---|---|---|---|
| | ACC↑ | $ACC^+$↑ | TOK | LAT | ACC↑ | $ACC^+$↑ | TOK | LAT | ACC↑ | $ACC^+$↑ | TOK | LAT |
| *vanilla* | | | | | | | | | | | | |
| ✗ | 57.03 | 52.52 | 6.58 | 65.89 | 48.07 | 41.01 | 5.48 | 62.45 | 52.55 | 46.76 | 6.03 | 64.17 |
| ✓ T | 60.44 | 56.27 | 8.17 | 90.10 | 50.09 | 45.43 | 6.69 | 80.36 | 55.26 | 50.85 | 7.43 | 85.23 |
| Δ | +3.41 | +3.75 | +1.58 | +24.21 | +2.01 | +4.42 | +1.21 | +17.91 | +2.71 | +4.09 | +1.39 | +21.06 |
| *InftyThink+* | | | | | | | | | | | | |
| ✗ | 59.03 | 54.34 | 5.02 | 27.50 | 49.28 | 42.05 | 4.73 | 26.36 | 54.16 | 48.20 | 4.88 | 26.93 |
| ✓ T | 67.70 | 62.60 | 8.21 | 42.22 | 55.83 | 47.38 | 9.27 | 49.19 | 61.77 | 54.99 | 8.74 | 45.71 |
| Δ | +8.67 | +8.25 | +3.19 | +14.72 | +6.55 | +5.33 | +4.54 | +22.83 | +7.61 | +6.79 | +3.86 | +18.78 |
| ✓ T+E | 67.42 | 61.91 | 4.66 | 23.9 | 53.85 | 45.92 | 4.64 | 23.62 | 60.63 | 53.91 | 4.65 | 23.76 |
| Δ | +8.38 | +7.56 | -0.36 | -3.60 | +4.57 | +3.87 | -0.09 | -2.74 | +6.48 | +5.72 | -0.22 | -3.17 |

## I.2. Evaluation Dynamics

To better understand how RL shapes the evolution of reasoning ability over time, we analyze the performance trajectories of intermediate model checkpoints throughout the RL training process. Rather than focusing solely on the final converged model, we periodically evaluate model checkpoints on a suite of benchmarks and visualize their performance as training progresses. This allows us to characterize not only the final gains brought by RL, but also the dynamics of learning, e.g., how quickly different capabilities emerge, whether improvements are monotonic, and how stable the training process is across domains. By plotting benchmark performance as curves over RL iterations, we obtain a more fine-grained view of how reasoning accuracy, generalization, and efficiency evolve during training.

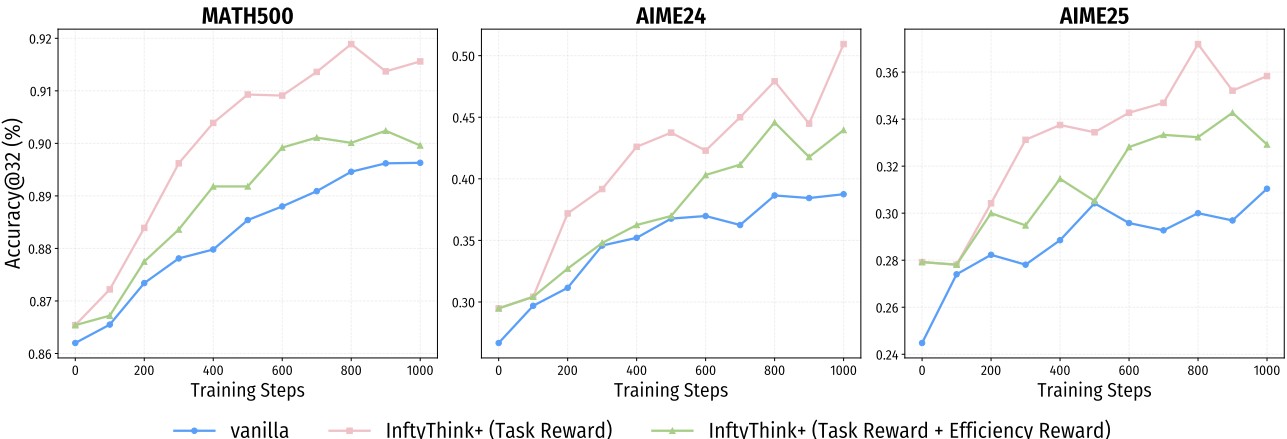

*Figure 10.* Performance evolution of the actor during RL training under different RL settings.

As shown in Figure 10, the actor's performance consistently improves over the course of RL training across all three benchmarks, while exhibiting clear differences among RL strategies. On MATH500, task-reward RL (InftyThink+ T) leads to a rapid and stable accuracy gain, consistently outperforming the vanilla baseline throughout training, indicating effective learning of improved mathematical reasoning strategies. On the more challenging AIME24 and AIME25 benchmarks, this gap becomes more pronounced: vanilla RL shows slower and less stable improvement, whereas InftyThink+ with task reward achieves substantially higher final accuracy. Notably, incorporating the efficiency reward (T+E) slightly slows early-stage accuracy growth but yields smoother training dynamics and competitive late-stage performance, suggesting a

better trade-off between reasoning quality and efficiency. Overall, these results demonstrate that InftyThink+ enables more effective and robust policy optimization during RL, particularly on harder reasoning tasks.

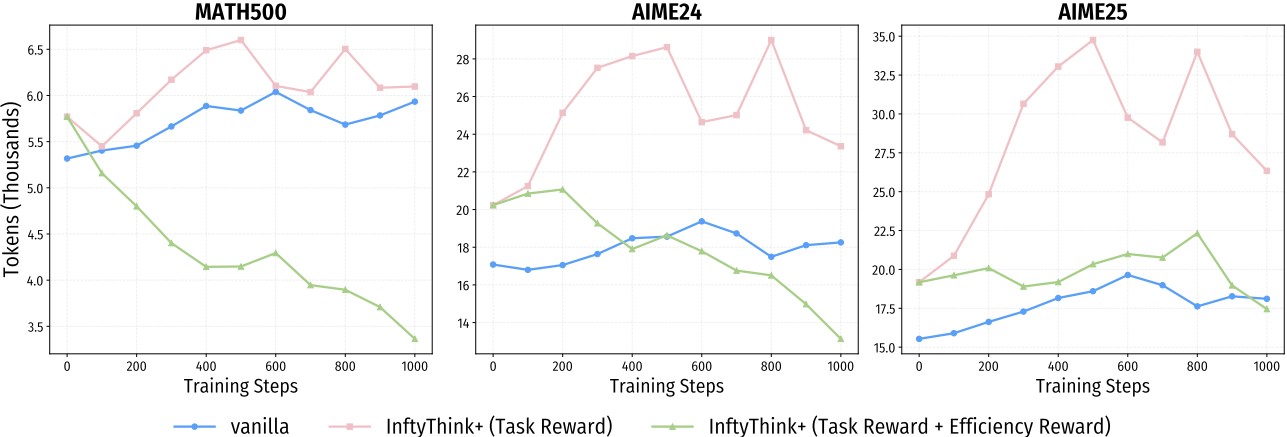

*Figure 11.* Generated tokens (K) of the actor on benchmarks during RL training under different RL settings.

As shown in Figure 11, different RL strategies lead to markedly different generation-length dynamics during training. Under task-reward-only RL (InftyThink+ T), the actor consistently increases its generated tokens across all benchmarks, especially on AIME24 and AIME25, indicating a strong tendency toward longer chains-of-thought as a means to improve accuracy. In contrast, incorporating the efficiency reward (T+E) substantially suppresses token growth and even drives a monotonic reduction in generation length on MATH500, while maintaining competitive or improving accuracy as shown in Figure 10. Notably, the vanilla baseline exhibits relatively stable but suboptimal token usage, failing to adapt its reasoning length effectively. These results suggest that efficiency reward enables the actor to internalize a length-aware reasoning policy, achieving better accuracy–efficiency trade-offs by discouraging unnecessary verbosity during RL optimization.

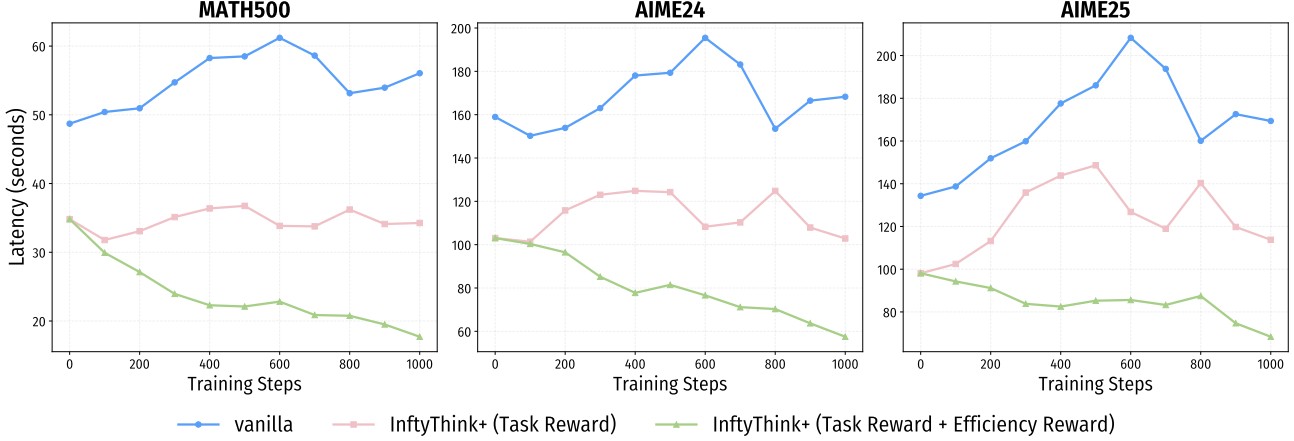

*Figure 12.* Inference latency (s) of the actor on benchmarks during RL training under different RL settings.

As shown in Figure 12, inference latency exhibits trends that closely mirror the generation-length dynamics in Figure 11. Under the vanilla setting, latency steadily increases with training steps across all benchmarks, reflecting the model's inability to control reasoning length during RL optimization. Task-reward-only RL (InftyThink+ T) substantially reduces latency compared to the vanilla baseline, but still shows noticeable fluctuations, especially on the harder AIME24 and AIME25 benchmarks. In contrast, incorporating the efficiency reward (T+E) leads to a consistent and monotonic reduction in latency throughout training, achieving the lowest inference time across all benchmarks. Notably, this latency reduction does not come at the expense of accuracy (Figure 10), indicating that efficiency reward enables the actor to learn more concise yet effective reasoning strategies. Together, these results demonstrate that InftyThink+ with efficiency-aware RL internalizes inference efficiency as a first-class optimization objective, rather than a by-product of shorter generation.

# J. Stability Analysis

In practical RL systems, both training and evaluation inevitably exhibit noticeable fluctuations due to various sources of nondeterminism in the computational environment. Specifically, even when random seeds and algorithmic configurations are fixed, factors such as hardware-dependent execution paths, parallel computation order, and the non-associativity of floating-point arithmetic can introduce subtle numerical perturbations. These perturbations may accumulate over long-horizon RL training and long-context reasoning, leading to divergent optimization trajectories and substantial variance in the performance of intermediate checkpoints. Similarly, during evaluation, reasoning models remain sensitive to decoding order, parallel execution, and low-level kernel implementations, such that repeated evaluations under nominally identical settings may yield inconsistent results. As a consequence, single-run training or evaluation results are often insufficient to faithfully characterize a method's true performance. Motivated by this observation, we conduct a dedicated stability analysis in this section, examining both training stability (Appendix J.1) and evaluation stability (Appendix J.2), in order to assess the robustness and reproducibility of our approach under realistic computational perturbations.

## J.1. Training Stability

To examine the stability of the RL training dynamics, we conduct a controlled study on the main 1.5B-parameter model under the task-reward-only setting. Specifically, we repeat the RL training process three times with identical configurations, including the same hyperparameters, data pipeline, optimization settings, and hardware environment. By holding all experimental conditions constant, any observed divergence across runs can be attributed to intrinsic nondeterminism arising from the underlying computational system rather than differences in configuration. As shown in the following figures, we track the evolution of key training signals, including the training reward, policy gradient loss, and entropy, throughout the RL process, and further evaluate intermediate checkpoints on AIME24, AIME25, and MATH500 to monitor the variance in downstream reasoning accuracy. This analysis enables a fine-grained characterization of how stable the optimization trajectory and performance gains are across repeated runs under nominally identical conditions.

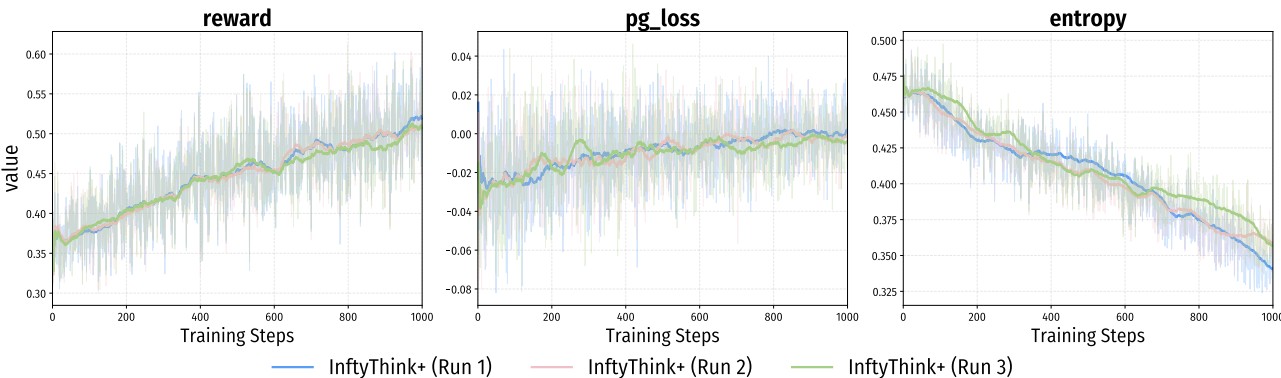

*Figure 13.* Training dynamics of InftyThink$^+$ under the task-reward-only setting across three independent runs with identical configurations.

**Consistent training dynamics.** As shown in Figure 13, despite noticeable short-term fluctuations induced by computational nondeterminism, the three runs exhibit highly consistent global training trends. The training reward increases steadily across all runs, indicating stable policy improvement with only limited variance in convergence speed and final reward magnitude. Similarly, the policy gradient loss follows a closely aligned trajectory, gradually stabilizing as training progresses, suggesting that the optimization dynamics are robust to run-to-run perturbations. The entropy curves consistently decrease over time, reflecting a coherent transition from exploration to more deterministic policies across all runs. Overall, while local noise is unavoidable, the strong alignment in both directionality and scale of these signals demonstrates that the proposed RL training procedure yields stable and reproducible optimization behavior under identical experimental settings.

**Consistent evaluation trending.** As shown in Figure 14, evaluation performance across the three benchmarks is largely consistent across repeated runs, with only moderate run-to-run variance. Notably, the accuracy curves on MATH500 are substantially smoother than those on AIME24 and AIME25. This difference can be attributed to the size of the evaluation sets: MATH500 contains 500 problems, whereas AIME24 and AIME25 each consist of only 30 problems. As a result,

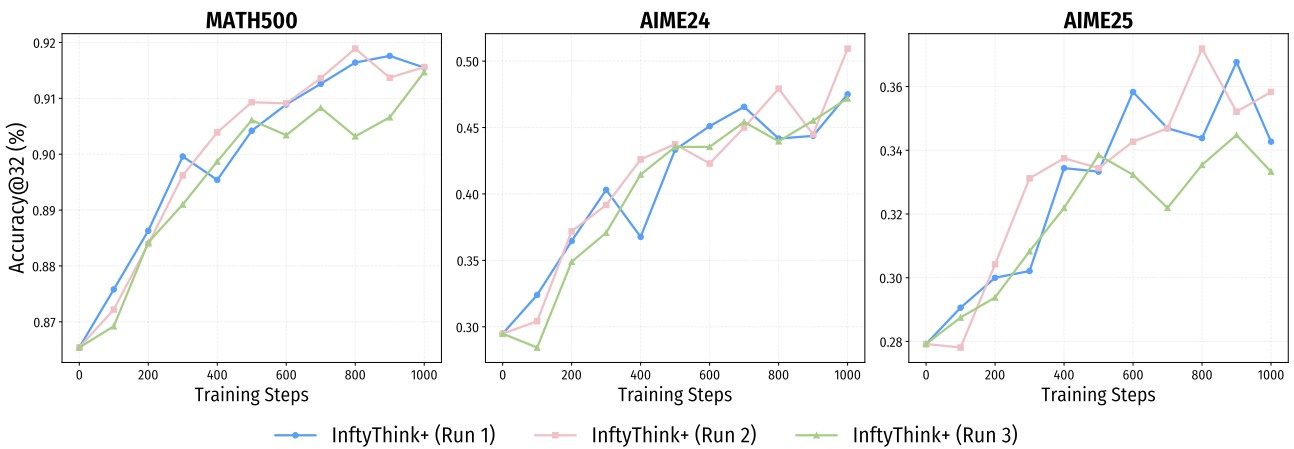

*Figure 14.* Accuracy of intermediate checkpoints evaluated on MATH500, AIME24, and AIME25 during InftyThink$^+$ RL training under the task-reward-only setting.

accuracy estimates on the AIME benchmarks are inherently more sensitive to small changes in model behavior, where a few solved or unsolved instances can lead to noticeable fluctuations in the reported accuracy. Despite this higher variance, all three runs exhibit aligned upward trends and converge to comparable final performance on AIME24 and AIME25, indicating that the observed improvements are stable rather than artifacts of evaluation noise.

## J.2. Evaluation Stability

To assess the stability of the evaluation process, we conduct a controlled study by re-evaluating all checkpoints produced during the task-reward RL training. Specifically, we perform the evaluation procedure three times under identical configurations, including the same decoding parameters, evaluation protocol, and hardware environment. By holding all evaluation conditions constant, this setup isolates the variance introduced by computational nondeterminism during inference and metric computation. As shown in Figure 15, we report the accuracy and latency trajectories on AIME24, AIME25, and MATH500 across training steps, enabling a systematic examination of the consistency of evaluation outcomes across repeated runs.

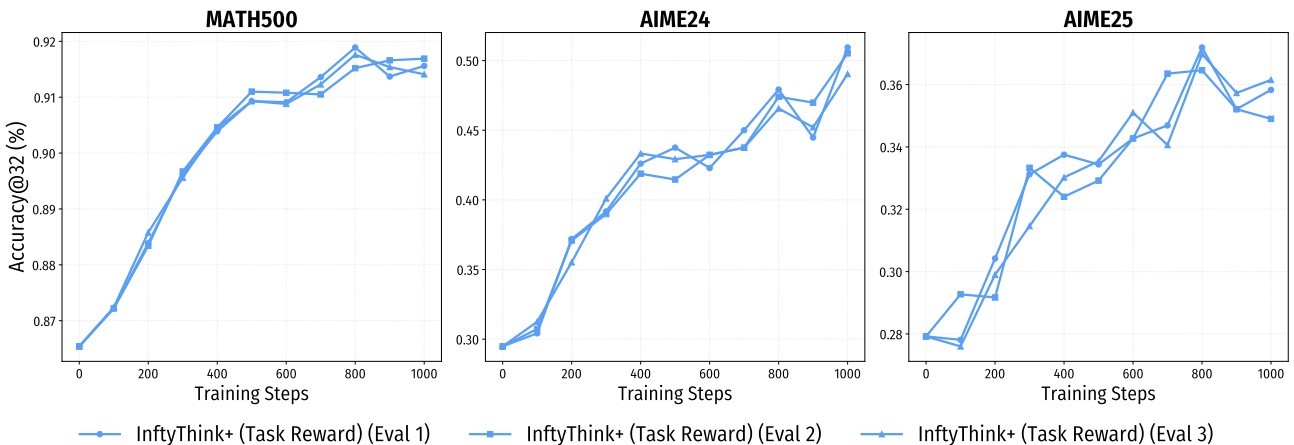

*Figure 15.* Evaluation stability (accuracy) across repeated runs.

**Accuracy.** Across all three benchmarks, the two evaluation trajectories closely overlap throughout training, indicating that the measured performance trends are highly reproducible under repeated runs. The agreement is particularly tight on MATH500, while AIME24/25 exhibit slightly larger pointwise fluctuations, which is expected given their much smaller test sizes (30 problems each), making the accuracy estimate more sensitive to minor nondeterminism in inference and metric

computation. Overall, the consistency between the two runs supports that the observed accuracy gains across RL steps reflect genuine model improvements rather than evaluation noise.

**Latency.** Figure 16 reports the inference latency measured on three benchmarks (MATH500, AIME24, and AIME25) across three repeated evaluation runs using identical checkpoints and evaluation settings. Overall, the latency curves exhibit highly consistent trends across runs, with only minor fluctuations at individual training steps. Importantly, the relative ordering and temporal evolution of latency remain stable throughout training, indicating that the observed variations are dominated by inherent system-level noise (e.g., runtime scheduling and decoding stochasticity) rather than evaluation instability. This consistency across repeated runs suggests that our latency measurements are robust and reproducible, and thus the reported efficiency comparisons and trends can be regarded as reliable. Together with the stable accuracy evaluations reported elsewhere, these results support the credibility of our evaluation protocol.

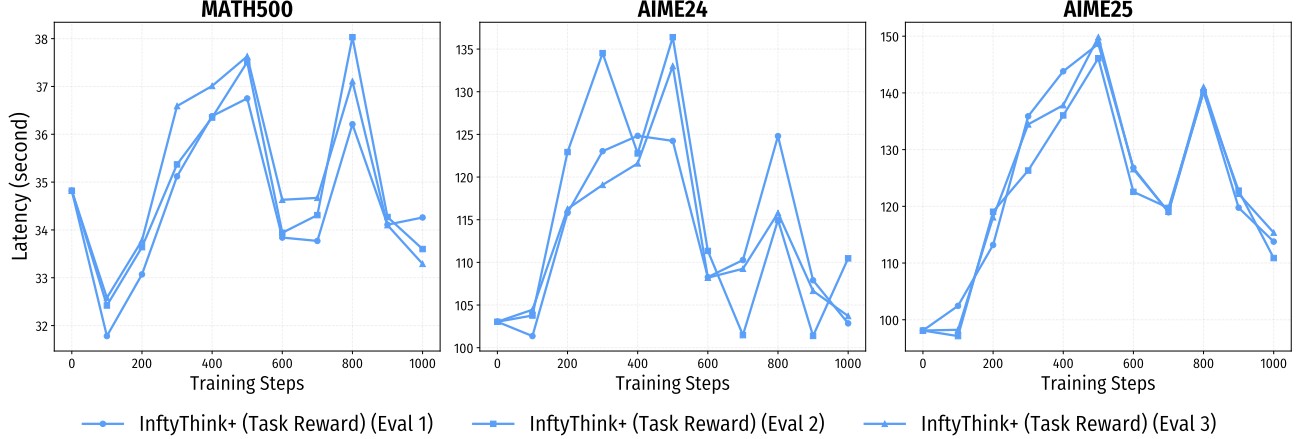

*Figure 16.* Evaluation stability (latency) across repeated runs.

# K. Theoretical Analysis

In this section, we provide theoretical justification for the design of InftyThink+. We first analyze why supervised learning is insufficient for iterative reasoning from an information-theoretic perspective (Section K.1), then establish the computational benefits of iterative reasoning over vanilla long-context generation (Section K.2).

### K.1. Information Bottleneck Analysis of Summary Quality

A fundamental question in iterative reasoning is: what constitutes a good summary? We formalize this using the Information Bottleneck framework (Tishby et al., 2000), which reveals why supervised learning is insufficient for learning optimal summaries.

#### K.1.1. PROBLEM SETUP

Let $Q$ denote the query, $R_{\leq i} = (R_1, \ldots, R_i)$ the reasoning history up to iteration $i$, $S_i$ the summary at iteration $i$, and $A \in \{0, 1\}$ the correctness of the final answer. A summary must balance two competing objectives: it should be *compressed* to fit within the context budget, yet *informative* enough to support correct subsequent reasoning.

#### K.1.2. OPTIMAL SUMMARY VIA INFORMATION BOTTLENECK

**Definition K.1** (Optimal Summary). The optimal summary $S_i^*$ at iteration $i$ is defined as the solution to the following Information Bottleneck optimization problem:

$$S_i^* = \arg\min_{S_i} \mathcal{L}_{\text{IB}}(S_i) \tag{32}$$

where the Information Bottleneck objective is:

$$\mathcal{L}_{\text{IB}}(S_i) = I(S_i; R_{\leq i} \mid Q) - \beta \cdot I(S_i; A \mid Q) \tag{33}$$

Here $I(X; Y \mid Z)$ denotes the conditional mutual information between $X$ and $Y$ given $Z$, and $\beta > 0$ is a Lagrange multiplier controlling the tradeoff between compression and informativeness.

**Interpretation.** The two terms in the objective capture the fundamental tradeoff in summarization:

- **Compression term** $I(S_i; R_{\leq i} \mid Q)$: This measures how much information the summary $S_i$ retains about the full reasoning history $R_{\leq i}$, given the query $Q$. Minimizing this term encourages the summary to discard redundant details and retain only essential information, yielding a more compressed representation.

- **Informativeness term** $I(S_i; A \mid Q)$: This measures how much information the summary preserves about the final answer correctness $A$, given the query $Q$. Maximizing this term (equivalently, minimizing its negation) ensures that the summary retains information critical for reaching the correct answer in subsequent iterations.

The parameter $\beta$ controls the relative importance of these objectives. When $\beta$ is large, the optimization prioritizes answer-relevant information; when $\beta$ is small, it prioritizes compression.

### K.1.3. LIMITATION OF SUPERVISED LEARNING

We now establish that supervised fine-tuning cannot optimize the Information Bottleneck objective, providing theoretical justification for the necessity of reinforcement learning.

**Proposition K.2** (Limitation of Supervised Learning). *Let $\mathcal{D} = \{(q^{(k)}, r^{(k)}, s^{(k)})\}_{k=1}^{N}$ be a training dataset where summaries $s^{(k)}$ are generated by an external model $M$ using fixed rules. Let $\pi_{\mathrm{SFT}}$ be the policy obtained by maximizing the log-likelihood objective:*

$$\mathcal{L}_{\mathrm{SFT}}(\theta) = \mathbb{E}_{(q,r,s) \sim \mathcal{D}} \left[ \log p_\theta(s \mid r, q) \right] \tag{34}$$

*Then $\pi_{\mathrm{SFT}}$ does not optimize the Information Bottleneck objective in Definition K.1.*

*Proof.* We prove this by showing that the SFT objective is independent of the answer correctness $A$.

**Step 1: Characterizing the SFT objective.** The SFT objective can be rewritten as:

$$\mathcal{L}_{\mathrm{SFT}}(\theta) = \mathbb{E}_{(q,r,s) \sim \mathcal{D}} \left[ \log p_\theta(s \mid r, q) \right] \tag{35}$$
$$= -H_{\mathcal{D}}(S \mid R, Q) - D_{\mathrm{KL}} \left( p_{\mathcal{D}}(S \mid R, Q) \,\|\, p_\theta(S \mid R, Q) \right) \tag{36}$$

where $H_{\mathcal{D}}(S \mid R, Q)$ is the conditional entropy of summaries in the dataset, and $D_{\mathrm{KL}}(\cdot \| \cdot)$ denotes the Kullback-Leibler divergence.

Since $H_{\mathcal{D}}(S \mid R, Q)$ is a constant with respect to $\theta$, maximizing $\mathcal{L}_{\mathrm{SFT}}(\theta)$ is equivalent to minimizing:

$$D_{\mathrm{KL}} \left( p_{\mathcal{D}}(S \mid R, Q) \,\|\, p_\theta(S \mid R, Q) \right) \tag{37}$$

**Step 2: Independence from answer correctness.** The data distribution $p_{\mathcal{D}}(S \mid R, Q)$ is determined entirely by the external model $M$ and the fixed transformation rules used to construct $\mathcal{D}$. Crucially, this distribution does not depend on the final answer correctness $A$ because:

1. The summaries in $\mathcal{D}$ are generated by $M$ based solely on $(Q, R)$, without access to whether the reasoning will ultimately lead to a correct answer.

2. The transformation rules are deterministic functions of the reasoning text, independent of answer correctness.

Therefore, the SFT objective can be written as:

$$\mathcal{L}_{\mathrm{SFT}}(\theta) = f(p_{\mathcal{D}}, p_\theta) \tag{38}$$

where $f$ is some function that does not involve $A$. This means $\frac{\partial \mathcal{L}_{\mathrm{SFT}}}{\partial I(S;A|Q)} = 0$, so SFT does not optimize the informativeness term $I(S_i; A \mid Q)$ in the Information Bottleneck objective.

**Step 3: Distribution mismatch.** Even if SFT perfectly fits the data distribution (i.e., $p_\theta = p_\mathcal{D}$), the resulting policy may still produce suboptimal summaries for the current policy $\pi_\theta$. This is because the summaries in $\mathcal{D}$ were generated by $M$, whose internal representations and continuation capabilities may differ from $\pi_\theta$. Formally, let $S_M^*$ denote the optimal summary for model $M$ and $S_\theta^*$ denote the optimal summary for policy $\pi_\theta$. In general:

$$S_M^* \neq S_\theta^* \tag{39}$$

because the information required for $M$ to continue reasoning correctly may differ from what $\pi_\theta$ requires.

**Conclusion.** Combining Steps 2 and 3, we conclude that $\pi_\text{SFT}$ optimizes neither the informativeness term (due to independence from $A$) nor produces summaries aligned with its own continuation capabilities (due to distribution mismatch). Therefore, $\pi_\text{SFT}$ does not optimize the Information Bottleneck objective. $\square$

*Remark* K.3 (How RL Addresses These Limitations). Reinforcement learning with outcome-based rewards addresses both limitations identified in Proposition K.2:

1. **Optimizing informativeness**: By using final answer correctness as the reward signal, RL directly optimizes for summaries that lead to correct answers. This implicitly maximizes $I(S_i; A \mid Q)$, as summaries that preserve answer-relevant information will receive higher rewards on average.

2. **Aligning with policy capabilities**: During RL training, the policy generates its own summaries and must continue reasoning from them. This closed-loop optimization naturally aligns the compression strategy with the policy's continuation capabilities, ensuring $S_\theta^*$ is optimized for $\pi_\theta$ rather than some external model $M$.

## K.2. Computational Complexity Analysis

We briefly analyze the computational benefits of InftyThink compared to vanilla long-context reasoning.

**Proposition K.4** (Complexity Reduction). *Let $L$ denote the total reasoning length under vanilla reasoning, and suppose InftyThink decomposes this into $n$ iterations, each generating at most $\ell$ reasoning tokens and $m$ summary tokens, where $L \approx n\ell$. Under the standard Transformer architecture with $O(L^2)$ self-attention complexity, the computational cost satisfies:*

$$\frac{\text{Cost}_\text{InftyThink}}{\text{Cost}_\text{Vanilla}} \approx \frac{n(\ell + m)^2}{L^2} = \frac{(\ell + m)^2}{n\ell^2} \tag{40}$$

*When $m \ll \ell$ and $n > 1$, this ratio is strictly less than 1, indicating reduced computational cost.*

*Proof.* Under vanilla reasoning, the model generates $L$ tokens in a single forward pass. Due to the $O(L^2)$ complexity of self-attention, the total computational cost scales as:

$$\text{Cost}_\text{Vanilla} = O(L^2) \tag{41}$$

Under InftyThink, the model performs $n$ iterations. At iteration $j$, the context consists of the query (length $|q|$), the previous summary (length $m$), and the current generation (up to $\ell + m$ tokens including reasoning and new summary). The per-iteration cost is:

$$\text{Cost}_\text{iter} = O((|q| + m + \ell + m)^2) = O((|q| + \ell + 2m)^2) \tag{42}$$

Assuming $|q| \ll \ell$ and summing over $n$ iterations:

$$\text{Cost}_\text{InftyThink} = O(n(\ell + 2m)^2) \tag{43}$$

Taking the ratio and using $L = n\ell$:

$$\frac{\text{Cost}_\text{InftyThink}}{\text{Cost}_\text{Vanilla}} = \frac{n(\ell + 2m)^2}{(n\ell)^2} = \frac{(\ell + 2m)^2}{n\ell^2} \tag{44}$$

When $m \ll \ell$, we have $(\ell + 2m)^2 \approx \ell^2$, so the ratio simplifies to approximately $1/n < 1$ for $n > 1$. $\square$

## L. Reinforcement Learning without Cold Start

**Why cold start is necessary.**    Although RL can effectively shape long-context reasoning behaviors, starting from a purely pretrained base model often leads to unstable optimization and poor sample efficiency, especially when the target behavior involves a *structured* multi-iteration reasoning protocol. In InftyThink+, the policy is required to (i) follow a strict iterative format, (ii) produce summaries that serve as valid intermediate-state representations, and (iii) continue reasoning conditioned on compressed context across iterations. Without an explicit warm-up, the initial policy typically fails to reliably emit well-formed summaries or to align its continuation strategy with the InftyThink+ paradigm; consequently, the reward signal becomes sparse/noisy and the policy-gradient updates are dominated by exploration artifacts rather than meaningful credit assignment. These issues make a cold-start stage (e.g., a brief SFT warm-up) practically indispensable for InftyThink+, as it anchors the model to the desired trajectory structure and ensures that RL operates on *on-manifold* rollouts.

**A fair RL-only baseline without cold start.**    Nevertheless, to ensure a fair comparison and to quantify the role of cold start, we conduct an additional experiment where we remove cold start entirely and apply RL directly on the vanilla model. This *RL-from-scratch* baseline uses the same training configuration and system environment as our main experiments (e.g., identical rollout setup, reward computation, optimization hyperparameters, and infrastructure), with the sole difference being the absence of cold-start initialization. We report both the training dynamics, trajectory reward, policy-gradient loss (`pg_loss`), and entropy, as well as downstream evaluation on MATH500, AIME24, and AIME25. Together, these results isolate the effect of cold start and validate our claim that InftyThink+ requires cold start for stable and effective RL, while providing an apples-to-apples RL-only baseline for comparison.

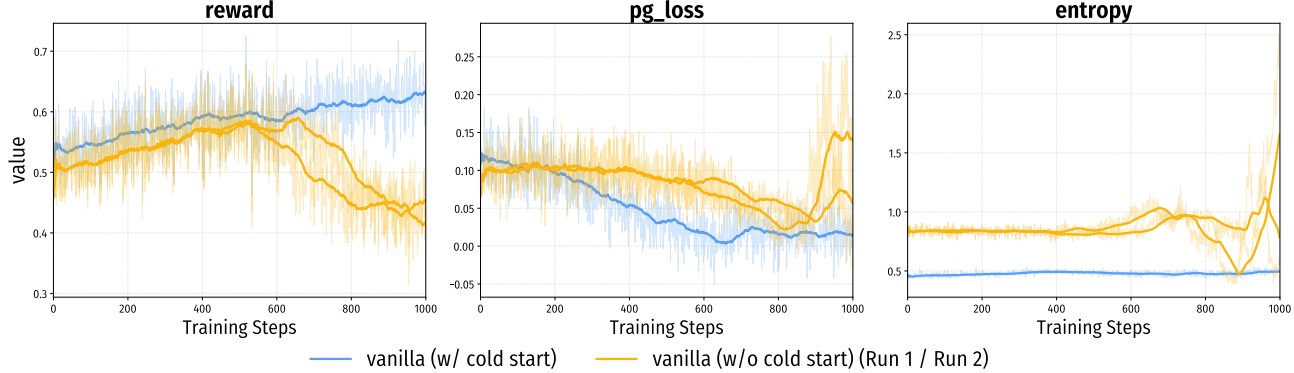

*Figure 17.* Training dynamics of vanilla RL with vs. without cold start.

Figure 17 shows that cold start is critical for stabilizing vanilla RL. With cold start (blue), training is well-behaved: reward increases steadily, `pg_loss` decreases smoothly, and entropy remains low and stable, indicating consistent policy improvement under a controlled exploration regime. In contrast, removing cold start (yellow) leads to a characteristic *collapse*: reward initially improves but then drops sharply after mid-training, accompanied by a pronounced rise and spike in `pg_loss` and a rapid entropy explosion near the end, suggesting that optimization becomes unstable and the policy drifts into high-variance, overly stochastic behaviors. Importantly, to rule out that this collapse is a one-off artifact, we rerun the same no-cold-start experiment under identical configurations (Run 1/Run 2); both runs reproduce the same failure pattern, confirming that the instability is systematic rather than accidental.

Figure 18 further corroborates the collapse observed in Figure 15 from a downstream evaluation perspective. With cold start (blue), performance improves monotonically (or near-monotonically) on all three benchmarks as training proceeds, indicating stable policy refinement. In contrast, the no-cold-start setting (yellow) shows a deceptive early gain, accuracy rises during the first half of training, but then undergoes a sharp degradation after the collapse point, with substantial drops on MATH500 and even more pronounced failures on the harder AIME24/AIME25. This synchronized late-stage decline aligns with the simultaneous reward drop, `pg_loss` spike, and entropy explosion in Figure 17, suggesting that once training becomes unstable, the policy drifts away from effective reasoning behaviors and loses generalization. Together, these results highlight that cold start is not merely a convenience but a prerequisite for preventing systematic RL collapse under our setting.

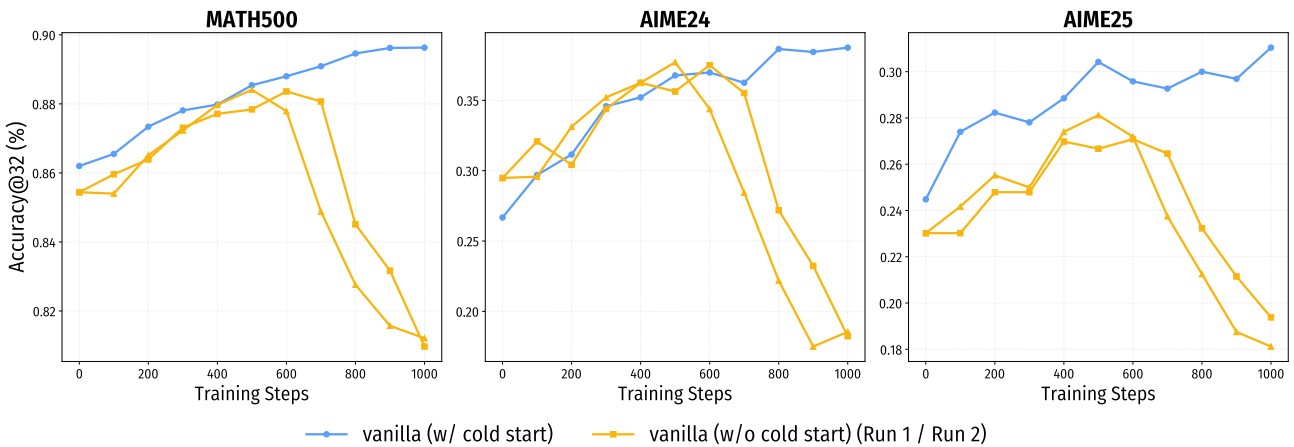

*Figure 18.* Benchmark performance along training for vanilla RL with vs. without cold start.

## M. Performance across Reasoning Iteration Rounds

Figure 19 characterizes how test-time iterative reasoning translates into measurable accuracy gains. Across all three benchmarks, increasing the number of reasoning rounds consistently improves performance, indicating that the intermediate summaries serve as effective state abstractions that enable progressive refinement rather than redundant continuation. Notably, the gains are highly front-loaded: the first few rounds yield the largest improvements, after which the curves gradually saturate, reflecting diminishing returns as additional rounds mainly polish already-correct trajectories or revisit similar subgoals.

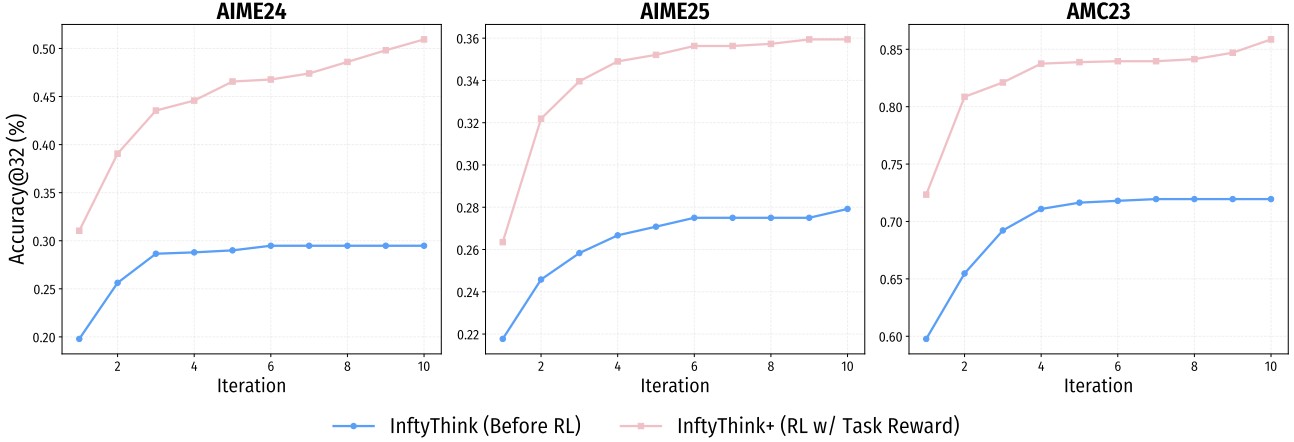

*Figure 19.* Accuracy across of reasoning iterations. We report Accuracy@32 on AIME24, AIME25, and AMC23 when allowing the model to perform up to $j$ InftyThink-style reasoning rounds (x-axis), where each round is connected by the model-generated summary.

A clear gap emerges between the two settings. Before RL, the model quickly reaches a plateau (especially on AIME24/25), suggesting that it cannot reliably exploit deeper iterative compute beyond a small number of rounds. In contrast, InftyThink+ trained with task-reward RL exhibits both *higher* accuracy at every iteration and a *stronger* scaling trend with more rounds, with improvements continuing into later iterations. This implies that RL not only boosts per-round competence, but also improves the model's ability to utilize additional iteration budget effectively, turning extra rounds into meaningful progress rather than unproductive repetition. Overall, these results motivate treating the iteration cap as an explicit test-time compute knob: larger budgets provide higher accuracy, but with diminishing marginal utility, and RL makes this knob substantially more effective.

# N. Inference Latency Distribution

Beyond average token budgets, practical deployment depends critically on the *distribution* of per-instance inference time. Figure 20 reports end-to-end latency (in seconds, log-scale) on MATH500, AIME24, and AIME25 for vanilla decoding and InftyThink+ under different training settings. Across all benchmarks, InftyThink+ consistently shifts the latency distribution left and reduces the heavy right tail, indicating fewer extremely long rollouts.

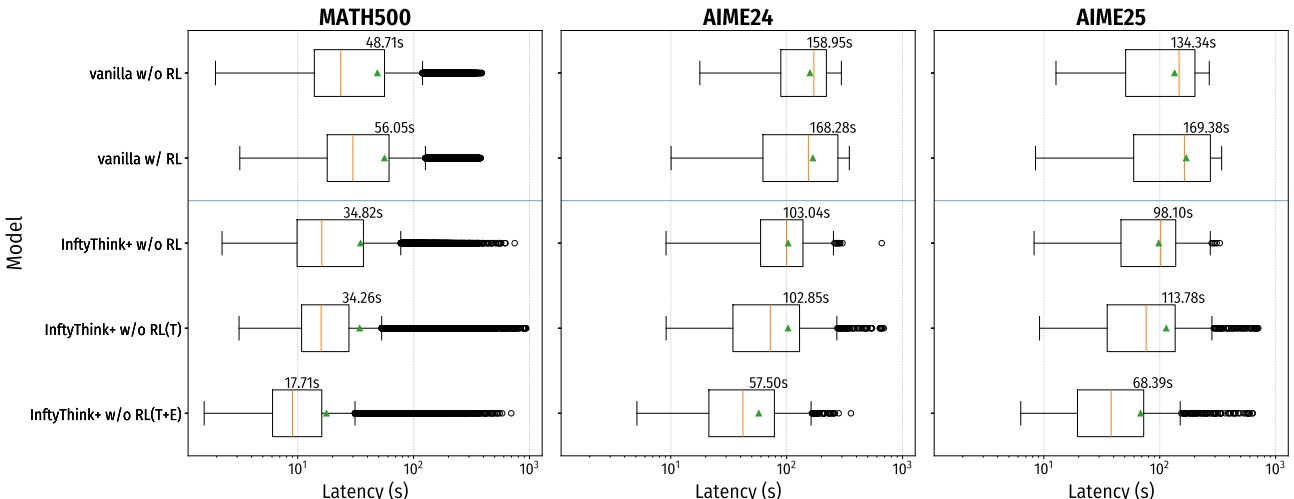

*Figure 20.* **Inference latency distribution** on MATH500, AIME24, and AIME25 (log-scale x-axis). Each row corresponds to one model variant. Boxes denote the interquartile range (IQR) with the median marked in orange; green triangles indicate the mean; whiskers extend to 1.5×IQR; dots are outliers. InftyThink+ shifts the distribution left and suppresses the long-latency tail, while adding the efficiency reward (RL(T+E)) yields the largest latency reduction.

A notable observation is that vanilla RL slightly *increases* latency (e.g., mean latency from 48.71s to 56.05s on MATH500, and from 134.34s to 169.38s on AIME25), suggesting that RL alone can encourage longer CoT without explicitly accounting for efficiency. In contrast, InftyThink+ achieves substantially lower latency even without RL (e.g., 34.82s on MATH500 and 103.04s on AIME24), reflecting the benefit of iterative summarization in constraining context growth and stabilizing generation length. When RL is applied with task reward only (RL(T)), the latency remains comparable to the InftyThink+ w/o RL setting (e.g., 34.26s vs. 34.82s on MATH500), implying that the InftyThink+ inference format already provides a strong inductive bias toward shorter rollouts.

Finally, incorporating the efficiency reward (RL(T+E)) yields the largest improvement, producing both lower central tendency and a visibly shorter tail. In particular, RL(T+E) reduces mean latency to 17.71s (MATH500), 57.50s (AIME24), and 68.39s (AIME25), corresponding to ∼2.75–3.16× speedup over vanilla on MATH500, ∼2.76–2.93× on AIME24, and ∼1.96–2.48× on AIME25. This aligns with our design: smaller iteration counts (and thus higher efficiency reward) are explicitly preferred, yielding faster and more predictable inference.

# O. Hyper-parameter Ablation Study

In this section, we conduct an ablation study of the key hyperparameters in InftyThink$^+$, aiming to disentangle how each design choice affects both effectiveness and efficiency. Concretely, we vary (i) the maximum number of iterative reasoning rounds $\varphi$, which directly controls the long-horizon depth of InftyThink-style rollouts (Appendix O.1) and (ii) the summarization context window size $\eta$, which determines how many recent tokens are retained when compressing intermediate reasoning states (Appendix O.2).

## O.1. Ablation of Iteration Cap Parameter $\varphi$.

The iteration cap $\varphi$ is a central hyperparameter in InftyThink$^+$, as it directly bounds the maximum number of InftyThink-style reasoning rounds within a single rollout, thereby controlling the effective reasoning horizon and the associated computation budget. To isolate its impact, we conduct an ablation study under the same setup as our main experiment with DeepSeek-R1-Distill-Qwen-1.5B, while only varying $\varphi \in \{3, 5, 10\}$. For each setting, we visualize the full training

dynamics, including the overall reward, policy-gradient loss (pg_loss), and entropy, together with InftyThink-specific signals such as the task reward, efficiency reward, and the realized number of turns per rollout (num_turns). We further report performance trajectories on MATH500, AIME24, and AIME25 throughout training, enabling a direct comparison of how different iteration budgets shape both optimization behavior and downstream reasoning capability.

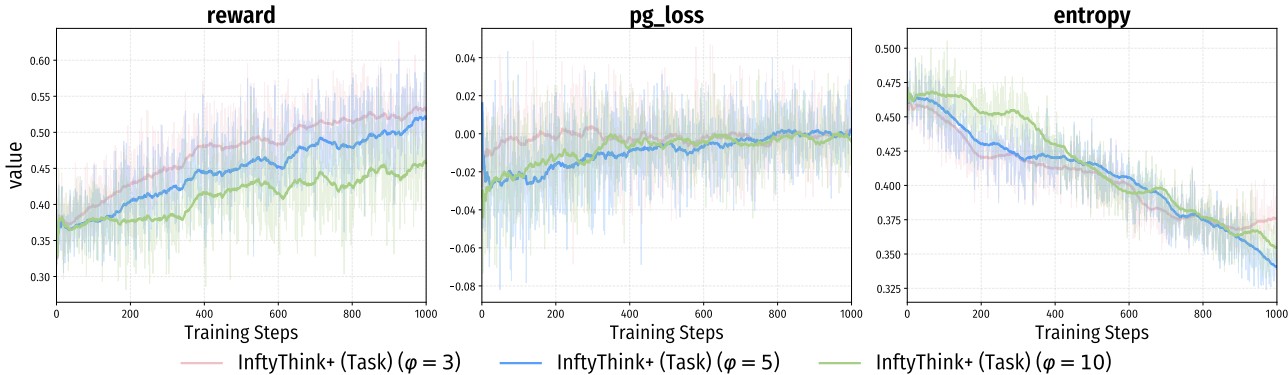

*(a)* Training dynamics of InftyThink$^+$ with **task reward only** under different iteration caps $\varphi \in \{3, 5, 10\}$.

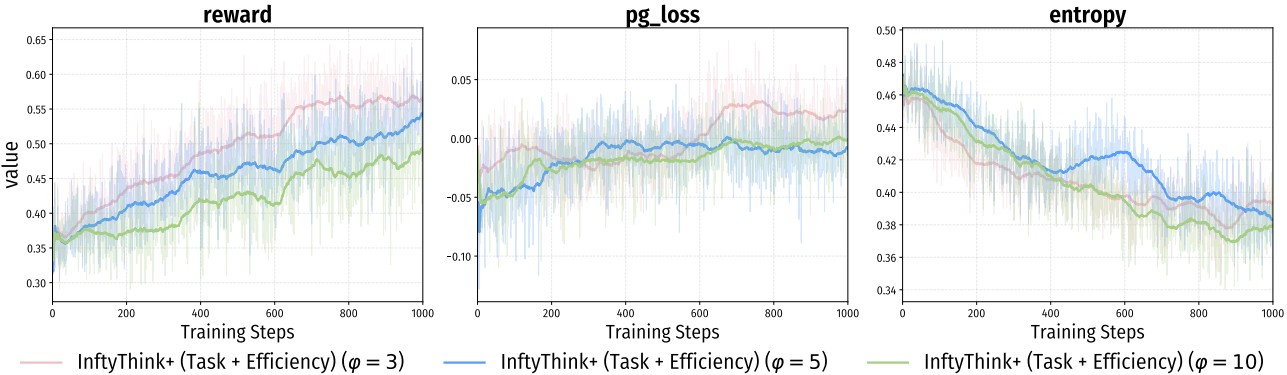

*(b)* Training dynamics of InftyThink$^+$ with **task + efficiency** rewards under different iteration caps $\varphi \in \{3, 5, 10\}$.

*Figure 21.* Effect of the iteration cap $\varphi$ on RL optimization dynamics in InftyThink$^+$. The top row uses task reward only, while the bottom row additionally incorporates an efficiency reward. Across settings, we compare reward, pg_loss, and entropy over training steps.

**RL optimization dynamics.** Figure 21 shows that $\varphi$ substantially affects both learning speed and the final optimization level. In both reward settings, smaller iteration caps consistently yield higher rewards throughout training: $\varphi = 3$ converges fastest and reaches the best reward, $\varphi = 5$ is a close second, while $\varphi = 10$ lags behind with a persistent gap. This trend is consistent with the intuition that larger $\varphi$ induces longer-horizon rollouts, which increases trajectory variance and makes credit assignment harder, thereby slowing down reward improvement. The pg_loss curves further suggest stable optimization across all $\varphi$ values, as the loss magnitude gradually shrinks toward 0. However, $\varphi = 10$ exhibits more conservative (closer-to-zero) updates for a longer portion of training, aligning with its slower reward gains. Entropy decreases monotonically in all cases, indicating reduced exploration as training proceeds; notably, the decay is more pronounced for larger $\varphi$ (especially in the Task+Efficiency setting), which is consistent with the policy being increasingly pressured to terminate earlier and avoid unproductive long rollouts. Overall, these results indicate a clear performance–horizon trade-off: allowing too many iterations ($\varphi = 10$) can hurt optimization efficiency and yield diminishing returns, while moderate iteration budgets ($\varphi \in \{3, 5\}$) strike a better balance between stable training and effective improvement.

**Effect on InftyThink-specific behavior.** Figure 22 clarifies the mechanism behind the reward differences observed in the training dynamics. **First**, $\varphi$ behaves as an effective *compute budget knob*: the realized number of turns (num_turns) strongly correlates with the cap. With task reward only (top), $\varphi=3$ quickly saturates at $\approx 2$ turns, $\varphi=5$ stabilizes around $\approx 3$ turns, while $\varphi=10$ climbs to $\approx 5$–6 turns, indicating that the model indeed exploits the larger iteration budget to

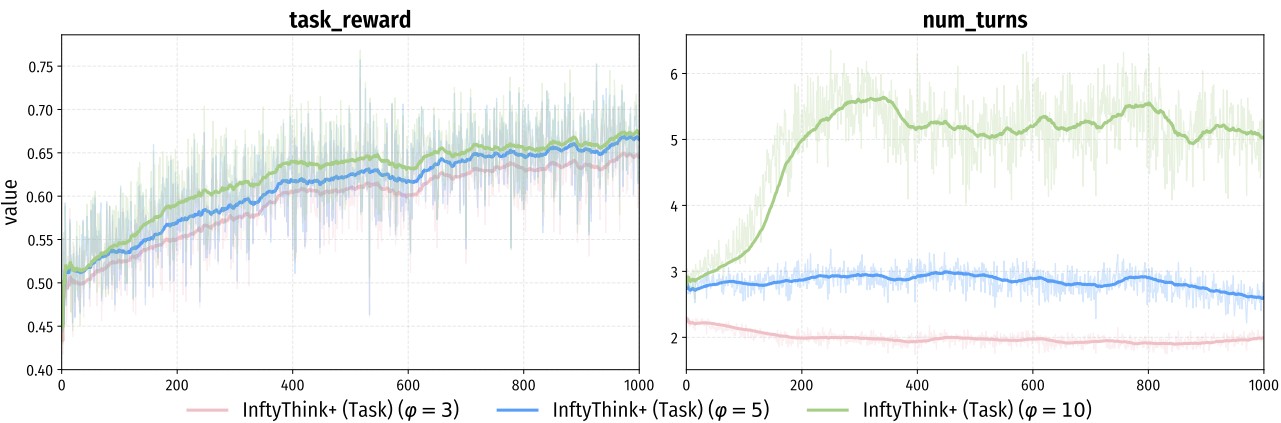

*(a)* InftyThink-related metrics of InftyThink$^+$ trained with **task reward only** under different iteration caps $\varphi \in \{3, 5, 10\}$.

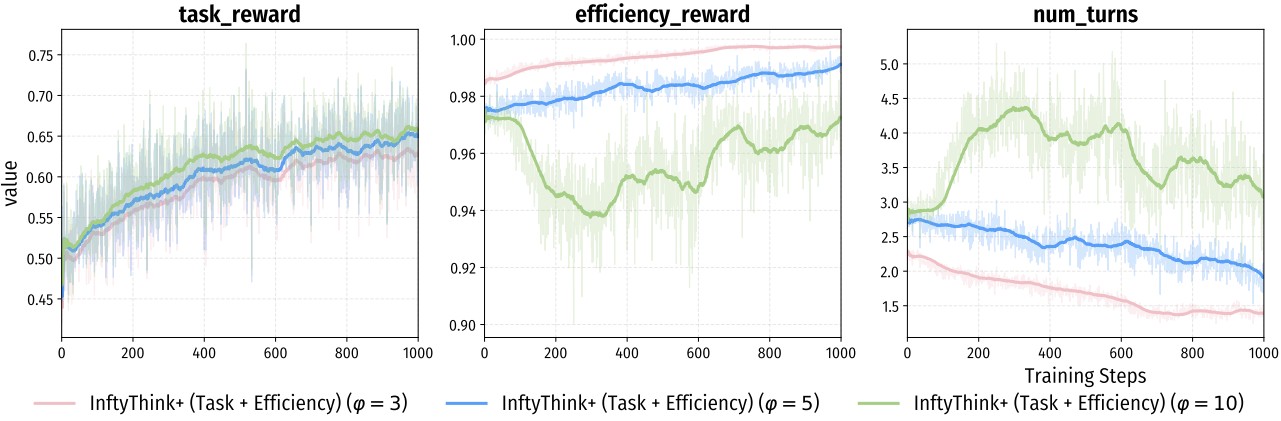

*(b)* InftyThink-related metrics of InftyThink$^+$ trained with **task + efficiency** rewards under different iteration caps $\varphi \in \{3, 5, 10\}$.

*Figure 22.* Impact of the iteration cap $\varphi$ on InftyThink-specific training metrics. The top row isolates task reward and realized turns, while the bottom row additionally includes the efficiency reward, revealing how $\varphi$ controls the rollout horizon and consequently the attainable efficiency signal.

perform more iterative refinement. **Second**, the task reward itself is *not* maximized by smaller $\varphi$. Across both settings, larger iteration budgets achieve slightly higher `task_reward` (most noticeably $\varphi$=10), suggesting that allowing additional rounds can improve task-level outcomes by enabling more opportunities for correction and refinement. This observation is crucial: the higher *overall* reward seen for small $\varphi$ in Figure 21 should not be interpreted as better task competence. **Third**, when the efficiency reward is enabled (bottom), the trade-off becomes explicit. Smaller $\varphi$ yields consistently higher `efficiency_reward`, because fewer turns translate into shorter rollouts and lower latency; conversely, $\varphi$=10 incurs a clear penalty in `efficiency_reward` as it produces longer trajectories. Meanwhile, the `num_turns` curves decrease mildly over training (especially for $\varphi$=3, 5), indicating that the efficiency shaping encourages the policy to learn earlier termination once sufficient progress is made.

Taken together, Figure 22 demonstrates that $\varphi$ primarily governs the *efficiency–performance frontier*: increasing $\varphi$ can modestly improve task reward by enabling deeper iterative reasoning, but this comes at a direct cost in efficiency reward due to longer rollouts. Therefore, comparing settings solely via the aggregated training reward can be misleading; a fair assessment must jointly consider `task_reward`, `efficiency_reward`, and the realized `num_turns`, and ultimately validate the trade-off on downstream benchmarks.

**Inference latency and efficiency trade-offs.** Figure 23 demonstrates that increasing the iteration cap generally improves downstream accuracy, especially on the more challenging AIME benchmarks. In the `Task` setting (top), $\varphi$=10 consistently achieves the strongest performance on MATH500 and reaches higher accuracy earlier, indicating that a larger iteration

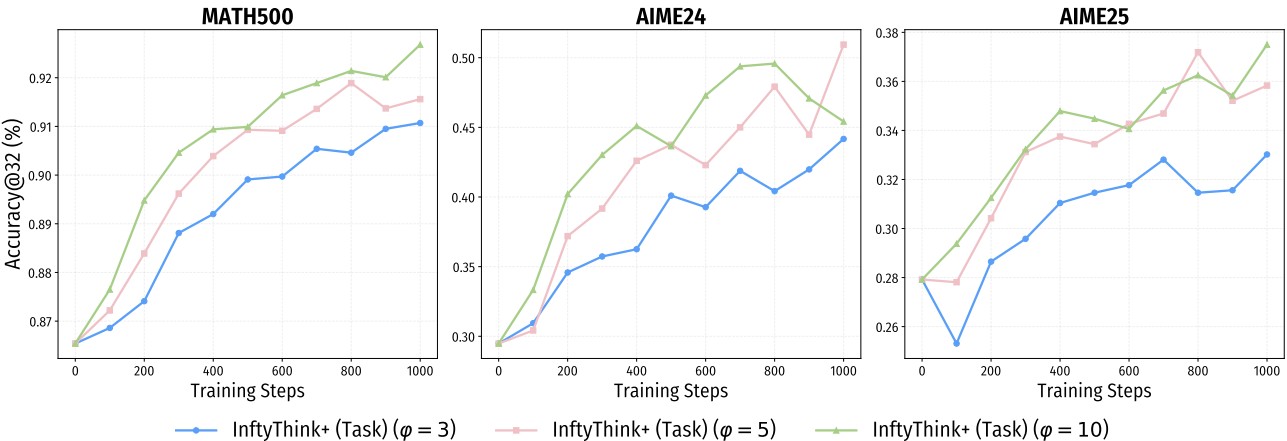

*(a)* Evaluation trajectories of InftyThink$^+$ trained with **task reward only** under different iteration caps $\varphi \in \{3, 5, 10\}$.

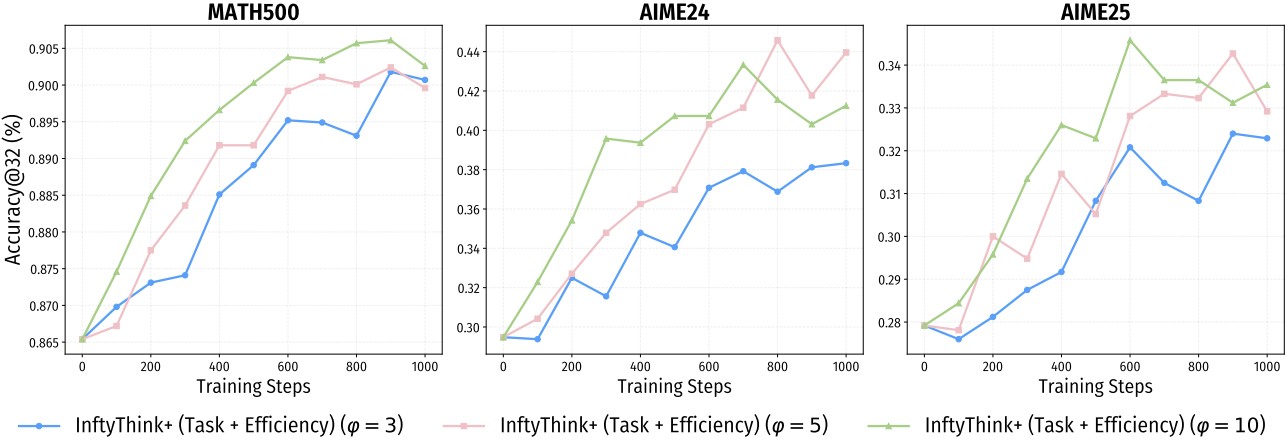

*(b)* Evaluation trajectories of InftyThink$^+$ trained with **task + efficiency** rewards under different iteration caps $\varphi \in \{3, 5, 10\}$.

*Figure 23.* Impact of the iteration cap $\varphi$ on downstream mathematical reasoning performance throughout RL training. The top row uses task reward only, while the bottom row additionally includes an efficiency reward. Each curve tracks avg@32 accuracy on MATH500/AIME24/AIME25 over training steps.

budget better supports long-horizon refinement for hard math problems. The gap is more pronounced on AIME24/AIME25: while $\varphi=3$ exhibits a slower and lower improvement curve, both $\varphi=5$ and $\varphi=10$ steadily climb to higher final accuracies, suggesting that additional reasoning rounds materially benefit difficult contest-style problems.

When enabling the efficiency reward (bottom), all settings incur a mild accuracy drop compared to task-only training, reflecting the expected trade-off between correctness and compute. Nevertheless, the relative ranking across $\varphi$ remains largely consistent: $\varphi=10$ still dominates most of training on MATH500 and remains competitive (often best) on AIME24/AIME25, whereas $\varphi=3$ tends to underperform. Interestingly, the margin between $\varphi=5$ and $\varphi=10$ becomes smaller under `Task+Efficiency`, which aligns with the efficiency shaping discouraging excessive turns and partially neutralizing the advantage of a very large iteration cap.

Taken together with Figures 21–22, these results suggest that (i) smaller $\varphi$ can achieve higher *training reward* mainly due to higher efficiency reward, but (ii) larger $\varphi$ delivers more consistent *task-level gains* on downstream benchmarks by allowing deeper iterative refinement. Therefore, $\varphi$ primarily controls the accuracy–efficiency frontier: $\varphi=3$ favors cheaper rollouts with weaker final accuracy, while $\varphi\in\{5, 10\}$ provides stronger reasoning performance, with $\varphi=5$ often offering a better balance when efficiency shaping is enabled.

## O.2. Ablation of Context Window Size Parameter $\eta$.

We study the effect of the context window size parameter $\eta$, which bounds the amount of summarized historical context available to each reasoning iteration. Analogous to the iteration cap $\varphi$, $\eta$ acts as a fundamental control variable in InftyThink$^+$, but along a complementary dimension: while $\varphi$ limits *how long* the reasoning process can unfold, $\eta$ determines *how much past information* each iteration can condition on. We evaluate $\eta \in \{4\text{k}, 6\text{k}, 8\text{k}\}$ under otherwise identical RL settings.

**RL optimization dynamics.** Figure 24 compares reward, policy gradient loss, and entropy across different $\eta$ values. Larger context windows consistently achieve higher rewards throughout training, with $\eta = 8\text{k}$ exhibiting the strongest asymptotic performance. This mirrors the trend observed when increasing $\varphi$: providing the model with a larger effective reasoning budget allows RL to more effectively convert additional capacity into improved task performance. Importantly, the policy gradient loss remains well-behaved across all settings, indicating that enlarging the context window does not introduce optimization instability. Entropy decreases steadily during training, while larger $\eta$ maintains higher entropy at comparable steps, suggesting delayed policy collapse due to an expanded effective action space induced by richer contextual conditioning.

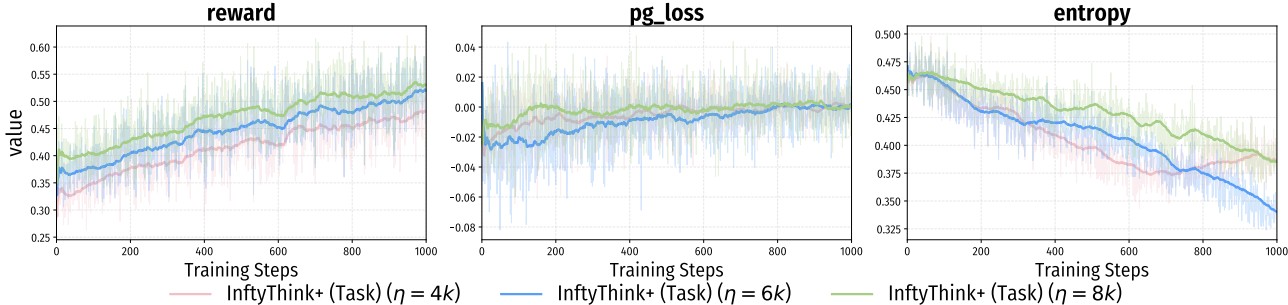

*Figure 24.* Effect of the context window size $\eta \in \{4k, 6k, 8k\}$ on RL optimization dynamics in InftyThink$^+$. Across settings, we compare reward, pg_loss, and entropy over training steps.

**Effect on InftyThink-specific behavior.** Figure 25 examines task reward and the average number of reasoning turns. While task reward improves across all settings, larger $\eta$ yields marginally higher final task rewards, indicating improved per-iteration reasoning quality. More notably, $\eta$ strongly regulates the rollout horizon: increasing $\eta$ substantially reduces the number of reasoning turns. This behavior is complementary to the effect of $\varphi$. Rather than explicitly allowing more iterations, a larger context window enables each iteration to absorb more summarized history, resulting in more information-dense reasoning steps and reducing the need for prolonged rollouts. As a consequence, the attainable efficiency reward becomes increasingly constrained for larger $\eta$, reflecting an inherent trade-off between context capacity and iteration-level efficiency incentives.

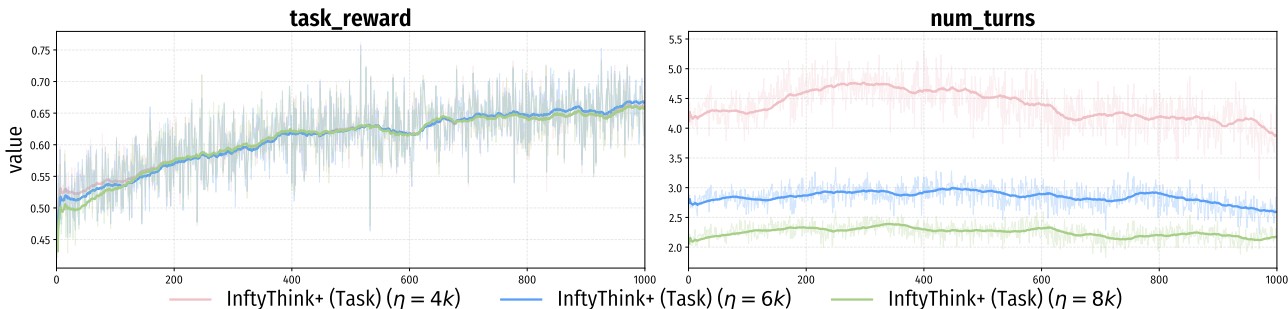

*Figure 25.* Impact of the context window size $\eta \in \{4k, 6k, 8k\}$ on InftyThink-specific training metrics, revealing how $\eta$ controls the rollout horizon and consequently the attainable efficiency signal.

**Downstream reasoning performance.** Figure 26 reports avg@32 accuracy on MATH500, AIME24, and AIME25 throughout RL training. Consistent with the training dynamics, larger context windows generally lead to stronger and more stable performance, particularly in later training stages. Smaller $\eta$ settings occasionally achieve competitive early gains but plateau earlier and exhibit higher variance. This parallels the behavior observed for smaller $\varphi$, reinforcing the conclusion that insufficient reasoning capacity—whether in iteration depth or contextual coverage—limits the model's ability to sustain long-context reasoning improvements.

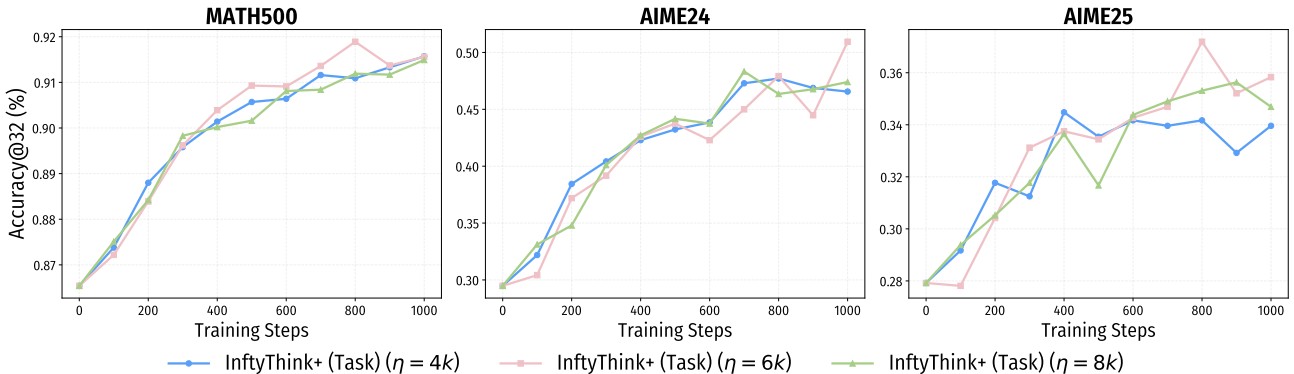

*Figure 26.* Impact of the context window size $\eta \in \{4k, 6k, 8k\}$ on downstream mathematical reasoning performance throughout RL training. Each curve tracks avg@32 accuracy on MATH500/AIME24/AIME25 over training steps.

**Inference latency and efficiency trade-offs.** Figure 27 shows inference latency across benchmarks. As expected, increasing $\eta$ leads to consistently higher latency due to increased per-step token processing costs. Compared to $\eta = 8k$, $\eta = 6k$ often achieves comparable accuracy with substantially lower latency, indicating a favorable efficiency–performance trade-off. Together with the reduced number of reasoning turns observed for larger $\eta$, these results highlight a nuanced interaction between context capacity and system-level efficiency.

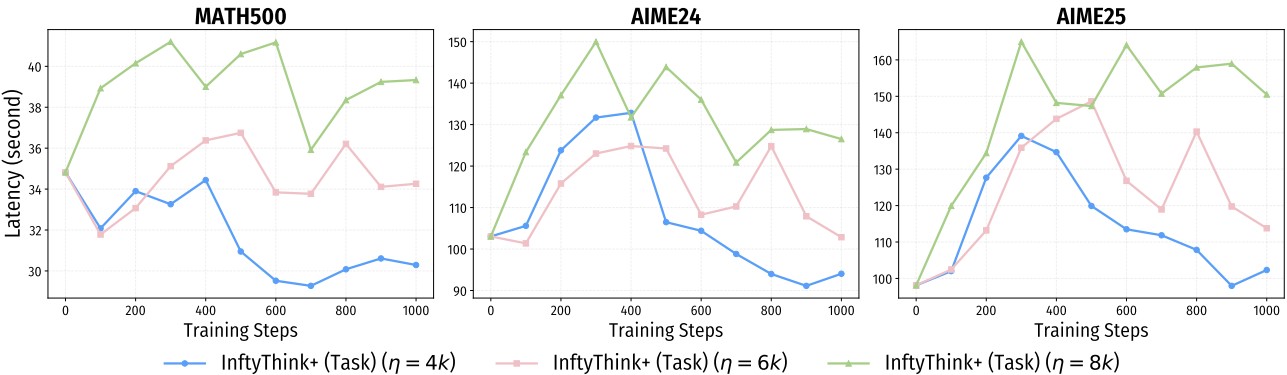

*Figure 27.* Impact of the context window size $\eta \in \{4k, 6k, 8k\}$ on inference latency (second) throughout RL training. Each curve tracks avg@32 latency on MATH500/AIME24/AIME25 over training steps.

**Discussion.** Taken together, $\eta$ serves as an explicit test-time and training-time compute knob, complementary to $\varphi$. While $\varphi$ governs the maximum reasoning depth, $\eta$ controls the breadth of historical context accessible at each step. Larger values of $\eta$ improve long-context reasoning and final accuracy, but at the cost of increased inference latency and diminished efficiency reward headroom. In practice, moderate context sizes offer a strong balance between reasoning strength and efficiency, whereas larger windows are most beneficial when maximizing reasoning performance is the primary objective.

## P. Ablation on Efficiency Reward Decay

We further study how the choice of the efficiency-reward decay affects the training dynamics of InftyThink+. Let $\tau$ denote a reasoning trajectory, and let $K(\tau)$ be the number of reasoning iterations used before the trajectory terminates. Given the maximum turn budget $\phi$, we define the normalized turn ratio as

$$x(\tau) = \frac{K(\tau) - 1}{\phi}. \tag{45}$$

We subtract one from $K(\tau)$ so that a trajectory completed within the first reasoning iteration receives no efficiency penalty. The efficiency reward is applied as a multiplicative factor to the task reward:

$$R(\tau) = R_{\text{task}}(\tau) \cdot R_{\text{eff}}(\tau), \tag{46}$$

which ensures that incorrect trajectories cannot obtain positive reward merely by terminating early. We compare three monotonic decay functions for $R_{\text{eff}}(\tau)$:

$$R_{\text{eff}}^{\text{quad}}(\tau) = 1 - x(\tau)^2, \tag{47}$$
$$R_{\text{eff}}^{\text{lin}}(\tau) = 1 - x(\tau), \tag{48}$$
$$R_{\text{eff}}^{\text{log}}(\tau) = 1 - \log_2(1 + x(\tau)). \tag{49}$$

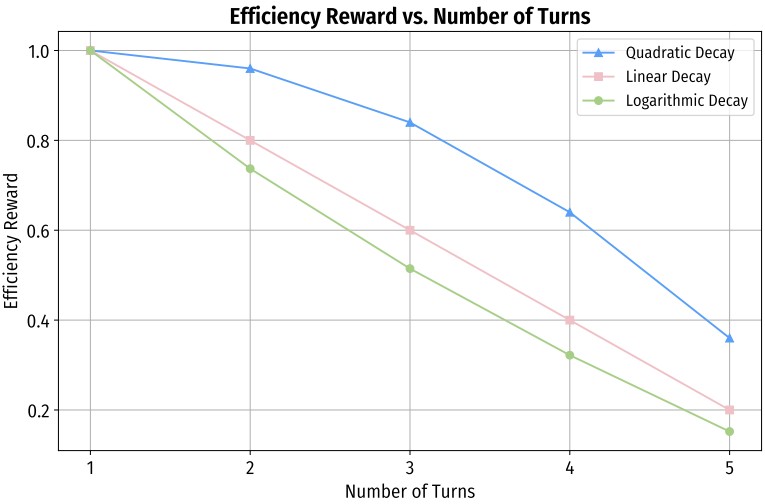

*Figure 28.* Illustration of how the efficiency reward varies with the number of reasoning iterations in a trajectory. We show the quadratic-style decay used in this paper, together with two comparison variants: linear-style decay and logarithmic-style decay.

All three variants assign the maximum efficiency reward to one-turn trajectories and decrease as more reasoning iterations are used. As illustrated in Figure 28, the quadratic decay is the most conservative in the early and middle stages of a trajectory, while the logarithmic decay imposes the strongest penalty. For $x \in (0, 1)$, we have

$$R_{\text{eff}}^{\text{quad}}(\tau) > R_{\text{eff}}^{\text{lin}}(\tau) > R_{\text{eff}}^{\text{log}}(\tau), \tag{50}$$

indicating that the quadratic decay encourages efficiency without aggressively suppressing additional reasoning iterations.

Figure 29 compares the downstream performance of the three decay functions on MATH500, AIME24, and AIME25. The quadratic decay consistently yields the most stable training curve and the strongest final performance. In contrast, the linear and logarithmic variants can improve performance during the early and middle stages of training, but they become noticeably less stable in the later stage. In particular, the linear decay exhibits a sharp performance collapse near the end of training on all three benchmarks, and the logarithmic decay also shows clear degradation on AIME24 and AIME25. These results suggest that overly aggressive efficiency pressure may cause the policy to prefer shorter trajectories before sufficient reasoning has been completed.

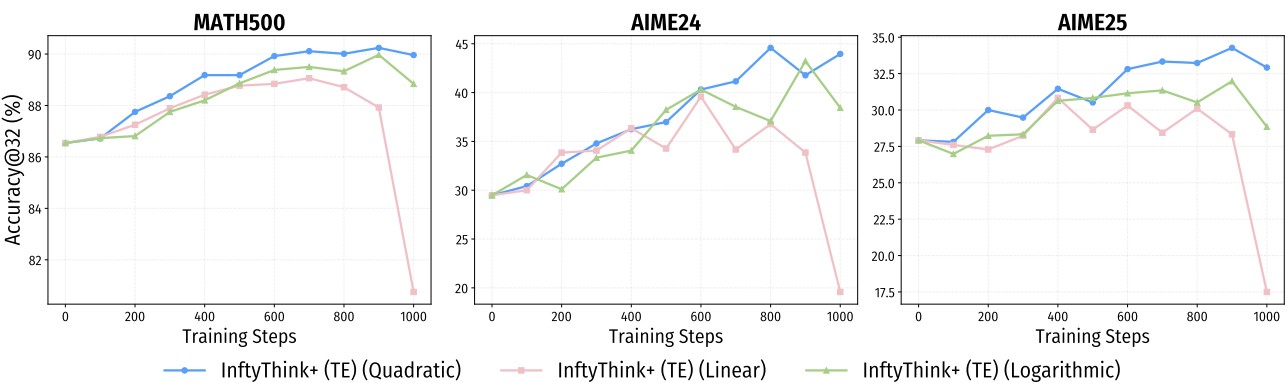

*Figure 29.* Benchmark performance of InftyThink+ trained with task reward and efficiency reward under quadratic-style decay, linear-style decay and logarithmic-style decay.

We further analyze the training dynamics in Figure 30. Under the linear and logarithmic decays, the average number of reasoning turns decreases much faster than under the quadratic decay. Although this leads to higher efficiency rewards, it is accompanied by a decline in task reward and benchmark accuracy in the later training stage. This indicates that the policy is partially optimizing the efficiency term by prematurely reducing the number of reasoning iterations, rather than by learning more effective reasoning strategies. By contrast, the quadratic decay reduces the number of turns more gradually while maintaining a higher and more stable task reward, leading to a better accuracy–efficiency trade-off.

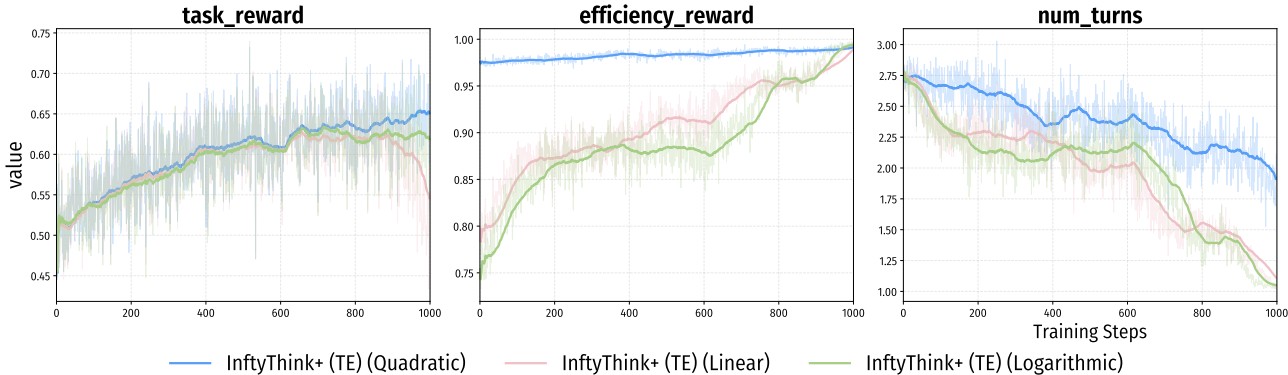

*Figure 30.* InftyThink-related metrics of InftyThink+ trained with task reward and efficiency reward under quadratic-style decay, linear-style decay and logarithmic-style decay.

Overall, this ablation shows that the efficiency reward should act as a soft regularizer rather than a dominant optimization target. A decay function that is too steep may bias the model toward premature termination and harm reasoning performance. The quadratic decay provides a more appropriate inductive bias: it mildly penalizes moderate additional reasoning turns, while still discouraging unnecessarily long trajectories. Therefore, we adopt the quadratic efficiency-reward decay in our main experiments.

## Q. Discussion: Comparison with Delethink.

Aghajohari et al. (2025) proposes a Markovian reasoning framework, Delethink, to address the inefficiency of long-chain reasoning in reinforcement learning-based large language models, where computation and memory grow rapidly with reasoning length. To resolve this, Delethink reformulate the reasoning process as a sequence of steps operating on a fixed-size state, decoupling reasoning depth from context length. Delethink introduce an RL environment that segments long reasoning into multiple short blocks, where the model periodically resets its context while preserving essential state information to maintain reasoning continuity. Under this formulation, the model learns to carry out extended reasoning across iterations.

### Q.1. Paradigm Design Comparison

We first analyze the differences between the InftyThink reasoning paradigm and the Delethink reasoning paradigm, followed by a discussion of their respective strengths and limitations. In the following, we provide an overview of the Delethink reasoning paradigm.

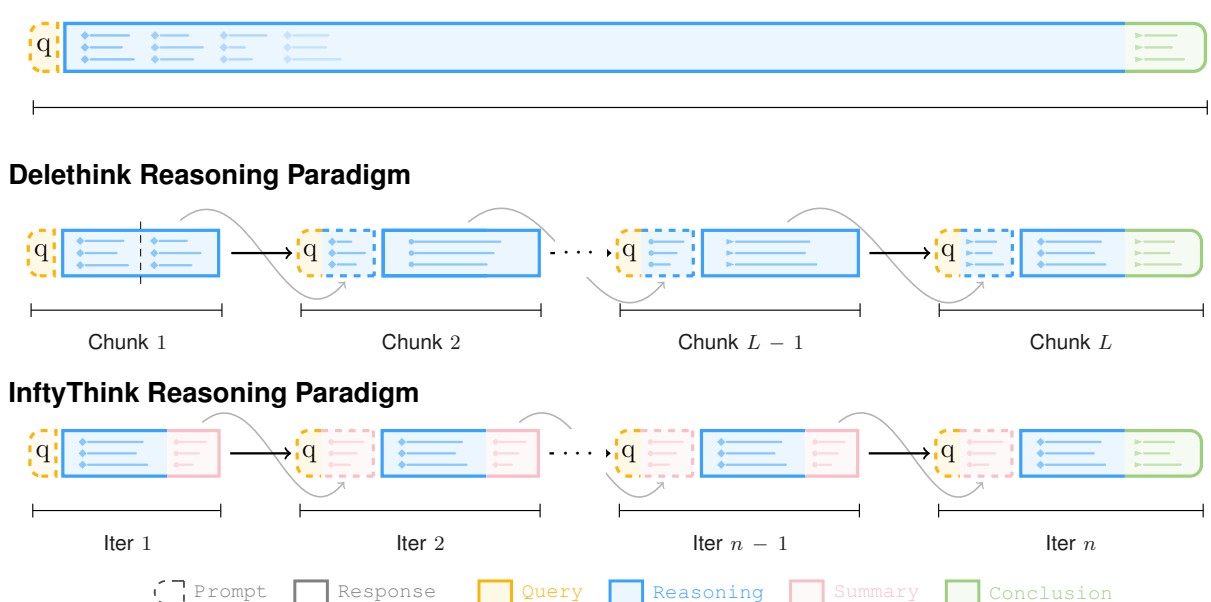

*Figure 31.* **Vanilla** reasoning paradigm VS. **Delethink** reasoning paradigm VS. **InftyThink** reasoning paradigm.

The primary distinction between InftyThink and Delethink lies in how intermediate information is carried across reasoning iterations. InftyThink explicitly generates a summary at each reasoning round; these summaries are represented as textual tokens and are injected as contextual inputs for the subsequent round. In contrast, Delethink does not produce explicit summaries. Instead, it directly reuses the trailing segment of reasoning tokens from the current iteration as the context for the next iteration.

Formally, DeleThink introduces three hyperparameters: the single-iteration reasoning budget $\mathcal{C}$, the number of tokens $m$ retained for context propagation, and the maximum number of reasoning iterations $\mathcal{I}$. For the first reasoning iteration ($i = 1$), the model takes the query $q$ as input and performs reasoning with the maximum generation length constrained to $\mathcal{C}$, producing the first reasoning segment $r_1$, which can be expressed as:

$$r_1 \sim \pi_\theta(\,\cdot \mid q;\, \texttt{max\_tokens} = \mathcal{C}). \tag{51}$$

For subsequent reasoning iterations, the model takes the last $m$ tokens from the previous round as its context and continues the reasoning process based on this truncated history, until it autonomously produces a *conclusion* and the $\texttt{eos}$ token.

$$r_i \sim \pi_\theta(\,\cdot \mid q, r_{i-1}[\text{-m:}];\, \texttt{max\_tokens} = \mathcal{C}). \tag{52}$$

We compare InftyThink and Delethink from the perspectives of state representation, cross-iteration information flow, and training–inference behavior. Each paradigm exhibits complementary strengths. Delethink offers several advantages primarily stemming from its minimalistic and implicit state transition design.

- **Delethink adopts a more Markovian formulation of the reasoning process.** By retaining only a short suffix of the reasoning tokens from the previous iteration as the context for the next one, each reasoning step depends on a compact, local state. This design better aligns with the Markov decision process assumption commonly used in reinforcement learning, which can facilitate more stable policy optimization and value estimation.

- **Delethink avoids the potential information bias introduced by explicit summaries.** In InftyThink, summaries are generated via lossy compression and may omit critical details or introduce abstraction noise. Delethink directly reuses raw reasoning tokens, thereby eliminating errors induced by imperfect summary generation and preventing the accumulation of summary-level distortions.

- **Delethink significantly simplifies the reasoning pipeline.** It removes the need for an auxiliary summarization module, as well as the associated prompt engineering, length constraints, and quality control mechanisms. This simplification reduces system complexity and lowers the engineering overhead in large-scale training and deployment. Besides, Delethink does not require the cold start phase.

- **Delethink preserves stronger local continuity in the reasoning trajectory.** By directly extending the tail of the previous reasoning, it maintains syntactic and semantic coherence at the token level, resulting in a more natural continuation of thought without abrupt abstraction shifts.

- **Delethink can reduce inference overhead in practice.** Since it *does not generate additional summary tokens*, it avoids extra generation steps, which may lower overall token usage and latency, especially in settings where summary generation requires retries or strict length control.

In contrast, InftyThink excels in scenarios that require explicit information structuring, long-range dependency preservation, and controllability.

- **InftyThink provides an explicit and interpretable mechanism for cross-iteration information transfer.** By generating a textual summary at the end of each reasoning iteration and using it as the primary context for the next step, InftyThink maintains a clear, high-level representation of the current progress. This explicit state is easier to analyze, debug, and regulate than implicitly truncated reasoning tokens.

- **InftyThink exhibits stronger long-range dependency retention.** Summaries serve as high-level abstractions that selectively preserve globally important information, mitigating the gradual information loss caused by repeatedly truncating token-level reasoning. This makes InftyThink more suitable for tasks requiring global consistency across many reasoning steps.

- **The summarization mechanism in InftyThink acts as an effective noise filter.** Raw reasoning traces often contain redundant explorations or incorrect intermediate branches. By compressing reasoning into summaries, InftyThink suppresses such noise and allows subsequent iterations to condition primarily on validated and task-relevant information, reducing error propagation.

- **InftyThink is particularly well-suited for complex and long-horizon reasoning tasks**, such as mathematical and logical reasoning. The summary at each iteration functions as a semantic milestone, helping the model maintain a clear notion of global objectives and progress rather than being overwhelmed by low-level token details.

- **Explicit summaries provide a natural interface for controllability and extensibility.** They enable the incorporation of length constraints, structured formats, external verification, and reward shaping in both SFT and RL settings, allowing fine-grained control over reasoning behavior during training.

- **The "reason–summarize–reason" loop in InftyThink closely mirrors human problem-solving strategies**, where intermediate conclusions are periodically abstracted before further reasoning. This facilitates reasoning at a higher semantic level rather than relying solely on local token continuation.

### Q.2. Experimental Comparison

Since our experiments adopt the same base model, training configuration, and dataset as Aghajohari et al. (2025), we can directly conduct an apples-to-apples comparison in the RL setting. DeleThink reports checkpoint-level evaluation results on AIME24 and AIME25; therefore, we align our evaluation protocol accordingly and compare the two methods under identical conditions. The resulting performance curves are summarized in Figure 32.

As shown in Figure 32, InftyThink$^+$ consistently outperforms DeleThink throughout training on both AIME24 and AIME25. On AIME24 (left), the *task-reward-only* variant achieves the strongest gains, reaching substantially higher accuracy than

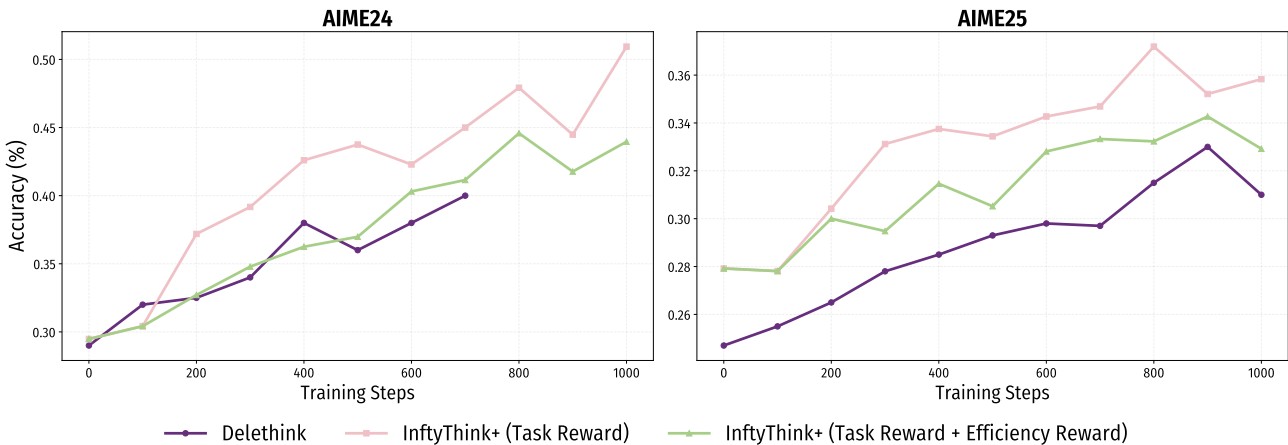

*Figure 32.* Checkpoint-level comparison with DeleThink on AIME24/25.We report accuracy (%) along RL training steps for DeleThink and our InftyThink$^+$ under matched base model, training configuration, and dataset. DeleThink results are copied from their paper and are available only up to 700 training steps on AIME24, hence the truncated curve in the left panel. InftyThink$^+$ is shown in two variants: *task reward only* and *task reward + efficiency reward*.

DeleThink at comparable steps, while the *task+efficiency* variant also remains above DeleThink despite being optimized with an additional efficiency objective. Notably, the DeleThink curve on AIME24 ends at 700 steps because their paper only reports checkpoint evaluations up to this point. On AIME25 (right), both InftyThink$^+$ variants yield clear improvements over DeleThink across the full training horizon, with *task reward only* again providing the best overall accuracy and *task+efficiency* tracking closely, indicating that incorporating efficiency optimization preserves most of the capability gains while improving runtime efficiency.

## R. Discussion: Why not Format Reward?

A natural question is whether we should explicitly introduce an additional *format reward* to encourage the model to follow the InftyThink protocol. In our framework, however, the reasoning process is *format-dependent* by design: each iteration relies on correctly parsing the model output into structured fields (e.g., `reasoning_content` and `summary`) to construct the next-round prompt. Once the format breaks, the parser fails, the iterative reasoning loop cannot proceed, and the rollout inevitably terminates without completing the task.

Consequently, *format compliance is already implicitly enforced by the task reward*. Any trajectory with malformed format is unable to reach a valid terminal answer and thus receives a low (often near-zero) task reward. In other words, the task reward naturally subsumes the supervision signal for format correctness, making a separate format reward largely redundant. Moreover, introducing an explicit format reward may skew optimization toward superficial template matching (over-emphasizing syntactic correctness) rather than improving the underlying reasoning quality, which is the primary goal of InftyThink+.

