# OpenReview forum: "InftyThink+: Effective and Efficient Infinite-Horizon Reasoning via Reinforcement Learning"
_ICML.cc/2026/Conference — ICML 2026 regular_

### Official Review · Reviewer_SfnC · 2026-03-07

**Soundness:** 2
**Presentation:** 3
**Significance:** 3
**Originality:** 1
**Overall Recommendation:** 4
**Confidence:** 4

**Summary:**

The paper extends InftyThink, which is an existing iterative reasoning method, to the RL setting. Through experiments on DeepSeek-R1-Distill-Qwen-1.5B and Qwen3-4B-Base, it shows that RL can learn when to summarize, what to preserve, and how to resume reasoning more effectively.

**Compliance With Llm Reviewing Policy:**

Affirmed.

**Final Justification:**

As the authors promise to revise the presentation to clearly state the research gap, `SFT-based InftyThink mainly learns the format of iterative reasoning, but not the decision policy`, I think this paper will better deliver some valuable empirical insights to the community due to a clear gap-solution-verification flow. So I raise my score **from 3 to 4**.


The methodological contribution is still a bit weak, because the RL algorithm is quite standard. However, it already meets the requirements for a conference paper from my view. That's the reason for a `weak accept` recommandation.

**Key Questions For Authors:**

## Writing
* line 144: I'm still confused on the differences between input-side and output-side context management, and why InftyThink belongs to the latter. Maybe can make it more clear.
* line 161: due to the SFT, "end-to-end RL optimization" should not be the case?
* line 227: $o_i \sim O_i$ is weird and informal, can just remove it.

## Evaluation
* Please carefully check whether it is `Qwen3-4B-Base` or `Qwen3-4B`. For the `Qwen3-4B-Base`, the vanilla model is with low accuracy and short CoTs, which is not the case in Table 1.

**Limitations:**

As stated above, I personally like this topic. But the presented methods are overly naive, lacking essential novelty and technical depth for an academic conference paper.

Additionally, the evaluation is barely satisfactory, also lacking of technical depth. The evaluated models are small dense models. The analysis is superfacial. It is good to show the `scaling` properties, in both token budget and model size, because the key words are test-time `scaling`. It is not insightful to show RL can learn when/how to compress and how to continue reasoning, because they are greatly expected with RL, especially when the intial model is based on SFT cold start.

So I personally prefer to reject this paper, despite the significance of this topic. Either improving the the technical depth or delivering more empicial insights (better do both) can change this decision.

**Strengths And Weaknesses:**

## Strengths
* The paper is well presented.
* I personally like this topic, which has great potential to enable more efficient reasoning.
* The results are good.

## Weakness
* The paper is more like a tech report. To be an academic conference paper, it kind of lacks some novelty. It is very straightforward to apply RL on InftyThink, and the results are greatly within expectations. The techniques beyond InftyThink is trivial and quite standard.
* It also lacks technical depth, especially for the reasoning efficiency part. [1] has shown that it is possible to improve both accuracy and efficiency compared to vanilla GRPO. However, the proposed efficiency reward is overly naive and superfacial, suffering great performance degradation.



[1] https://arxiv.org/pdf/2505.13438

---

> ### Author Rebuttal · Authors · 2026-03-31
>
> Dear Reviewer SfnC,
>
> Thank you for the effort and time you dedicated to reviewing our paper, and for your recognition of our motivation and contribution. We have greatly benefited from your valuable feedback. Below are our responses to the weaknesses you pointed out.
>
> ---
> > **[W1] About academic novelty.**
>
> Thank you for the comment. We believe there is a misunderstanding regarding our motivation and contribution.
>
> - **Not "applying RL to InftyThink".**: Our starting point is an empirical gap: **SFT-based InftyThink mainly learns the *format* of iterative reasoning, but not the *decision policy*** (i.e., *when to compress, how to compress, and how to continue*).  Our goal is to **optimize the reasoning paradigm itself**, with RL serving as a *tool* rather than the contribution.
>
> - **Novel problem setting.**: We explicitly study **decision-making over intermediate reasoning states in iterative reasoning**, which, to our knowledge, has not been systematically addressed in prior work. Demonstrating that RL can **actually realize** these decisions, rather than merely being **expected to help**, is itself non-trivial and empirically validated in our paper.
>
> - **Non-trivial methodological adaptations.**: Applying GRPO to InftyThink is **not straightforward**, due to the multi-round, compress-then-continue structure. We have designed **trajectory-level rollout, trajectory-aware optimization and round-level efficiency modeling** tailor to the RL optimization of InftyThink-style models. For more details, please refer to `[Q1@HRS2]`.
>
> In summary, our contribution is **not a direct RL application**, but identifying a missing capability in iterative reasoning and **designing an end-to-end framework to learn it**, which we believe constitutes meaningful novelty beyond standard RL usage.
>
> > **[W2] Technical depth for the reasoning efficiency part.**
>
> Thank you for the insightful question. We would like to clarify that our goal is not to demonstrate that *it is possible* but *how to better* improving both accuracy and efficiency.
>
> Under the same training data and base model, InftyThink+ achieves substantially better performance than the results reported in [1] (e.g., AIME24: 50.94 vs. ~35.00). We also provide additional comparisons with other efficient reasoning RL methods in https://anonymous.4open.science/r/InftyThink-Plus-Rebuttal-AECD/C.pdf, further demonstrating the superiority of InftyThink+ in improving both reasoning performance and efficiency.
>
> In addition, we would like to clarify that the observed degradation from incorporating the efficiency reward is measured relative to the task-reward-only setting. Compared with vanilla RL, w/ efficiency reward still achieves simultaneous improvements in both performance and efficiency. Please refer to `[Q1@fUNs]` for a more detailed discussion.
>
> Regarding the efficiency reward, we conducted additional ablations during the rebuttal. Specifically, we compare with linear and logarithmic decay variants. The results show that **our quadratic decay consistently yields more stable training dynamics and better final performance**. We further analyze the underlying reasons behind these observations; please refer to `[Q2@RZoM]` for detailed discussions.
>
> > **[Q1] Differences between input-side and output-side context management.**
>
> Thank you for pointing this out. In our view, **input-side context management focuses on compressing or filtering already completed historical messages** before they are fed into the model, whereas **output-side context management operates on the ongoing generation process**, dynamically managing how the current reasoning trajectory is produced.
>
> > **[Q2] About "end-to-end RL optimization" claim.**
>
> To clarify, our statement in Line 161 is that *InftyThink+ introduces an end-to-end RL optimization*, not that *InftyThink+ is synonymous with end-to-end RL optimization*, we can view InftyThink+ as a *SFT + end-to-end RL* framework. Furthermore, We refer to it as end-to-end because the complete iterative reasoning trajectory is optimized in a unified manner, rather than splitting summarization, reasoning continuation, and intermediate decision making into independently trained components. Therefore, we believe the end-to-end claim is well justified.
>
> > **[Q3] Equation writing.**
>
> We will clarify and present it more formally in the revision.
>
> > **[Q4] Evaluation clarification.**
>
> Our base model is Qwen3-4B-Base, i.e., the pretrained model without alignment. We want to note that the vanilla ($\times$) results in Table 1 do not correspond to the raw base model. Instead, they refer to the model after the cold-start SFT stage on top of the base model, without the subsequent RL stage. We have clarified this point in the caption of Table 1.
>
> ---
> Thank you again for the thorough review. We think that our rebuttal can address your main concern, if you have any further questions, we would be happy to answer them.
>
> Sincerely,
> InftyThink+ Authors

---

> > ### Author Rebuttal · Reviewer_SfnC · 2026-04-01
> >
> > The major concern still persists: as also mentioned by Reviewer `RZoM`, the methodological contribution beyond InftyThink is incremental and trivial. Although the authors claim applying GRPO to InftyThink is non-trivial and not straightforward, I respectively disagree, as pointed by Reviewer `RZoM`, this is a quite standard application of multi-turn RL.
> >
> > > Our starting point is an empirical gap: SFT-based InftyThink mainly learns the format of iterative reasoning, but not the decision policy
> >
> > I don't think this is how the current paper is presented. I list some evidences below.
> > - Introduction: `InftyThink (Yan et al., 2025) allows models to autonomously decide when to summarize, but relies exclusively on supervised fine-tuning: the model learns to format iterative reasoning by imitating training data. This analysis reveals a key insight: what makes iterative reasoning effective is not the format itself, but the ability to make optimal decisions at each iteration`
> > - Contribution: `We introduce reinforcement learning into the iterative reasoning paradigm, enabling end-to-end optimization of when to summarize, what to preserve, and how to continue across iterations.`
> > - The flow of Methods: `3.1. InftyThink Reasoning Paradigm` -> `3.2. Cold Start` -> `3.3. Reinforcement Learning`.
> >
> > **There is no explicit evidence in the main text to support the claim of `Our starting point is an empirical gap`. At least one separate section or subsection is expected to clearly state the research gap.**
> >
> > To be honest, if the paper had really been presented in this way, explicitly identifying the current research gap **with clear evidence and solid empirical study**, and how it is resolved via RL, I would definitely give a higher score because it delivers clear insights to the community. So far, this work feels like: completing some unfinished work of InftyThink (without solid methodology contribution and too much empicial insights). This is awkward, as most of the novelty and contribution have already been taken by the prior work. As I said in the comment I personally like this topic, if this paper were together with InftyThink, I would give a 6.
> >
> > >  we compare with linear and logarithmic decay variants
> >
> > I appreciate these new experiments. These results partially resolve my concern because the explainations make sense to me. But still have concerns on the technical depth because reward shaping is trivial (can never guarantee an optimal reward shaping due to huge space) and the baseline is naive.
> >
> >
> > ------------------------------
> >
> > **To AC**:
> >
> > From my view:
> > - In summary, the topic is significant, and the empirical results are impressing. However, the methodology contribution is incremental, because most of novelty and contribution have been taken by the prior work InftyThink.
> > - I'm okay for both accept and reject.
> > - If the authors could provide **clear evidence and solid empirical study** to support the claimed starting point (or research gap) `SFT-based InftyThink mainly learns the format of iterative reasoning, but not the decision policy` and change the presentation accordingly, I would raise my score to 4.

---

> > > ### Author Response · Authors · 2026-04-04
> > >
> > > Dear Reviewer SfnC,
> > >
> > > Thank you for your insightful feedback. We strongly agree with your opinion and appreciate the opportunity to clarify our empirical findings.
> > >
> > > ---
> > >
> > > > **On the empirical gap of SFT-based training.**
> > >
> > > We provide three pieces of empirical evidence supporting our claim:
> > >
> > > 1. **Suboptimal decision of *when to summarize*:** SFT-based models exhibit clear room for improvement in deciding *when* to perform summarization. We replace the model's autonomous decisions with two rule-based strategies: *Fixed* (summarizing at predetermined token intervals) and *Random* (summarizing at randomly sampled positions). We find that SFT-based models fail to outperform these simple baselines (48.55% vs. 48.53% vs. 48.34%), indicating that they do not learn an effective policy for timing summarization to achieve better information compression.
> > >
> > >     | Strategy | MATH500 | AIME24 | AIME25 | AMC23 | GPQA_diamond | MathOdyssey | Average|
> > >     | -|-|-|-|-|-|-|-|
> > >     | SFT-based InftyThink| 86.54	|29.48	|27.92	|71.64|	14.17	|61.56|	48.55|
> > >     | Random | 86.68	| 28.54| 	26.25| 	72.58| 	14.23	| 62.91	| 48.53|
> > >     | Fixed | 86.91	|28.44	|26.04|	72.03	|14.4	|62.21	|48.34|
> > >
> > > 2. **Limited *how to summarize* capability.** We further evaluate the quality of summaries by replacing the model-generated summaries with those produced by a strong general-purpose LLM. This substitution consistently improves downstream performance (average 48.55% → 49.73%), suggesting that SFT-based models are weak at extracting salient information from intermediate reasoning steps. This limitation directly constrains the effectiveness of iterative reasoning.
> > >
> > >     | Strategy | MATH500 | AIME24 | AIME25 | AMC23 | GPQA_diamond | MathOdyssey | Average|
> > >     | -|-|-|-|-|-|-|-|
> > >     | SFT-based InftyThink | 86.54| 	29.48| 	27.92	| 71.64	| 14.17	| 61.56	|48.55|
> > >     | External Summazier |  87.38| 	32.4| 28.75	| 73.75	| 13.54	| 62.58	| 49.73 |
> > >
> > > 3. **Insufficient *how to continue reasoning*.** We also assess continuation ability by feeding summaries generated by SFT models into the *pre-SFT base model* for subsequent reasoning. Surprisingly, the base model matchces or outperforms the SFT model under this setup. This indicates that SFT primarily teaches the *format* of continuation rather than enabling the model to internalize effective continuation strategies.
> > >
> > >     | Strategy | MATH500 | AIME24 | AIME25 | AMC23 | GPQA_diamond | MathOdyssey | Average|
> > >     | -|-|-|-|-|-|-|-|
> > >     | SFT-based InftyThink | 86.54| 	29.48| 	27.92	| 71.64	| 14.17	| 61.56	|48.55|
> > >     |continue@iter1	|85.14	|26.46	|24.58	|69.37	|12.29	|58.79	|46.10|
> > >     |continue@iter2	|86.25	|28.02	|26.56	|69.22	|13.02	|60.09	|47.19|
> > >     |continue@iter3	|86.88	|28.12	|26.25	|72.73	|13.53	|61.47	|48.16|
> > >     |continue@iter4	|87.07	|30.10	|26.56	|72.81	|14.02	|62.04	|48.76|
> > >
> > > **Conclusion.**
> > > Based on these observations, we conclude that SFT-based methods exhibit a clear *empirical gap* when applied to the InftyThink paradigm. Specifically, SFT enables models to imitate the *format* of iterative reasoning, but does not improve their *capability* to reason effectively under this paradigm. **These findings are systematically analyzed in our ablation study (Section 5.1), where we further show that RL substantially improves all three aspects.**
> > >
> > > **Planned revision.**
> > > We fully agree that our current presentation does not sufficiently emphasize this point. At present, we only briefly state in the introduction (Lines 74-85) that *the model learns to format iterative reasoning by imitating training data*, without providing a detailed discussion. In the revision, we will strengthen this aspect by explicitly highlighting the above empirical evidence. In particular, **we will introduce a new subsection between Section 3.1 and Section 3.2 to systematically analyze the limitations of SFT-based methods**, which will serve as a clearer motivation for our work.
> > >
> > > > **About the design of efficiency reward.**
> > >
> > > We agree with your point. We are happy that the current efficiency reward design effectively improves the reasoning efficiency of the InftyThink paradigm, so we have included it in the main body of the paper. We will further discuss the limitations of the efficiency reward in the limitations section. Thank you for this valuable suggestion.
> > >
> > > ---
> > >
> > > We sincerely thank you for this valuable suggestion, which will significantly improve the clarity and strength of our paper. We hope that our response addresses your concerns and may encourage you to reconsider the evaluation of our work.
> > >
> > > We also greatly appreciate the time and effort you have devoted to reviewing our paper.
> > >
> > > Sincerely,
> > > InftyThink+ Authors

---

### Official Review · Reviewer_fUNs · 2026-03-11

**Soundness:** 3
**Presentation:** 3
**Significance:** 2
**Originality:** 3
**Overall Recommendation:** 4
**Confidence:** 4

**Summary:**

This paper proposes InftyThink+, a reinforcement learning (RL) framework for training iterative reasoning LLMs.
Built upon InftyThink, this method shifts iterative reasoning training from SFT-only to end-to-end RL training.

The method uses a two-stage pipeline: (1) cold-start SFT followed by (2) trajectory-level RL with GRPO, using a composite reward that combines task correctness $\mathcal{R}_\text{task}$ and an efficiency penalty $\mathcal{R}_\text{eff}$.
Experiments on 1.5B and 4B models show gains on math benchmarks (MATH500, AIME24/25, GPQA diamond) and lower inference latency.

**Compliance With Llm Reviewing Policy:**

Affirmed.

**Final Justification:**

My major concerns are not fully addressed by the reviewers. I would like to adjust the score if the authors provide additional explanation or empirical evidence.

**Key Questions For Authors:**

Please refere to the "Summary of Weaknesses" section

**Limitations:**

yes

**Strengths And Weaknesses:**

## Summary of Strengths
- **End-to-end RL for iterative reasoning.** The paper tackles a key challenge in long-horizon reasoning: training the model to decide iteration depth by itself. Using RL to minimize the "turns of logic changes" feels more reasonable than methods that only optimize the total number of reasoning tokens.
- **Practical efficiency gains.** The reported 32.8% latency reduction on AIME25 and 18.2% training speedup are concrete. The ~5% average improvement (on the 4B variant) is also valuable.

## Summary of Weaknesses
The weaknesses are ranked by priority (high to low). I'm happy to raise my score if the authors provide satisfying responses.
1. **The efficiency reward hurts accuracy non-negligibly.** The main results table shows that InftyThink+ with the combined Task+Efficiency (T+E) reward causes a substantial accuracy drop — e.g., AIME24 falls from 50.94 to 43.96 on the 1.5B model, and there is about a 2–3 point average drop on both 1.5B/4B models. Do we really want to sacrifice such the performance gains for around 10% latency reduction?

2. **Weak baselines and limited scale.** Experiments are only on 1.5B and 4B models. At this scale, variance can be high and results may not generalize. It would also help to compare against stronger iterative reasoning baselines, instead of only a basic "cold start only" model.

### Minor Weaknesses
3. The paper uses "InftyThink+" as a name, but its relationship to the original InftyThink is not always clearly stated. I understand the value of InftyThink paradigm (Figure 5), but why is there no direct comparison with InftyThink? Is InftyThink equivalent to the "cold start only" InftyThink+ variant?
4. AIME24/25 each have only 30 problems. The reported scores should include confidence intervals or statistical tests.

---

> ### Author Rebuttal · Authors · 2026-03-30
>
> Dear Reviewer fUNs,
>
> Thank you for the effort and time you dedicated to reviewing our paper, and for your recognition of our motivation and contribution. We have greatly benefited from your valuable feedback. Below are our responses to the weaknesses you pointed out.
>
> ---
> > **[W1] The efficiency reward hurts accuracy non-negligibly.**
>
> Thank you for the insightful question. We believe there may be a gap in understanding our main experimental results in Table 1.
>
> * **Method design.** The efficiency reward is optional. Even with task reward alone, InftyThink+ already outperforms vanilla RL in both performance (53.96 vs. 47.31) and efficiency (100.21 vs. 149.44). The efficiency reward is introduced only to further reduce latency, with a possible accuracy trade-off.
>
> * **Empirical results.** We agree that the accuracy drop is real, but it is relative to the *task-reward-only* setting, not vanilla RL. **On the 1.5B model, InftyThink+ with efficiency reward still outperforms vanilla RL by about +3 pp in accuracy**. Also, the stated "10% latency reduction" is inaccurate: InftyThink+ TE reduces latency by about **50%** vs. task-reward-only InftyThink+ (48.37 vs. 100.21), and about **67%** vs. vanilla RL (48.37 vs. 149.44).
>
> > **[W2] Weak baselines and limited scale.**
>
> Thank you for the question. We highlight three additional experiments that more clearly demonstrate the effectiveness of InftyThink+.
> + **Scaling to a larger model.** During the rebuttal period, we additionally experimented with a stronger MoE base model, Qwen3-30B-A3B-Base (https://anonymous.4open.science/r/InftyThink-Plus-Rebuttal-AECD/B.pdf). We observed gains that are consistent with those reported in the main paper. Due to time and resource constraints, however, we only trained this model for 400 RL steps.
> + **Comparison with other iterative reasoning baselines.** In Appendix P, we already include a comparison with DeleThink, a strong RL-based iterative reasoning training framework (https://openreview.net/pdf?id=tyul8kXaJU#page=45). Our method substantially outperforms the performance reported for DeleThink in its paper.
> + **Comparison with other efficient reasoning methods.** We further added experimental comparisons against a range of efficient reasoning methods (https://anonymous.4open.science/r/InftyThink-Plus-Rebuttal-AECD/C.pdf). Under the same base model, InftyThink+ consistently achieves clearly stronger performance than these alternative efficient reasoning approaches.
>
> > **[W3] Relationship of InftyThink and InftyThink+.**
>
> We apologize for not clearly explaining the relationship between InftyThink+ and InftyThink.
> + **Theoretically**, InftyThink [1] introduces the iterative reasoning paradigm that breaks a long reasoning process into multiple short stages connected by self-generated summaries. However, [1] mainly relies on SFT to teach this format. InftyThink+ instead is an end-to-end training framework that further optimizes this paradigm, with both a cold-start stage and an RL stage to improve when to compress, how to compress, and how to continue reasoning.
> + **Empirically**, as you noted, [1] can be viewed as a cold-start-only variant of InftyThink+. Our implementation also includes several refinements over [1], such as a different summarizer and a new hyperparameter $\gamma$. Moreover, because our evaluation also considers latency, directly comparing with the numbers reported in [1] would not be fully fair. We therefore re-trained the SFT-only version ourselves; in Table 1, the rows with RL marked as $\times$ correspond to this setting, which can be regarded as our reproduction of [1]. As shown in Table 1, adding the RL stage brings a 21% accuracy gain on AIME24 over this SFT-only variant.
>
> Thanks for raising this point. In the revision, we will clarify this relationship and add an appendix discussion comparing our reproduced SFT-only results with those reported in [1].
>
> [1] Yan, et al. InftyThink: Breaking the Length Limits of Long-Context Reasoning in Large Language Models.
>
> > **[W4] Evaluation stability.**
>
> Thanks for the question. We agree that the limited size of AIME24/25 can lead to relatively high evaluation variance. To mitigate this issue, we follow the common practice adopted in prior work by **sampling 32 completions for each query and reporting averaged metrics, thereby reducing evaluation noise**.
> In Appendix J.2, we further analyze the evaluation variance and find that **the accuracy fluctuation on AIME is only around 1-2%, which is substantially smaller than the performance gains achieved by InftyThink+**, supporting the effectiveness of our method. In the revision, we are also adding more evaluations and will report confidence intervals for the main results in Table 1.
>
> ---
> Thank you again for the thorough review. We think that our rebuttal can address your main concern, if you have any further questions, we would be happy to answer them.
>
> Sincerely,
> InftyThink+ Authors

---

> > ### Author Rebuttal · Reviewer_fUNs · 2026-04-03
> >
> > Thank you for the rebuttal, which addresses some of my questions. However, I still feel that several aspects have not been clearly analyzed or compared — in particular, the core distinction between this work and InftyThink/DeleThink.
> >
> > > [1] mainly relies on SFT to teach this format. InftyThink+ instead is an end-to-end training framework that further optimizes this paradigm, with both a cold-start stage and an RL stage to improve when to compress, how to compress, and how to continue reasoning.
> >
> > I acknowledge the effectiveness of the iterative reasoning paradigm, but if the contribution is primarily designing corresponding rewards and adapting the GRPO optimization algorithm, is this sufficient to constitute a standalone technical contribution? The authors should provide a more systematic discussion demonstrating: (1) what unique improvements InftyThink+ introduces over prior iterative reasoning methods; and (2) what effects these improvements bring.
> >
> > If we are simply applying RL optimization with corresponding rewards to improve performance and reduce token consumption, the results seem expected and may not warrant a separate paper.
> >
> > > Moreover, because our evaluation also considers latency, directly comparing with the numbers reported in [1] would not be fully fair.
> >
> > I do not understand why this comparison would be unfair. I remain confused about the key benefits of InftyThink+ — is it supposed to enhance capability, or reduce latency? I still do not see why we cannot use an existing published method as a baseline for direct comparison.
> >
> > I would like to hold my score for now, and look forward to seeing the author's reply.
> >
> > P.S. If certain comparisons or discussions are important enough (directly comparison against baselines), I believe they should be placed in the main body of the paper, such as reported in Table 1 or main-body figures. This is how the 8-page constraint is supposed to work.

---

> > > ### Author Response · Authors · 2026-04-04
> > >
> > > Dear Reviewer fUNs,
> > >
> > > We sincerely thank you for your response. We are pleased that our rebuttal has addressed part of your concerns, and we would like to take this opportunity to further clarify several aspects that you highlighted.
> > >
> > > ---
> > > > **Clarification of the methodology of InftyThink+**
> > >
> > > We would like to further clarify our motivation. **The starting point of this work is our empirical analysis of the limitations of existing iterative reasoning methods.** Specifically, we find that current SFT-based InftyThink mainly learns to imitate the *format* of iterative reasoning, rather than internalizing the underlying capability. In this sense, RL is not the motivation of our work itself, but the means we use to improve model performance under the InftyThink reasoning paradigm. We refer the reviewer to our reply to Reviewer `SfnC` for a more detailed discussion of the limitations of current InftyThink-style methods.
> > >
> > > **Methodological Clarification**
> > >
> > > Regarding the methodology, we would also like to emphasize that **InftyThink+ is not obtained by simply porting multi-turn GRPO to an InftyThink-style model.** The InftyThink reasoning paradigm does not correspond to a standard multi-turn rollout, but rather to a *multi-sequence rollout*. In conventional multi-turn RL, one rollout produces a single sequence consisting of multiple messages. In contrast, due to the context-refresh mechanism in InftyThink, a single rollout produces multiple sequences. To the best of our knowledge, our work is the first to apply RL to optimize this paradigm. For a more detailed discussion of the methodological novelty, please refer to our response to Reviewer `RZoM (Q1)`.
> > >
> > > **Responses to the raised two questions.**
> > > 1. *What unique improvements does InftyThink+ introduce over prior iterative reasoning methods?*
> > >     * **Compared with InftyThink**, we introduce RL-based optimization, which enables the model to internalize experience and improve its capability through large-scale rollout training.
> > >     * **Compared with DeleThink**, we introduce an explicit summarization mechanism. DeleThink simply uses the last $M$ tokens as the representation of historical information, whereas our method allows the model to generate summaries autonomously, leading to more effective information transfer across multiple rounds of iterative reasoning.
> > > 2. *What effects do these improvements bring?*
> > >    * **Compared with InftyThink**, our analysis shows that InftyThink+ substantially improves the model's ability in *when to compress*, *how to compress*, and *how to continue reasoning* (see Section 5.1).
> > >    * **Compared with DeleThink**, both our methodological design and empirical results demonstrate clear advantages. By introducing the summarization mechanism, InftyThink+ achieves stronger final performance. Detailed analysis can be found in Appendix P (P.1 for conceptual and P.2 for empirical).
> > >
> > > We suspect that our current presentation may not have communicated these points as clearly as intended, and we will revise the paper to make our motivation, methodology, and empirical findings clearer.
> > >
> > > > **Experimental comparison.**
> > >
> > > Thank you for following up on this issue. We would like to clarify it from two perspectives.
> > > + **Comparison against InftyThink as the baseline.** As we already explained in the rebuttal, the baseline reported in Table 1 is effectively our reproduction of the method in [1]. In other words, **we do compare against this method in the main body**; the comparison is conducted under exactly the same training and evaluation settings, which ensures a fair and controlled assessment.
> > > * **Why directly comparing the reported latency is not fair.** The experiments in [1] were conducted on NVIDIA A100 GPUs for both training and inference, whereas our experiments use NVIDIA H200 GPUs. Since inference latency is highly sensitive to the underlying hardware, we believe that directly comparing the reported latency numbers across these two settings would be unfair.
> > >
> > > **Response to the P.S.** Our point was not that the comparison will be moved to the appendix. Rather, as noted above, the comparison has already been included in Table 1. The additional appendix results are only intended to help readers better understand the rationale behind this presentation.
> > >
> > > ---
> > > We sincerely thank you for following up on these issues, which have been highly valuable in helping us further improve the paper. We hope that our responses have adequately addressed your remaining concerns, and would be grateful if you could reconsider the score.
> > >
> > > > **[Update]** We would also like to note that Reviewer `SfnC` and Reviewer `RZoM` raised a similar major concern about the methodology novelty. After reading our new experiments and analyses, they **considered the work to meet the requirements for a conference paper and increased the score**.
> > >
> > > Once again, we greatly appreciate the time and effort you have devoted to the review process.
> > >
> > > Sincerely,
> > > InftyThink+ Authors

---

### Official Review · Reviewer_RZoM · 2026-03-11

**Soundness:** 3
**Presentation:** 3
**Significance:** 1
**Originality:** 2
**Overall Recommendation:** 4
**Confidence:** 4

**Summary:**

This paper proposes InftyThink+, an end-to-end reinforcement learning (RL) framework designed to optimize iterative reasoning trajectories in large reasoning models. The method achieves significant gains in accuracy (e.g., +21% on AIME24) while simultaneously reducing inference latency by applying efficiency reward in RL training.

**Compliance With Llm Reviewing Policy:**

Affirmed.

**Final Justification:**

After the rebuttal and discussion, I decided to increase my score to 4. I still have some reservations about novelty, since I view the core paradigm as having already been introduced by the prior SFT work, and this paper mainly demonstrates that RL can further improve that framework. So I remain unconvinced that the methodological novelty is particularly strong.

However, I do think the work is solid overall. The paper is carefully done, the added experiments are useful, and the rebuttal made the contribution more convincing in practice. Even though I still see novelty as the main weakness, I now think the work is sufficiently solid to be accepted as a conference paper.

**Key Questions For Authors:**

1. Could the authors explicitly clarify the fundamental methodological novelty of InftyThink+ beyond the direct application of standard GRPO to the existing InftyThink pipeline?

2. Could the authors provide deeper theoretical insights or ablation evidence to justify why this specific formulation is superior to other potential reward shaping methods (e.g., linear or exponential penalties) for optimizing intermediate reasoning states?

3. The current evaluation primarily compares InftyThink+ against the Vanilla long-CoT paradigm and InftyThink. To better demonstrate the method's competitive advantage, could the authors include comparisons with other established LLM thinking or reasoning methods？

**Limitations:**

Yes

**Strengths And Weaknesses:**

**Strengths**:

1. The paper demonstrates solid empirical improvements over the SFT-only baseline (InftfThink), particularly on challenging mathematical reasoning tasks like AIME24 and AIME25.

2. The authors provide extensive analyses, including the necessity of the cold-start phase, the impact of the efficiency reward, and the effect of the iteration cap hyperparameter $\varphi$.

**Weaknesses**:

1. The core iterative reasoning paradigm (segmentation, summarization, and continuation) is heavily inherited from the prior work, InftyThink. The main difference of this paper is replacing the SFT with GRPO. Given that applying RL to reasoning models has become a standard practice in the community, the methodological contribution here appears incremental.

2. The proposed trajectory-level rollout and shared advantage mechanisms are essentially standard applications of multi-turn RL. The reward design (binary task reward multiplied by a heuristic quadratic efficiency penalty) lacks deeper theoretical insights.

---

> ### Author Rebuttal · Authors · 2026-03-31
>
> Dear Reviewer RZoM,
>
> Thank you for the effort and time you dedicated to reviewing our paper, and for your recognition of our motivation and contribution. We have greatly benefited from your valuable feedback. Below are our responses to the weaknesses you pointed out.
>
> ---
> > **[W1&W2&Q1] Methodological novelty of InftyThink+.**
>
> Thank you for the question. We agree that InftyThink provides the iterative reasoning paradigm underlying our work. However, the motivation of InftyThink+ is not simply to replace SFT with RL. Rather, our starting point is the observation that **SFT mainly transfers the iterative reasoning format, but does not sufficiently optimize the model’s actual decision-making ability within this paradigm.** Based on this observation, we introduce RL as a means to optimize these abilities end-to-end.
>
> Concretely, InftyThink+ introduces several targeted adaptations beyond vanilla GRPO:
> * **Trajectory-level rollout.** In standard GRPO, each sampled output is a single token sequence. In InftyThink+, each sampled trajectory may contain multiple reasoning rounds with repeated compression and continuation. We therefore redesign rollout to support variable-length, multi-round iterative trajectories.
> * **Trajectory-aware optimization.** Since one trajectory consists of multiple generated sequences, optimizing each sequence independently would create imbalance across samples with different numbers of rounds. We thus use a trajectory-level averaging scheme for policy optimization.
> * **Round-level efficiency modeling.** Beyond conventional token-length penalties, we explicitly model the cost of excessive reasoning rounds, which is specific to iterative reasoning and, to our knowledge, is not considered in prior efficient reasoning RL methods.
>
> Therefore, we view InftyThink+ not as a direct application of standard GRPO to InftyThink, but as **a targeted RL framework for optimizing when to compress, how to compress, and how to continue reasoning** in iterative reasoning.
>
> > **[Q2] Reward shaping ablations and analyses.**
>
> Thank you for this insightful question. During the rebuttal period, we further ablated the decay form of the efficiency reward. Besides the **quadratic-style decay** used in our paper, we also considered **linear** and **logarithmic-style** variants. The reward curves over the number of reasoning rounds within each trajectory, together with the corresponding training dynamics and performance results, are provided in (https://anonymous.4open.science/r/InftyThink-Plus-Rebuttal-AECD/A.pdf).
>
> Our main observations are:
> * **Better task performance.** Quadratic-style decay consistently achieves better performance than both linear and logarithmic variants on MATH500, AIME24, and AIME25.
> * **More stable training dynamics.** Under linear and logarithmic decay, training becomes unstable in later stages: task reward stagnates or even collapses, while the policy increasingly exploits the efficiency reward, which suppresses further gains in problem-solving ability. In contrast, quadratic decay better balances efficiency and task objectives, leading to more stable optimization.
>
> The advantage of quadratic-style decay lies in its **non-uniform penalization profile**: it decays more slowly when the iteration count is small, but much more sharply once the trajectory becomes long. This gives two benefits:
> * **Preserving early exploration.** It imposes a weaker penalty in early iterations, allowing sufficient reasoning depth and avoiding premature truncation of useful reasoning.
> * **Discouraging overly long trajectories.** It imposes a much stronger penalty on excessive iterations, suppressing unnecessarily prolonged reasoning.
>
> By contrast, linear and logarithmic decays penalize relatively more in early iterations but less in later ones, making optimization less stable and more likely to trade task performance for efficiency.
>
> > **[Q3] Broader evaluation baselines.**
>
> Thank you for the question. We highlight two additional experiments that more clearly demonstrate the  effectiveness of InftyThink+.
> + **Comparison with other iterative reasoning baselines.** In Appendix P, we already include a comparison with DeleThink, a strong RL-based iterative reasoning training framework (https://openreview.net/pdf?id=tyul8kXaJU#page=45). Our method substantially outperforms the performance reported for DeleThink in its paper.
> + **Comparison with other efficient reasoning methods.** We further added experimental comparisons against a range of efficient reasoning methods (https://anonymous.4open.science/r/InftyThink-Plus-Rebuttal-AECD/C.pdf). Under the same base model, InftyThink+ consistently achieves clearly stronger performance than these alternative efficient reasoning approaches.
> ---
> Thank you again for the thorough review. We think that our rebuttal can address your main concern, if you have any further questions, we would be happy to answer them.
>
> Sincerely,
> InftyThink+ Authors

---

> > ### Author Rebuttal · Reviewer_RZoM · 2026-04-02
> >
> > The rebuttal improves some empirical aspects, especially the broader baseline coverage, but my main concern remains unresolved because it targets the core novelty of the work. The paper’s main claim is that RL provides more than format imitation and enables better strategic iterative reasoning decisions.
> >
> > However, the rebuttal mainly offers implementation details and additional ablations, without clearly demonstrating a substantive methodological advance beyond applying standard RL to the existing InftyThink framework. The RL component appears closer to a direct adaptation of existing trajectory-level multi-turn RL formulations, rather than a clearly new RL methodology. Many similar patterns already appear in agent-RL work. Also, I remain unconvinced that the reward design is novel. Comparing it with a few alternative formulations and showing that it works better empirically is useful, but it is still a post-hoc preference rather than a principled justification.
> >
> > For these reasons, I remain at my original score.

---

> > > ### Author Response · Authors · 2026-04-04
> > >
> > > Dear Reviewer RZoM,
> > >
> > > We sincerely thank you for your response. We would like to take this opportunity to provide further clarification on several points.
> > >
> > > ---
> > >
> > > > **Why this is not just applying standard multi-turn RL on InftyThink.**
> > >
> > > In conventional multi-turn RL, each rollout produces a single sequence consisting of multiple turns. In contrast, the context-refresh mechanism in InftyThink causes a single rollout to generate **multiple independent sequences**, each conditioned on a compressed summary rather than the full history. The model must jointly learn to produce compressed intermediate states (summaries) and to reason effectively from them, creating a tight coupling between generation and conditioning that does not arise in standard agentic RL settings where the environment provides observations. Our contribution lies in developing an end-to-end RL framework that jointly optimizes **when to summarize, what to preserve, and how to continue** under this unique structure.
> > >
> > > > **Empirical evidence of qualitative capability changes.**
> > >
> > > To demonstrate that this framework produces qualitative capability changes rather than expected quantitative improvements, we provide three new pieces of evidence (also presented in our response to Reviewer SfnC):
> > >
> > > *RL learns when to compress.* We replace the model's autonomous summarization timing with two rule-based strategies: Fixed and Random. The SFT-based model **fails to outperform these naive baselines** on average (48.55 vs. 48.53 vs. 48.34), indicating no meaningful timing policy was learned through SFT:
> > >
> > > | Strategy | MATH500 | AIME24 | AIME25 | AMC23 | GPQA | MathOdyssey | Average |
> > > | - | - | - | - | - | - | - | - |
> > > | SFT-based InftyThink | 86.54|29.48|27.92|71.64|14.17|61.56|48.55|
> > > | Random | 86.68 | 28.54 | 26.25 | 72.58 | 14.23 | 62.91 | 48.53 |
> > > | Fixed | 86.91 | 28.44 | 26.04 | 72.03 | 14.40 | 62.21 | 48.34 |
> > >
> > > After RL, adaptive timing outperforms Fixed/Random by **2.50–3.02% on AIME24** and **2.00–2.83% on AIME25** (Table 2). This gap emerges entirely from RL optimization, not from the format itself.
> > >
> > > *RL learns how to compress.* We replace the model's summaries with those from a strong external LLM. Before RL, this substitution **consistently improves performance** (48.55 → 49.73), showing that SFT-trained summaries are suboptimal in content despite being correct in format:
> > >
> > > | Strategy | MATH500 | AIME24 | AIME25 | AMC23 | GPQA | MathOdyssey | Average |
> > > | - | - | - | - | - | - | - | - |
> > > | SFT-based InftyThink | 86.54 | 29.48 | 27.92 | 71.64 | 14.17 | 61.56 | 48.55 |
> > > | External Summarizer | 87.38 | 32.40 | 28.75 | 73.75 | 13.54 | 62.58 | 49.73 |
> > >
> > > After RL, **the same replacement degrades AIME24 from 50.94 to 48.42** (Table 3). This reversal is the most critical evidence of the tight coupling we described above: RL **fundamentally changes the nature of the summarization component** — from format imitation into a policy that co-adapts with continuation. The model simultaneously learns what information to preserve and how to leverage it, which is precisely the challenge unique to the compress-then-continue structure.
> > >
> > > *RL learns how to continue.* We feed SFT-model summaries into the pre-SFT base model for continuation. The base model **matches or outperforms the SFT model**, indicating SFT teaches continuation format rather than effective strategy:
> > >
> > > | Strategy | MATH500 |AIME24| AIME25 | AMC23 |GPQA|MathOdyssey|Average|
> > > | - | - | - | - | - | - | - | - |
> > > | SFT-based InftyThink |86.54|29.48|27.92|71.64|14.17|61.56|48.55|
> > > | continue@iter1|85.14|26.46|24.58|69.37|12.29|58.79|46.10|
> > > | continue@iter2|86.25|28.02|26.56| 69.22|13.02|60.09|47.19|
> > > | continue@iter3|86.88|28.12|26.25|72.73|13.53|61.47|48.16|
> > > | continue@iter4|87.07|30.10|26.56|72.81|14.02|62.04|48.76|
> > >
> > > After RL, InftyThink+ significantly outperforms vanilla continuation (**50.94 vs. 43.75 on AIME24**, Figure 2b), confirming that RL produces a co-adapted continuation strategy that cannot be replicated by models outside the framework.
> > >
> > > > **About the design of efficiency reward.**
> > >
> > > We acknowledge that the current design is empirical. Nevertheless, it yields a **67% latency reduction** (from 149.44s to 48.37s) while still **outperforming vanilla RL in accuracy** (50.58 vs. 47.31), demonstrating clear practical value. We will discuss the limitations and potential directions for more principled approaches in the revised paper.
> > >
> > > ---
> > >
> > > We hope this reply, together with the new empirical evidence, demonstrates that InftyThink+ constitutes a substantive advance. We would be grateful if you could reconsider the score.
> > >
> > > > **Update:** We would also like to note that Reviewer SfnC raised a similar major concern. After reading our analysis of the empirical gap in current SFT-based iterative reasoning, they considered this paper to **meet the requirements for a conference paper and updated the score from `3 to 4`.**
> > >
> > > Thank you again for your time and effort throughout the review process.
> > >
> > > Sincerely,
> > > InftyThink+ Authors

---

### Official Review · Reviewer_HRS2 · 2026-03-13

**Soundness:** 3
**Presentation:** 4
**Significance:** 3
**Originality:** 3
**Overall Recommendation:** 4
**Confidence:** 4

**Summary:**

This paper proposes InftyThink+, a two-stage training framework for iterative reasoning in LLMs. The approach builds on the InftyThink paradigm, which decomposes a single long CoT into multiple shorter reasoning iterations connected by textual summaries. The key contribution is replacing the purely supervised learning approach of the original InftyThink with an end-to-end RL framework that optimizes the full iterative reasoning trajectory. The training proceeds in two stages: (1) a cold-start phase using SFT on transformed data to teach the model the iterative format, and (2) an RL phase using GRPO with trajectory-level shared advantages to optimize when to summarize, what to preserve, and how to continue reasoning. The authors also introduce an efficiency reward based on quadratic decay over iteration count. Experiments on DeepSeek-R1-Distill-Qwen-1.5B and Qwen3-4B-Base show improvements over both SFT-only iterative reasoning and vanilla long-context RL on mathematical and scientific benchmarks with reductions in inference latency.

**Compliance With Llm Reviewing Policy:**

Affirmed.

**Key Questions For Authors:**

Please see weaknesess

**Limitations:**

yes

**Strengths And Weaknesses:**

Strengths
- The three challenges of long-context reasoning (quadratic cost, context limits, lost-in-the-middle) are clearly discussed, and the paper makes a convincing argument for why supervised learning is insufficient for iterative reasoning. The key argument is that "when to summarize, what to preserve, how to continue" are sequential decisions where RL is good at.

- The paper includes carefully designed ablations that study each component of the framework. The "when to compress" ablation comparing adaptive vs. fixed vs. random timing, the "how to compress" ablation swapping internal vs. external summaries, and the "how to continue" analysis each provide clean evidence for the claimed benefits. Table 3 shows external summaries help before RL but hurt after RL — is the evidence that RL learns tightly coupled summarization-continuation strategies.

- The results demonstrate improvements on both accuracy and efficiency simultaneously. The 21% improvement on AIME24 and the 30-40% latency reduction are substantial.

- The paper provides a comprehensive set of supplementary materials: stability analysis across multiple runs (Appendix J), detailed training dynamics (Appendix H), hyperparameter ablations (Appendix O), theoretical analysis (Appendix K), RL-without-cold-start experiments (Appendix L), etc. Those details strengthens reproducibility and trust in the results.

Weakness
- This paper is limited in experiments scale. All experiments are conducted on 1.5B and 4B parameter models. It remains unclear whether the benefits of InftyThink+ persist at larger scales, where models may have stronger long-context capabilities and the relative benefit of iterative summarization could diminish. The paper would be substantially strengthened by one experiment on a larger model. Evaluation benchmarks can be improved as well. The mathematical benchmarks dominate the analysis and are also in-distribution for the RL training data (DeepScaleR). The OOD evaluations on other domains are somewhat shallow (HumanEval and MBPP are relatively easy coding benchmarks). More challenging OOD tasks (e.g., long-horizon agentic tasks, or tasks that require many reasoning iterations) would better demonstrate the claimed "infinite-horizon" reasoning capability.

- The cold-start stage introduces significant dependencies and complexity. The data transformation pipeline requires an external LLM (Qwen3-4B-Instruct) to generate summaries, a segment length hyperparameter, a summary length constraint, and careful prompt engineering (two different prompts for different iterations). Appendix L shows that RL without cold start collapses, which means the method is fundamentally dependent on this supervised initialization. This raises a question: how sensitive is the final performance to the quality and design choices of the cold-start stage?

- The efficiency reward design may be too simplistic. The quadratic decay reward penalizes iteration count uniformly regardless of problem difficulty. There is no mechanism for the model to receive a higher reward for appropriately using more iterations on harder problems versus fewer on easier ones. Table 1 shows that T+E consistently underperforms T on accuracy, so efficiency reward may be trading off accuracy in some cases.

Minor: The experimental comparison in Figure 24 uses DeleThink results "reproduced from their paper." It is unclear whether DeleThink's hyperparameters (chunk size C, context window m) were optimally tuned for this comparison. There is no quantitative analysis of the generated summaries before and after RL. Are RL-trained summaries shorter or more structured? Do they preserve different types of information? The paper notes that the AIME benchmarks contain only 30 problems each, which introduces high variance. Some of the claimed improvements (e.g., the 21% on AIME24) should be interpreted cautiously given this small sample size.

---

> ### Author Rebuttal · Authors · 2026-03-31
>
> Dear Reviewer HRS2,
>
> Thank you for the effort and time you dedicated to reviewing our paper, and for your recognition of our motivation and contribution. We have greatly benefited from your valuable feedback. Below are our responses to the weaknesses you pointed out.
>
> ---
> > **[W1] Limited experiments scale.**
>
> Thank you for the question. We want to show two additional experiments that more clearly demonstrate the effectiveness of InftyThink+.
>
> + **Scaling to a larger model.** During the rebuttal period, we additionally evaluated a stronger MoE base model, Qwen3-30B-A3B-Base (https://anonymous.4open.science/r/InftyThink-Plus-Rebuttal-AECD/B.pdf). We observe gains that are consistent with those reported in the main paper. Due to time and resource constraints, however, we trained this model for only 400 RL steps.
> + **More OOD evaluations.** Our current experimental setting does not yet support agentic tasks, so we are unable to include such evaluations at this stage. **We have therefore discussed agentic tasks as a future direction in Appendix A.3**. Instead, we additionally evaluate on BBH and BBEH, two benchmarks that require substantially more reasoning steps. **We again consistent improvements from InftyThink+**. The detailed results are provided in https://anonymous.4open.science/r/InftyThink-Plus-Rebuttal-AECD/D.pdf.
>
> > **[W2] Significant dependencies and complexity of cold-start stage.**
>
> Thank you for raising this important concern. We agree that InftyThink+ depends on a cold-start stage, which introduces additional components and complexity. That said, we would like to emphasize that such supervised initialization is now common in frontier reasoning models such as DeepSeek-R1 and Qwen3. In our case, the added cold-start procedure is a lightweight data transformation pipeline that is fast to construct and does not rely on fragile design choices.
>
> More importantly, our results suggest that it is not overly sensitive to the specific design of the cold-start stage. Prior work has already shown that this pipeline is broadly robust across choices such as hyperparameters, summarizer models, and training data. To further address this concern, we conducted two additional experiments during rebuttal by varying the summarizer model and the segment-length hyperparameter (https://anonymous.4open.science/r/InftyThink-Plus-Rebuttal-AECD/E.pdf). The resulting performance changes are stable and moderate, indicating that InftyThink+ does not hinge on a narrowly tuned cold-start configuration. Instead, the main role of cold start is to provide paradigm transfer, that is, to initialize the model into the iterative reasoning format, while the subsequent RL stage is responsible for learning the actual decision policy.
>
> > **[W3] The efficiency reward design may be too simplistic.**
>
> Thank you for this insightful question. We would like to clarify that it is not necessary to design a mechanism that receive different efficiency reward based on query difficulty, for two main reasons:
>
> + **GRPO already compares trajectories on a per-query basis.**
> In GRPO, the advantage of each trajectory is computed relative to other trajectories sampled from the same query, and reward scale differences across queries of different difficulty are naturally canceled out during advantage normalization. In other words, the optimization signal focuses on which trajectories are better for the same problem, rather than on absolute reward magnitudes across different problems.
> + **The efficiency reward is introduced multiplicatively and only matters when the answer is correct.**
> As shown in Eq. (5), the efficiency reward only takes effect when a trajectory already solves the problem correctly. The goal is to encourage the model to reduce unnecessary reasoning rounds while preserving correctness, thereby improving efficiency.
>
> During the rebuttal period, we conducted additional ablations on the design of the efficiency reward; for more details, please refer to `[Q2@RZoM]`.
>
> > **[M1] Details of DeleThink comparision.**
>
> We apologize for the unclear presentation here. In Appendix P.2, our comparison with Delethink uses the metrics reported in their paper rather than results reproduced by us, which is why the curve in Figure 24 appears truncated. **We directly adopt Delethink's best reported performance without any additional hyperparameter selection or special tuning on our side.**
>
> > **[M2] Quantitative analysis of the generated summaries.**
>
> We have conducted this analysis. Due to space limitations, please refer to https://anonymous.4open.science/r/InftyThink-Plus-Rebuttal-AECD/F.pdf for the corresponding tables and figures.
>
> > **[M3] Evaluation fluctuation.**
>
> Thanks for the question. Due to character limits, please refer to `[W4@fUNs]`.
>
> ---
> Thank you again for the thorough review. We think that our rebuttal can address your main concern, if you have any further questions, we would be happy to answer them.
>
> Sincerely,
> InftyThink+ Authors

---

> > ### Author Rebuttal · Reviewer_HRS2 · 2026-04-04
> >
> > The rebuttal improves my understanding of the apper. I maintain the positive rating.

---

> > > ### Author Response · Authors · 2026-04-04
> > >
> > > Dear Reviewer HRS2,
> > >
> > > We sincerely thank you for your response. We are glad that our rebuttal has fully addressed your concerns.
> > >
> > > In light of this, **we would like to kindly ask whether you would consider updating your rating**, potentially after the discussion phase with other reviewers, to better reflect the consensus reached during the rebuttal.
> > >
> > > > **[Update]** We would also like to note that, after reading our analysis of the empirical gap in current SFT-based iterative reasoning, Reviewer `SfnC` and Reviewer `RZoM` considered this paper to meet the requirements for a conference paper and increased the score.
> > >
> > > We greatly appreciate the time and effort you have dedicated to reviewing our work.
> > >
> > > Sincerely,
> > > InftyThink+ Authors

---

### Decision · Program_Chairs · 2026-04-30

**Decision:**

Accept (regular)

**Comment:**

The paper trains with RL models which summarize their own context to extend the prior line of research InfinyThink. This is a key contribution towards more efficient use of test time compute, but the methodological novelty is fairly limited to the application of RL techniques